# WebGen-R1: Incentivizing LLMs to Generate Functional and Aesthetic Websites with Reinforcement Learning

## Abstract

Large Language Models (LLMs) have demonstrated strong capabilities in functional-level code generation, yet their performance remains limited in project-level scenarios such as generating large-scale multi-page websites. Such tasks require coherent multi-file structures, handling of intricate cross-page dependencies, and visually appealing designs. Prior works address only partial aspects of this challenge. For instance, WebDev Arena[1] focuses exclusively on single-page static sites, while agent-based frameworks decompose tasks into subtasks coordinated through multi-turn execution, often relying on proprietary models and suffering from fragile integration, particularly in visual coherence and stylistic consistency. In this work, we introduce WebGen-R1, pushing toward a more ambitious and practically relevant goal of training a small-scale LLM via reinforcement learning (RL) to generate the entire multi-page websites in an end-to-end manner. A key obstacle lies in reward design. Unlike functional code generation where correctness can be verified by passing automated test suites, web aesthetics covering layout harmony, typographic consistency, and stylistic alignment are inherently subjective, and functional verification often requires dynamic execution across pages where rule-based reward function tend to be brittle. To address these limitations, we design a vision–language–model-based reward model that jointly optimizes functional correctness and aesthetic quality, enabling the model to produce websites that are both visually coherent and faithful to the intended task specification. Extensive experiments across real-world benchmarks demonstrate that WebGen-R1 consistently outperforms, or is comparable to, strong proprietary and open-source baselines in a multi-dimensional evaluation protocol. To facilitate future research in end-to-end multi-page website generation, we release our code and data at `https://anonymous.4open.science/r/WebGen-R1`.

## 1 Introduction

Recent advances in Large Language Models (LLMs) have markedly expanded their capabilities in automated code generation (Jaech et al., 2024; Hui et al., 2024; Jiang et al., 2025; Guo et al., 2025; Team et al., 2025; Yang et al., 2025a; Zheng et al., 2025), achieving human competitive performance on established functional-level benchmarks such as the HumanEval (Chen et al., 2021) dataset or even complex International Olympiad in Informatics (IOI) programming tasks (Li et al., 2022). These achievements demonstrate impressive syntactic and semantic reasoning over standalone problems. However, moving from constrained, function-level snippets to project-level code generation that meets real-world software engineering demands remains a frontier challenge (Jimenez et al., 2023; Bi et al., 2024; Zan et al., 2025; Badertdinov et al., 2025). Among the various categories of such tasks, end-to-end website generation, encompassing multi-page routing, dynamic functionality, modern user interface (UI) design, and responsive layouts, presents especially rich and consequential opportunities for LLM research. Website generation represents a particularly challenging instantiation of project-level code generation (Wan et al., 2024; Xiao et al., 2024; Lu et al., 2025; Zhang et al., 2025). Unlike single-function problems, real-world websites demand consistent architectural patterns, multi-file codebases with intricate dependencies, long-range contextual coherence,

---

[1]https://blog.lmarena.ai/blog/2025/webdev-arena

and conformity to design principles that balance functionality with visual appeal. This necessitates reasoning not only over software engineering constraints but also over aesthetic and user-experience considerations, which have traditionally been difficult to formalize and evaluate in automated code generation.

Despite promising early steps, current approaches to LLM-driven website generation exhibit notable limitations. One line of work has chosen to simplify the generation task drastically. For example, WebDev Arena (LMArena, 2025) is constrained to generating single-page static sites. While this makes evaluation tractable, it abstracts away essential complexities such as dynamic routing, state management, authentication flows, and cross-page navigation. Another line of research adopts multi-agent orchestration frameworks, in which different specialized LLMs handle discrete subtasks (UI layout, backend logic, testing), and their outputs are subsequently integrated (Hong et al., 2023; He et al., 2024a;b; Lu et al., 2025). However, such modularity introduces brittle inter-agent dependency chains, where small inconsistencies in contracts, file names, or interface definitions can cascade into non-functional builds. Moreover, both paradigms rarely incorporate formal optimization of aesthetic quality or human-aligned design sensibilities, an omission that leads to websites which, while functional, often fail to meet the expectations of end-users in visual polish. Further discussion of related work is provided in Appendix D.

In this work, we push toward a more ambitious and practically relevant goal of *training a small-scale LLM to generate an entire multi-page, functional, and visually aesthetic website project from scratch, in an end-to-end manner, without external decomposition into subtasks*. This departure invites several formidable challenges: **1** Global structural reasoning over project-level architectures, including framework-specific conventions (e.g., Next.js routing, Vue plugin registration) and modular directory organization. **2** Maintaining multi-file consistency for cross-referencing components, dependencies, and dynamic import paths across the codebase. **3** Ensuring cohesive visual design and modern aesthetics, beyond merely placing elements on a page, by rendering layouts that exhibit balance, alignment, accessible color contrasts, and brand coherence. **4** Capturing complex interactive behaviors such as animations, drag-and-drop, and responsive state updates within the generated code. **5** Overcoming limited long-context reasoning, as project code often exceeds the context window available in current LLMs, creating difficulties in tracking dependencies over hundreds of lines across multiple files.

To address these challenges, we propose WebGen-R1, a novel framework that integrates reinforcement learning (RL) directly into the end-to-end website generation process. Nevertheless, a central obstacle in bringing RL into such open-ended generative tasks lies in the design of a reliable reward signal (Guo et al., 2025; Zeng et al., 2025; Wen et al., 2025; Yang et al., 2025b; Mroueh, 2025). Different from tasks like complex mathematics and competitive programming, where correctness is objectively verifiable by exactly matching unambiguous ground-truth answers or passing automated test suites, it faces a serious challenges with website generation: **A** Not all desired qualities are reducible to scalar outcomes, as visual appeal, design cohesion, and user experience have no trivial Boolean test. **B** Comprehensive functional verification for complex, multi-page websites often requires running the site and inspecting behaviors in varied scenarios, making static rule definition brittle. **C** The scarcity of annotated, fully verifiable website outputs limits scalability for this domain.

To overcome these limitations, we replace handcrafted rule-based scorers with a reward model that incorporates both task specification cognition and visual rendering perception. Concretely, after the model generates the complete web project code and directory structure, we execute a standardized front-end development pipeline, which includes parsing and verifying the scaffolded file organization, installing dependencies, building the project, launching a local development or production server, and rendering the pages in a browser. The rendered page screenshots, together with the original task specification, are then fed into a state-of-the-art vision-language model (VLM) for joint functional and aesthetic evaluation, producing a graded scalar reward. This pipeline preserves verifiability for functional aspects while enabling nuanced assessment of open-ended design quality, making large-scale RL training both stable and meaningful.

We conduct extensive experiments on real-world benchmarks to evaluate the effectiveness of WebGen-R1. When employing Qwen2.5-Coder-Instruct-7B as our base model, WebGen-R1 exhibits substantial gains in multi-page website generation, achieving an improvement in functional quality metrics from 1.59% to 29.21%, a 44.32% increase in aesthetic scoring, and a drastic increase

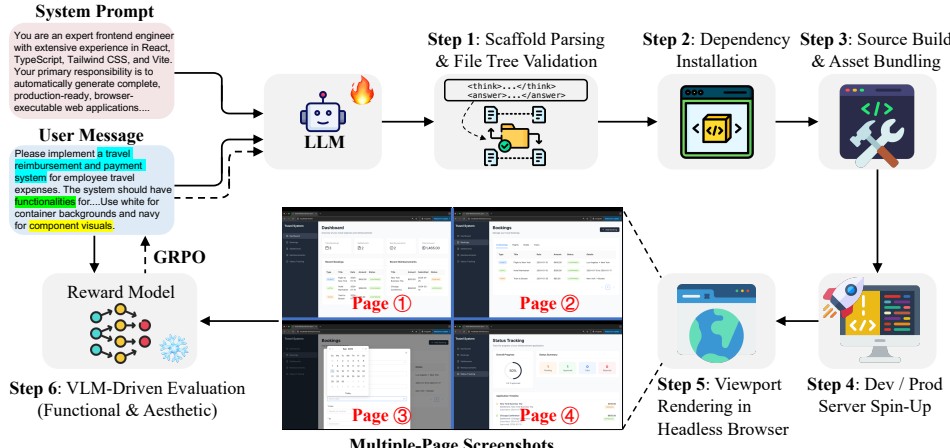

Figure 1: The architecture of our proposed WebGen-R1 for end-to-end multi-page website generation using a single LLM. A user provides a natural-language website design request, which the LLM generates the entire project source code. The generated scaffold is then processed through a standardized front-end build pipeline, comprising parsing and validation of the file organization, dependency installation, project build, server spin-up (in development or production mode), and rendering of the resulting webpages in a headless browser environment to obtain multiple page screenshots. These rendered page images, along with the original user request, are subsequently fed into a state-of-the-art VLM for joint evaluation of functional correctness and visual design quality, producing a scalar reward signal in the discrete range 0-5. This automated evaluation framework ensures both objective verification of executable functionality and nuanced assessment of open-ended aesthetic attributes. The reward signal is then used to fine-tune the LLM via the GRPO objective, enabling stable and semantically meaningful reinforcement learning at scale.

in valid render ratio from 30.56% to 95.89%. Importantly, despite having significantly fewer parameters, WebGen-R1 attains aesthetic quality and valid render ratios that surpass those of much larger and more powerful baselines such as Gemini-2.5-Pro and DeepSeek-R1, while maintaining comparable performance in functional evaluation. These results provide strong empirical evidence that our WebGen-R1 with reinforcement learning can effectively enhance both the functional reliability and visual appeal of generated websites. In summary, our contributions are as follows:

- To our best knowledge, we are the first to introduce reinforcement learning, namely WebGen-R1, for end-to-end generating the entire multi-page websites without relying on task decomposition or proprietary model orchestration.

- We design a vision–language-model-based reward model that jointly measures functional correctness and aesthetic quality, capturing layout harmony, typographic consistency, and stylistic alignment. This resolves the brittleness of rule-based rewards in project-level code generation and aligns optimization signals with human preferences.

- We establish a multi-dimensional evaluation protocol covering functional correctness, visual coherence, deployability, and human-perceived quality, and demonstrate through extensive experiments on real-world benchmarks that our WebGen-R1 outperforms or matches advanced proprietary and open-source baselines, even in challenging scenarios requiring fine-grained inter-page coordination.

- We release our codebase, datasets, RL training framework, and model checkpoints to enable reproducibility and to foster future research on end-to-end, LLM-driven multi-page website generation.

## 2 METHODOLOGY

### 2.1 END-TO-END OPEN-ENDED WEBSITE CODE GENERATION

Given a task specification $x \in \mathcal{X}$, the policy $\pi_\theta$ must produce an entire project $y \in \mathcal{Y}$ containing a coherent multi-file structure, framework-specific directories, and interdependent code mod-

ules. Unlike template-based generation, the space $\mathcal{Y}$ is open-ended $\mathcal{Y} = \bigcup_{K=1}^{\infty} \mathcal{Y}_K, \quad \mathcal{Y}_K = \{y :$ $y$ contains $K$ valid files in a consistent project graph}. We model generation as an autoregressive process $\pi_\theta(y \mid x) = \prod_{t=1}^{T} \pi_\theta(y_t \mid y_{<t}, x)$, where the token sequence $y_{1:T}$ is subsequently parsed into a file set $\mathcal{F}(y) = \{f_k\}_{k=1}^{K}$ and a directory graph $\mathcal{S}(y)$. We denote the resulting website project instance as $\mathcal{W}eb(y) = \{\mathcal{F}(y), \mathcal{S}(y)\}$. By emitting all code in one globally conditioned sequence, the policy is forced to maintain cross-file variable references, routing conventions, and naming coherence. These properties are frequently broken in iterative, file-by-file generation, and their absence leads to non-functional builds. This holistic treatment is particularly advantageous for web projects, where front-end routing, shared styles, and state management are tightly coupled.

## 2.2 AUTOMATED FRONT-END BUILD AND RENDERING PIPELINE

A generated website codebase carries no operational meaning until it is executed within an actual front-end development pipeline. We model this execution as a deterministic environment transformation:

$$o = \mathcal{E}(y) = \mathcal{R}(\mathcal{L}(\mathcal{B}(\mathcal{I}(\mathcal{W}eb(y))))), \tag{1}$$

where $\mathcal{I}$ installs dependencies, $\mathcal{B}$ builds the compiled bundle, $\mathcal{L}$ launches a server, and $\mathcal{R}$ renders pages through a headless browser. As a result, the output $o$ contains:

$$o = \left(\{I_p\}_{p=1}^{P}, \Gamma\right), \tag{2}$$

with $\{I_p\}_{p=1}^{P}$ denoting rendered screenshots for $P$ routes and $\Gamma$ collecting build and runtime logs. This execution step enforces realizability constraints so that only codebases that install, build, and run successfully yield complete visual evidence, directly grounding learning signals in executable behavior. In website generation, this is critical as projects that pass linters can still fail at runtime due to subtle integration errors, broken imports, or misconfigured frameworks.

## 2.3 REWARD MODEL FOR FUNCTIONAL AND AESTHETIC EVALUATION

While functional correctness covering aspects such as hyperlink validity, navigation integrity, and component responsiveness can be partially verified via deterministic, rule-based checks, *purely algorithmic verification fails to capture subjective, perceptual qualities of web design*. Aesthetic factors, including layout harmony, typographic consistency, and stylistic coherence with the task specification, require perceptual evaluation signals that are inherently non-deterministic. To jointly capture both dimensions, we propose a multimodal reward evaluator $\phi_\psi$ built upon a state-of-the-art vision–language model (VLM). Formally, given an input tuple $z = \left(x, \{I_p\}_{p=1}^{P}, \Gamma\right)$, where $x$ denotes the natural language specification, $\{I_p\}_{p=1}^{P}$ are rendered page snapshots from multiple responsive breakpoints, and $\Gamma$ represents runtime execution logs, the evaluator jointly attends to textual, visual, and behavioral modalities to produce a scalar score:

$$s_{\langle \text{func,vis} \rangle} = g_{\phi_\psi}[s_{\text{func}}, \ s_{\text{vis}}] = \phi_\psi^{\text{VLM}}(z), \tag{3}$$

where $s_{\text{func}}$ measures functional soundness and $s_{\text{vis}}$ evaluates visual design quality. The unified score $s_{\langle \text{func,vis} \rangle} \in [0, 5]$ thus reflects both executable fidelity and human-perceived aesthetics in a single quantitative metric.

Beyond these two primary criteria, we incorporate additional signals to enforce structural and reasoning quality in generated code. Inspired by (Guo et al., 2025), we assess code format correctness $s_{\langle \text{code} \rangle}$, where the generated website code is parsed into a file set $\mathcal{F}(y) = \{f_k\}_{k=1}^{K}$ and a corresponding directory graph $\mathcal{S}(y)$. This check ensures that the output can be executed without modification, providing a direct, verifiable reward signal for structural validity. Moreover, to incentivize long-horizon and explicit reasoning in project organization, such as planning directory hierarchies, configuring frameworks appropriately, and maintaining coherent shared state, we define a reasoning format reward $s_{\langle \text{cot} \rangle}$. Here, the model is required to externalize its reasoning process between dedicated <think>...</think>tags, enabling downstream parsing and reward estimation on reasoning quality. Finally, these components are fused into a multi-dimensional scalar reward:

$$R(y) = s_{\langle \text{func,vis} \rangle} + \gamma \, s_{\langle \text{code} \rangle} + \lambda \, s_{\langle \text{cot} \rangle}, \tag{4}$$

with weight coefficients $\gamma, \lambda \in [0, 1]$ controlling the contribution of code format and reasoning quality to the overall signal. Thus, in the context of LLM-based website generation, this integrative

scoring mitigates the common pitfall of over-optimizing for mechanically verifiable checks at the expense of human-perceived functionality and visual appeal, thereby aligning model outputs more closely with end-user expectations.

## 2.4 GROUP RELATIVE POLICY OPTIMIZATION FOR WEBSITE GENERATION

We formulate our reinforcement learning objective as maximizing the expected task-specific reward under the learned policy $\pi_\theta$ $\max_\theta$ $\mathbb{E}_{x\sim\mathcal{D},\ y\sim\pi_\theta(\cdot|x)}\big[R(y)\big]$, where $x$ denotes a website generation prompt, $y$ is a candidate structured website output, and $R(\cdot)$ measures a composite reward capturing both functional correctness and visual quality (see Eq. 4). Unlike standard PPO(Schulman et al., 2017), GRPO (Shao et al., 2024; Guo et al., 2025; Yu et al., 2025) removes the need for an explicit value function by normalizing rewards within a group of sampled responses for the same prompt, enabling more stable optimization and mitigating inter-prompt variance in difficulty. Specifically, for each website specification $x$, we sample a group of $G$ candidate responses $\{y_i\}_{i=1}^G$ from the behavior policy $\pi_{\theta_{\text{old}}}(\cdot\mid x)$, where each sequence is tokenized as $y_i = (y_{i,1}, y_{i,2}, \ldots, y_{i,|y_i|})$. We obtain a scalar reward $R_i$ for each $y_i$ by executing the generated website and evaluating functional validation and visual fidelity score. Following PPO-style clipped surrogate objective with an explicit KL penalty to constrain divergence from a frozen reference policy $\pi_{\text{ref}}$, the GRPO objective is:

$$\mathcal{J}_{\text{GRPO}}(\theta) = \mathbb{E}_{x\sim\mathcal{D},\ \{y_i\}_{i=1}^G\sim\pi_{\theta_{\text{old}}}(\cdot|x)}\left[\frac{1}{G}\sum_{i=1}^G \frac{1}{|y_i|}\sum_{t=1}^{|y_i|}\right.$$

$$\left(\min\left[r_{i,t}(\theta)\,\hat{A}_{i,t}, \text{clip}\big(r_{i,t}(\theta), 1-\varepsilon, 1+\varepsilon\big)\,\hat{A}_{i,t}\right] - \beta\, D_{\text{KL}}\big(\pi_\theta(\cdot\mid x)\,\|\,\pi_{\text{ref}}(\cdot\mid x)\big)\right)\bigg], \tag{5}$$

$$\hat{A}_{i,t} \triangleq \frac{R_i - \text{mean}(\{R_j\}_{j=1}^G)}{\text{std}(\{R_j\}_{j=1}^G)}, \quad r_{i,t}(\theta) \triangleq \frac{\pi_\theta\big(y_{i,t}\mid x, y_{i,<t}\big)}{\pi_{\theta_{\text{old}}}\big(y_{i,t}\mid x, y_{i,<t}\big)}, \tag{6}$$

where $\hat{A}_{i,t}$ denotes the group-relative normalized advantage for token position $t$ in response $i$, $\varepsilon > 0$ is the clipping parameter, and $\beta$ controls the strength of the KL regularization. The detailed procedure of the algorithm is presented in Algorithm 1 (see Appendix A).

# 3 EXPERIMENTS

## 3.1 EXPERIMENTAL SETUP

**Datasets and Benchmarks.** We leverage the WebGen-Instruct (Lu et al., 2025) as our training corpus. This dataset comprises 6,667 end-to-end website generation tasks spanning a broad spectrum of real-world web application domains. For evaluation, we employ the WebGen-Bench (Lu et al., 2025) benchmark, which contains 101 carefully curated website generation tasks. These tasks range from minimal single-page designs to complex corporate websites with rich interactivity and data-driven dashboards. Each benchmark instance is paired with a comprehensive, repeatedly validated test suite, ensuring reliable measurement of both the functional behavior and stylistic conformance of generated websites. The natural-language task descriptions explicitly specify functional requirements and visual design expectations, enabling precise evaluation. The detailed statistics of the dataset and the benchmark are summarized in Table 5 (Appendix B). Figure 2 illustrates the distributions of prompt and response lengths, where each prompt is obtained by concatenating the system prompt with the web task query, for several state-of-the-art commercial LLMs on the end-to-end website generation task.

**Baselines.** We benchmark WebGen-R1 against a broad spectrum of state-of-the-art LLMs, encompassing eight proprietary models and seven open-source models. We exclude WebGen-LM (Lu et al., 2025) from comparison as it is specifically fine-tuned on Bolt.diy [2] website-generation trajectories gathered via DeepSeek-V3 (Liu et al., 2024), making it bound to a particular agent-based framework and not directly applicable in our evaluation setting.

**Metrics.** To thoroughly evaluate LLMs for end-to-end website generation, we use several quantitative metrics, including (1) Functional Success Rate (FSR): the percentage of generated websites

---

[2]https://github.com/stackblitz-labs/bolt.diy

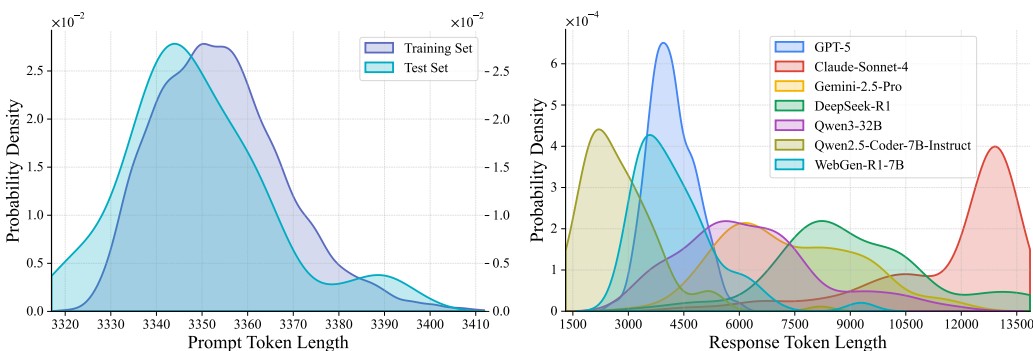

Figure 2: Token length distributions of prompts and generated responses for several state-of-the-art LLMs on the end-to-end multi-page website generation task across WebGen-Instruct (training set) and WebGen-Bench (test set). The response length distributions are aggregated over test set.

Table 1: Performance comparison of WebGen-R1 and various state-of-the-art LLMs from multiple institutions on the WebGen-Bench benchmark, evaluated by FSR, AAS, and VRR metrics. We show the score improvement ($\pm$) of our model relative to its base. Bold values indicate the best results.

| Institution | Model | FSR(%) | AAS | VRR(%) |
|---|---|---|---|---|
| OpenAI | GPT-5 | 46.53 | 3.34 | 90.43 |
| | GPT-4.1 | 43.91 | 3.78 | 82.09 |
| | o3 | 42.86 | 3.55 | 81.08 |
| | o4-mini | 27.29 | 3.31 | 56.52 |
| | GPT-4o | 21.60 | 3.31 | 85.71 |
| Anthropic | Claude-Sonnet-4 | 46.13 | 3.86 | 86.05 |
| | Claude-3.7-Sonnet | **57.72** | 3.90 | 84.00 |
| Google | Gemini-2.5-Pro | 36.31 | 3.89 | 83.33 |
| DeepSeek | DeepSeek-R1 | 30.25 | 3.67 | 42.86 |
| Alibaba | Qwen2.5-Coder-7B-Instruct | 1.59 | 2.73 | 30.56 |
| | Qwen2.5-72B-Instruct | 2.54 | 3.14 | 8.86 |
| | Qwen3-8B | 3.72 | 2.57 | 12.50 |
| | Qwen3-32B | 18.69 | 3.39 | 59.42 |
| | Qwen3-30B-A3B-Thinking-2507 | 9.30 | 2.60 | 23.81 |
| | Qwen3-Coder-30B-A3B-Instruct | 6.06 | 2.90 | 32.81 |
| Ours | **WebGen-R1-7B** | 29.21₊₂₇.₆₂ | **3.94**₊₄₄.₃₂ | **95.89**₊₆₅.₃₃ |

that pass predefined interactive checks such as button clicks and form submissions; (2) Aesthetic Alignment Score (AAS): the average reward model score, measuring how well function and design align with human aesthetic preferences. **It is worth noting that AAS evaluates the alignment of the rendered webpage with user requirements and aesthetic preferences from a visual perspective. Unlike FSR, which directly measures functional correctness, AAS infers certain aspects of functional correctness through visual cues, such as detecting the presence of forms, dropdown menus, or search inputs, and thus serves as a complementary proxy metric;** (3) Valid Render Ratio (VRR): the percentage of generated websites that render without major errors; (4) Lint & Dependency Pass Rate (LDPR): the fraction of projects that pass static code analysis (like ESLint) and resolve dependencies automatically, indicating readiness for deployment. These metrics form a multi-dimensional evaluation protocol that jointly accounts for execution correctness, visual attractiveness, deployability, and human-perceived quality, providing a holistic assessment of LLM-driven website generation.

**Implementation Details.** We conduct all experiments on a cluster equipped with $64\times$ NVIDIA H100 GPUs (80 GB memory each), using the open-source Open-R1 framework [3] (Hugging Face, 2025). We fine-tune the Qwen2.5-Coder-Instruct-7B model for 400 optimization steps

---
[3]https://github.com/huggingface/open-r1

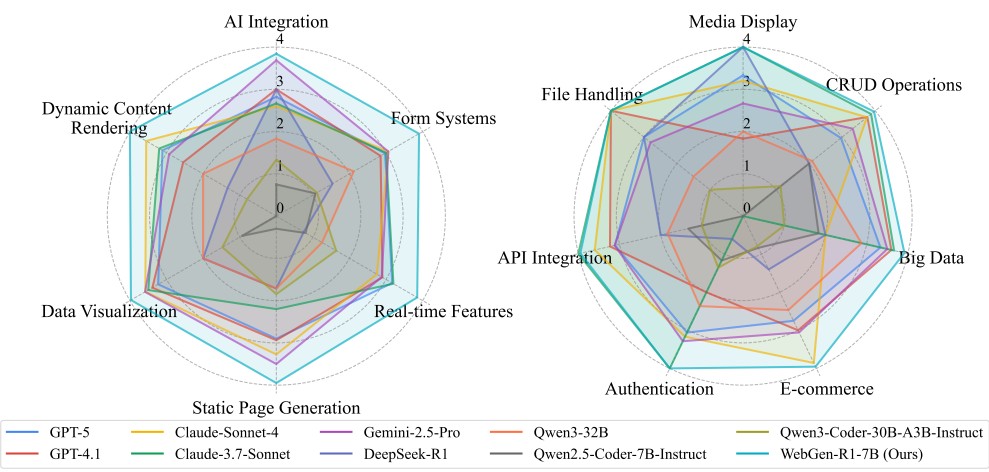

Figure 3: Performance comparison of WebGen-R1 and various state-of-the-art LLMs on 13 hetero-geneous multi-scenario front-end development tasks encompasses diverse requirements, providing a rigorous testbed for both functionality and visual fidelity, sourced from the WebGen-Bench.

Table 2: Category-wise performance comparison of WebGen-R1 and various state-of-the-art LLMs from multiple institutions on the WebGen-Bench benchmark, evaluated by FSR metric, showing score improvement ($\pm$) of our model relative to its base. The first three columns correspond to categories of website-generation instructions, and the last three correspond to categories of test cases, following (Lu et al., 2025).

| Institution | Model | Content Presenta-tion | User Inter-action | Data Manage-ment | Func-tional Testing | Data Display Testing | Design Validation Testing |
|---|---|---|---|---|---|---|---|
| OpenAI | GPT-5 | 55.29 | 46.21 | 36.81 | 36.93 | 53.14 | 62.73 |
| | GPT-4.1 | 46.27 | 31.03 | 61.26 | 38.05 | 44.85 | 57.69 |
| Anthropic | Claude-Sonnet-4 | 52.88 | 39.42 | 53.33 | 41.09 | 47.67 | 55.36 |
| | Claude-3.7-Sonnet | 66.67 | 43.40 | 64.29 | 46.48 | 72.09 | 62.86 |
| Google | Gemini-2.5-Pro | 31.54 | 37.34 | 39.71 | 29.60 | 43.05 | 44.90 |
| DeepSeek | DeepSeek-R1 | 31.11 | 20.45 | 58.62 | 24.69 | 37.25 | 33.33 |
| Alibaba | Qwen2.5-Coder-7B-Instruct | 0.00 | 3.17 | 3.12 | 0.00 | 3.57 | 0.00 |
| | Qwen3-32B | 24.81 | 17.02 | 14.41 | 11.42 | 22.73 | 32.47 |
| | Qwen3-Coder-30B-A3B-Instruct | 16.44 | 1.06 | 8.82 | 2.81 | 13.25 | 5.80 |
| Ours | **WebGen-R1-7B** | 35.29 +35.29 | 27.25 +24.08 | 26.88 +23.76 | 15.90 +15.90 | 30.43 +26.86 | 54.92 +54.92 |

using the GRPO objective. Following (Lu et al., 2025), the VLM reward model leverages a state-of-the-art GPT-4o evaluator to score both the functional correctness and visual aesthetics of generated websites. **Importantly, we do not adopt GUI Agent as a functional verifier because the WebGen-Instruct training dataset lacks standardized test cases required for reliable interaction-based UI evaluation. The detailed comparison of GUI-agent evaluation and VLM-based reward for functional correctness is provided in Appendix E.7.** To ensure stable optimization, the reward function is calibrated to align closely with human preferences through prompt engineering and normalization. More implementation details, including the hyperparameter settings, are provided in Appendix C.

## 3.2 MAIN RESULTS

We compare WebGen-R1 with state-of-the-art LLMs on the WebGen-Bench using FSR, AAS, and VRR to examine whether reinforcement learning can optimize functionality, design, and render reliability. As shown in Table 1, Claude-3.7-Sonnet achieves the highest FSR (57.72%), showing strong execution, while WebGen-R1 gets the highest AAS score (3.94), outperforming all models, includ-

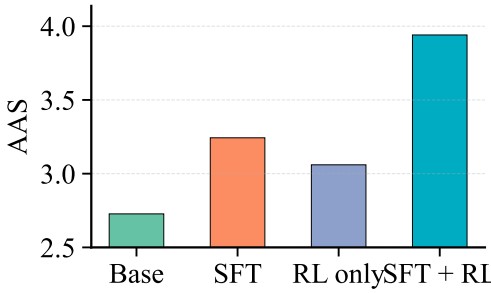
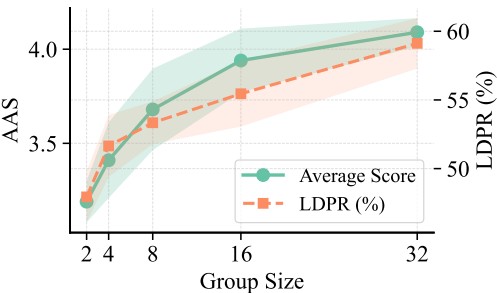

Figure 4: Performance comparison of SFT, RL-only, and SFT+RL on the WebGen-Bench benchmark under the AAS metric.

Figure 5: Impact of group size $G$ in GRPO on WebGen-Bench performance measured by AAS and LDPR metrics.

ing Claude. WebGen-R1 also attains the best VRR (95.89%), clearly higher than all baselines, which we attribute to our reward design focusing on code format correctness. This balance of visual quality and reliable rendering is achieved even though WebGen-R1 is a 7B model, proving the effectiveness of our RL training. Smaller Qwen3 models generally have lower FSR, while some larger variants like Qwen3-32B (AAS 3.39) and Qwen2.5-72B-Instruct (AAS 3.14) reach relatively high AAS. *This pattern suggests that generating functionally correct website is substantially more challenging than achieving visual appeal, as functional requirements in front-end development often involve complex logic and interactions beyond visual layout.* Overall, our RL optimization enables the model to produce visually attractive websites while still maintaining functional correctness, demonstrating that targeted reward shaping can balance these multiple objectives in website generation.

## 3.3 IN-DEPTH ANALYSIS AND INSIGHTS.

**Multi-Scenario Web Environments.** We evaluate WebGen-R1 and a variety of state-of-the-art LLMs on multi-scenario front-end development tasks, including AI Integration, Form Systems, Real-time Features, Static Page Generation, Data Visualization, Dynamic Content Rendering, Media Display, CRUD Operations, Big Data, E-commerce, Authentication, API Integration, and File Handling, sourced from WebGen-Bench (Lu et al., 2025). As shown in Figure 3, our WebGen-R1 achieves superior performance across all 13 categories on AAS metric, which indicates a consistent improvements in both functional correctness and UI/UX quality. Moreover, following (Lu et al., 2025), we report FSR results for each category of website instructions and test cases in Table 2. Compared with the baseline, our WebGen-R1 achieves substantial gains across all categories. Our WebGen-R1 also delivers competitive performance relative to other strong models; for instance, it consistently surpasses DeepSeek-R1 and Gemini-2.5-Pro in the Content Presentation and Design Validation Testing categories. We attribute these improvements to the incorporation of the functional and aesthetic reward design. Such uniform improvements across diverse front-end scenarios demonstrate that our WebGen-R1 is task-agnostic, robust to domain shifts, and effective in harmonizing execution-level correctness with aesthetic alignment, highlighting the practical applicability to real-world, mixed-requirement web development.

Table 3: Performance comparison of SFT, RL-only, and SFT+RL on the WebGen-Bench benchmark under the FSR and VRR metrics.

Table 4: Impact of group size $G$ in GRPO on WebGen-Bench performance measured by FSR and VRR metrics.

| Metrics | Base | SFT | RL only | SFT + RL |
|---------|------|-----|---------|----------|
| FSR | 1.59 | 20.08 | 18.23 | 29.21 |
| VRR | 30.56 | 30.69 | 26.82 | 95.89 |

| Metrics | 2 | 4 | 8 | 16 | 32 |
|---------|---|---|---|----|----|
| FSR | 22.39 | 22.92 | 24.99 | 29.21 | 34.79 |
| VRR | 90.98 | 91.07 | 93.52 | 95.89 | 98.27 |

**RL Fine-Tuning.** We compare RL and supervised fine-tuning (SFT) on the web generation task. Using 600 GPT-4.1-generated examples for SFT, we observe an 18.68% performance boost over the baseline model (Qwen2.5-Coder-7B-Instruct), significantly improving functional and coherent webpage generation, as shown in Figure 4. RL-only (R1-Zero) gives a 12.09% improvement, showing the value of vision–language model rewards for optimizing both function and appearance. SFT

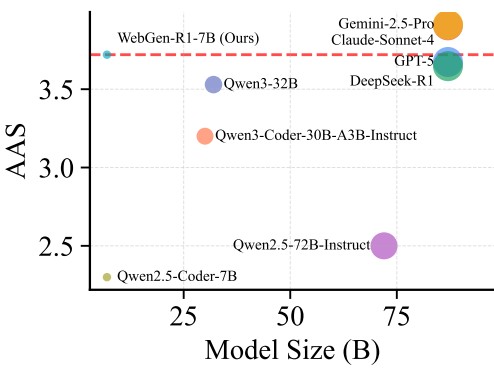 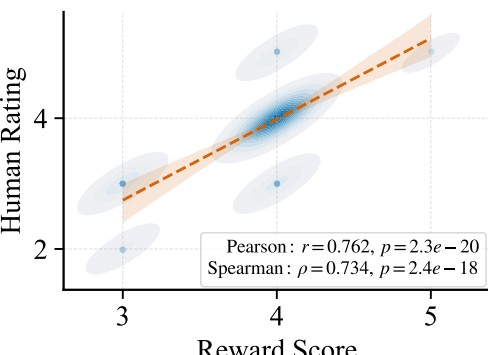

Figure 6: Performance of WebGen-R1 on the WebDev Arena benchmark with different domains and prompt distributions.

Figure 7: Alignment between reward model evaluations and human ratings on WebGen-Bench websites with strong correlations.

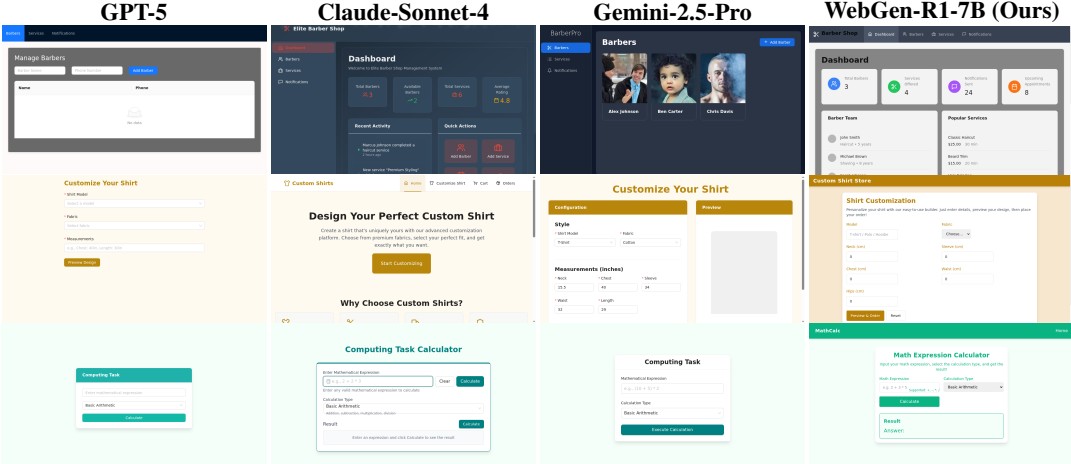

Figure 8: Case study of our WebGen-R1-7B against three leading baselines on in-distribution tasks from WebGen-Bench.

brings higher gains than RL alone, but combining SFT initialization with RL achieves even better results, with performance 21.60% and 28.76% higher than SFT and RL-only respectively. A similar pattern holds for the FSR and VRR metrics as well, as presented in Table 3. These findings indicate that SFT enables the model to acquire a robust structural and semantic prior for webpage generation, while RL expands the model's exploration capabilities and reward-driven optimization, allowing it to discover and produce higher-quality, more aesthetically pleasing websites.

**Group Size in GRPO.** We investigate the effect of the group size parameter $G$ in GRPO, with $G \in \{2, 4, 8, 16, 32\}$, while keeping all other hyperparameters fixed. For each setting, we track the evolution of both the mean and standard deviation of the reward per training step (see Appendix Figure 16), and assess the final policy on the WebGen-Bench using four metrics. As shown in Figure 5 and Table 4, larger group sizes consistently yield superior performance across the AAS, LDPR, FSR, and VRR metrics. We hypothesize that this improvement arises from the enhanced exploration capacity provided by larger groups, which increases the diversity of candidate trajectories and improves the likelihood of discovering high-quality website designs.

**Generalization and Robustness.** We evaluate the ability of WebGen-R1 to generalize its reasoning processes and visual design sensibility to settings where both the domain and prompt distributions differ substantially from those seen during RL, which was conducted on the WebGen-Instruct. To this end, we adopt the WebDev Arena benchmark, which features instruction distributions and task categories that are not covered in our training set. Detailed dataset statistics are provided in Table 5 in the Appendix. The quantitative results are presented in Figure 6. **Notably, we do not report FSR scores because WebDev Arena benchmark lacks standardized test cases, which pre-**

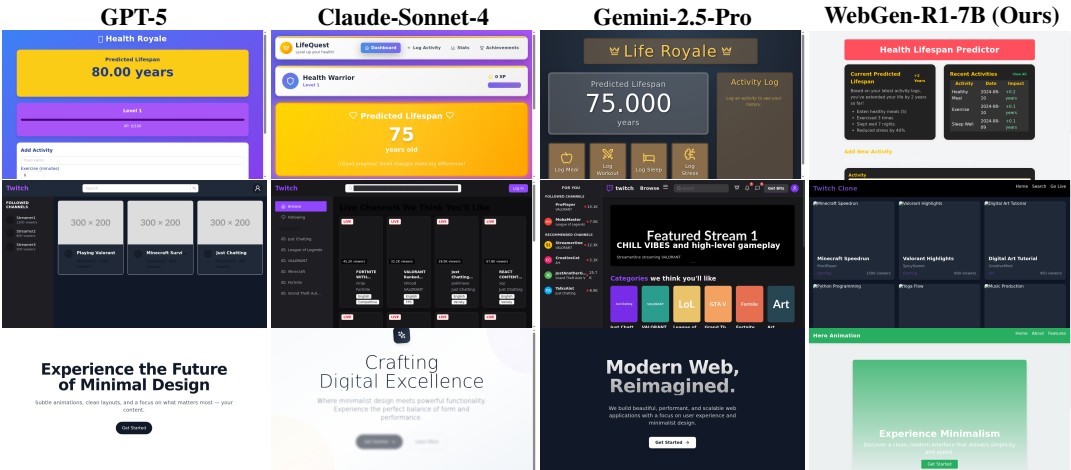

Figure 9: Case study of our WebGen-R1-7B against three leading baselines on out-of-distribution tasks from WebDev Arena.

**vents the GUI agent from performing functional website verification.** As illustrated in Figure 6, WebGen-R1 consistently outperforms a range of state-of-the-art proprietary and open-source baselines (*e.g.,* DeepSeek-R1, GPT-5, and Qwen3-32B) across AAS metric. This performance suggests that WebGen-R1 has learned architecture-level and style-level abstractions that remain effective in previously unseen web domains. These findings show that WebGen-R1 maintains strong applicability and robustness in real-world deployment scenarios where specifications change over time.

**Human Alignment Study.** We examine whether our vision–language-model-based reward accurately reflects human preferences for functionality and aesthetics. Since RL relies entirely on this reward, any misalignment could lead to outputs that do not meet user expectations. To evaluate this, we compare our reward model to human judgments by having three experienced front-end developers rate 101 websites from WebGen-Bench on functionality and visual appeal. We aggregate the human scores and compare them to the reward model's outputs. As shown in Figure 7, the results show strong correlations (Pearson $r = 0.762$, Spearman $\rho = 0.734$), indicating that the model's ratings closely match human evaluations. This demonstrates that our vision–language model can reliably assess both functional fidelity and aesthetic appeal in generated websites.

### 3.4 CASE STUDIES

To qualitatively evaluate WebGen-R1's ability to improve website functionality and visual quality, we conduct case studies using user instructions from in-distribution WebGen-Bench and out-of-distribution WebDev Arena sources, as shown in Figure 8 and 9, respectively. We observe that compared to strong baseline models, WebGen-R1 generates websites with organized layouts, coherent and attractive designs, and responsive behaviors that match the detailed instructions in WebGen-Bench. In contrast, WebDev Arena provides far less detailed instructions, leaving WebGen-R1 more dependent on its world knowledge and sometimes missing newer design trends. We plan to mitigate this using latest backbone models in future work. Overall, the results show that LLMs trained with perceptually grounded RL can effectively balance engineering requirements and design principles in web development. Additional case studies are provided in Appendix H.

## 4 CONCLUSION

In this work, we introduce WebGen-R1, a reinforcement learning framework that enables small-scale LLMs to generate the entire multi-page websites in an end-to-end manner while meeting both functional and aesthetic requirements. Extensive experiments across two real-world benchmarks demonstrate that our WebGen-R1 consistently improves functional robustness, visual coherence, and deployability, surpassing or matching to advanced proprietary and open-source baselines. Notably, our work demonstrates that RL with perceptually grounded rewards can substantially advance structured, multi-modal generation tasks beyond the functional level. We believe these insights open new directions for end-to-end LLM training in full-stack application development, and we release all resources to catalyze future research in this emerging area.

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

CONTENTS OF APPENDIX

---

**Algorithm 1:** WebGen-R1: GRPO for End-to-End Website Generation with a Single LLM

---

**Input:** Task distribution $\mathcal{D}_{\text{webgen}}$, initial policy $\pi_{\theta_0}$, frozen reference policy $\pi_{\text{ref}}$, multimodal reward evaluator $\phi_\psi^{\text{VLM}}$, group size $G$, clip parameter $\varepsilon$, learning rate $\eta$, KL coefficient $\beta$

1 **for** iter $= 1$ **to** $N_{\text{iter}}$ **do**
2     Sample prompt $x \sim \mathcal{D}_{\text{webgen}}$
3     Initialize reward list $\mathcal{R} \leftarrow [\ ]$ and $\theta_{\text{old}} \leftarrow \theta$
4     **for** $i = 1$ **to** $G$ **do**
        `// Generate candidate website sequence`
5         $y_i \sim \pi_{\theta_{\text{old}}}(\cdot \mid x)$
        `// Parse, install, build, launch, and render in the`
        `   sandbox environment`
6         $(\{I_{i,p}\}_{p=1}^{P}, \Gamma_i) \leftarrow \mathcal{E}(y_i) = \mathcal{R}(\mathcal{L}(\mathcal{B}(\mathcal{I}(\mathcal{W}eb(y_i)))))$
        `// Multimodal functionality and aesthetics evaluation`
7         $s_{\langle\text{func,vis}\rangle} \leftarrow \phi_\psi^{\text{VLM}}(x, \{I_{i,p}\}_{p=1}^{P}, \Gamma_i)$
8         $s_{\langle\text{code}\rangle} \leftarrow \text{CHECKCODEFORMAT}(y_i)$
9         $s_{\langle\text{cot}\rangle} \leftarrow \text{CHECKREASONINGFORMAT}(y_i)$
        `// Final fused reward (Eq. 4)`
10        $R_i \leftarrow s_{\langle\text{func,vis}\rangle} + \gamma\, s_{\langle\text{code}\rangle} + \lambda\, s_{\langle\text{cot}\rangle}$
11        Append $R_i$ to $\mathcal{R}$
    `// Compute group-relative normalized advantages`
12     $\bar{R} \leftarrow \text{mean}(\mathcal{R})$, $\sigma_R \leftarrow \text{std}(\mathcal{R})$
13     **for** $i = 1$ **to** $G$ **do**
14         **for** $t = 1$ **to** $|y_i|$ **do**
15             $\hat{A}_{i,t} \leftarrow (R_i - \bar{R})/\sigma_R$
16             $r_{i,t}(\theta) \leftarrow \frac{\pi_\theta(y_{i,t}|x,y_{i,<t})}{\pi_{\theta_{\text{old}}}(y_{i,t}|x,y_{i,<t})}$
17             $L_{i,t} \leftarrow \min\left(r_{i,t}(\theta) \cdot \hat{A}_{i,t},\ \text{clip}(r_{i,t}(\theta), 1-\varepsilon, 1+\varepsilon) \cdot \hat{A}_{i,t}\right)$
    `// Aggregate GRPO loss with KL regularization to reference`
    `   model`
18     $\mathcal{J}_{\text{GRPO}}(\theta) \leftarrow \frac{1}{G} \sum_{i=1}^{G} \frac{1}{|y_i|} \sum_{t=1}^{|y_i|} \left[ L_{i,t} - \beta\, D_{\text{KL}}\left(\pi_\theta(\cdot \mid x) \,\|\, \pi_{\text{ref}}(\cdot \mid x)\right) \right]$
    `// Gradient ascent step`
19     $\theta \leftarrow \theta + \eta\, \nabla_\theta \mathcal{J}_{\text{GRPO}}(\theta)$

---

## A   ALGORITHM FOR WEBGEN-R1

The full algorithmic procedure of our proposed WebGen-R1 is detailed in Algorithm 1.

## B   DATASET STATISTIC AND ANALYSIS

WebGen-Instruct and WebGen-Bench are based on web application categories distilled from real freelance and crowdsourcing project listings on platforms such as Upwork, Freelancer, and Proginn, which were further expanded through expert-generated specifications. This grounding in authentic industry project distributions ensures that the datasets better reflect practical web development scenarios.

As presented in Table 5, both WEBGEN-BENCH and WEBDEV ARENA constitute highly open-ended web generation benchmarks, yet they differ substantially in the distribution of instruction lengths and in the coverage of web development categories. WEBGEN-BENCH comprises 101 samples with moderately long natural language instructions (median 84 tokens, mean 86.06, max 135) and 647 executable test cases, and covers 13 heterogeneous front-end development scenarios ranging from static and dynamic rendering to AI integration and big data handling. This configuration indicates tasks with rich functional requirements and multi-modal constraints, compelling the model to interpret specifications that couple precise functional logic with explicit visual styling instructions.

Table 5: Statistics for the WEBGEN-BENCH and WEBDEV ARENA benchmarks including sample number, variations in instruction length (in tokens, measured by tiktoken's cl100k_base tokenizer), number of test cases, technical categories, and an example.

| Benchmark | Samples | Instruction Length | | | | # Test Cases | Category | Examples |
|---|---|---|---|---|---|---|---|---|
| | | # Min | # Median | # Max | # Avg. | | | |
| WebGen-Bench | 101 | 52 | 84 | 135 | 86.06 | 647 | Static Page Generation Dynamic Content Rendering Data Visualization Media Display Form Systems Authentication Real-time Features E-commerce AI Integration CRUD Operations API Integration Big Data File Handling | *Please develop a web-based Texas Hold'em poker game with features such as game lobby, table games, and chat functionality. Users should be able to create or join game rooms, play Texas Hold'em, view game records, and manage their account information. The game lobby should display available game rooms, current game status, and player information. The table game should display player hand cards, community cards, betting information, and action buttons. Implement azure for the page background and midnight blue for the elements.* |
| WebDev Arena | 119 | 3 | 20 | 119 | 23.13 | 0 | Website Design Game Development Clone Development App Development Web Development UI Design Digital Tools App Design AI Applications Simulations Creative Humor | *Make me a clone of WhatsApp Chat App.* |

Figure 10: Token length distributions of prompts and generated responses for several state-of-the-art commercial LLMs on the end-to-end website generation task on WebDev Arena. For each instance, the prompt is constructed by concatenating the system prompt with the corresponding natural-language web design instruction. The response length distributions are aggregated over the WebDev Arena, providing a quantitative characterization of input–output verbosity across models, which is relevant for assessing model efficiency and design complexity in realistic web generation scenarios.

In contrast, WEBDEV ARENA contains 119 carefully selected tasks characterized by substantially shorter instructions (median 20 tokens, mean 23.13, max 119), yet encompassing a wider thematic spectrum, including creative design, simulations, and game/app cloning. Unlike WEBGEN-BENCH, these tasks do not include predefined test cases, rendering evaluation criteria inherently more subjective and loosely specified. Consequently, models must make higher-level design decisions and infer multiple underspecified details, a defining aspect of open-ended generation tasks. This duality of explicit, testable functional requirements and underspecified creative objectives produces a non-trivial unified setting, in which attaining robust performance requires models to (i) process prompts of variable length and varying informational density, (ii) jointly address discrete source code generation and consistent aesthetic design, and (iii) adapt to fundamentally different reward structures. The

resulting open-endedness expands the effective policy search space and induces optimization landscapes with heterogeneous and non-uniform reward geometry, making the environment an effective stress test for generalization in RL-based web generation.

Figure 2 depicts the distributions of prompt and response lengths for several state-of-the-art LLMs in the end-to-end website generation task. Each prompt is formed by concatenating the system prompt with the task-specific query. Prompt lengths (see the left side of Figure 2) are concentrated between 3.2k and 3.3k tokens, whereas response length distributions (see the right side of Figure 2) vary considerably across models. Claude-sonnet-4 produces the longest responses, with most outputs around 12k tokens, followed by Gemini-2.5-pro at approximately 6k tokens and Qwen3-32 at roughly 4.5k. Interestingly, GPT-5 and our RL-trained WEBGEN-R1-7B exhibit similar distributions centered near 3.5k tokens, closely matching DeepSeek-R1. Qwen2.5-Coder-7B-Instruct generates shorter outputs clustered around 2k tokens. Notably, compared with Qwen2.5-Coder-7B-Instruct, which is the base model used for our RL training, our WEBGEN-R1-7B consistently produces longer outputs, indicating that RL optimization enhances reasoning in open-ended website generation.

## C  MORE IMPLEMENTATION DETAILS

**Baselines.** For the proprietary category, we evaluate via official API access the latest general-purpose models, including GPT-5 (OpenAI, 2025a), GPT-4.1 (Achiam et al., 2023), o3 (OpenAI, 2025b), o4-mini (OpenAI, 2025b), GPT-4o (Hurst et al., 2024), Claude-Sonnet-4 (Anthropic, 2025b), Claude-3.7-Sonnet (Anthropic, 2025a), and Gemini-2.5-Pro (Comanici et al., 2025). For open-source models, we include high-performing models with publicly available weights such as DeepSeek-R1 (Guo et al., 2025), Qwen2.5-Coder-7B-Instruct (Hui et al., 2024), Qwen2.5-72B-Instruct (Team, 2024), Qwen3-8B (Yang et al., 2025a), Qwen3-32B (Yang et al., 2025a), Qwen3-30B-A3B-Thinking-2507 (Yang et al., 2025a), and Qwen3-Coder-30B-A3B-Instruct (Yang et al., 2025a).

**Hyperparameter Settings.** The key hyperparameters are global batch size of 256, group size $G = 8$, clipping parameter $\epsilon = 0.2$, learning rate $\mathrm{lr} = 5 \times 10^{-6}$, KL-divergence coefficient $\beta = 0.01$, reward weighting factors $\gamma = 0.1, \lambda = 0.1$, and rollout number $n_{\mathrm{rollout}} = 16$. The maximum context length is set to 4,096 tokens for prompts and 8,192 tokens for model outputs. For text generation during rollouts, we adopt a decoding temperature of $0.7$ and a nucleus sampling $(\mathrm{top}_p)$ value of $0.95$, which we empirically find to balance exploration and output determinism.

**Website Generation and Execution Framework.** To ensure both functional correctness and visual design quality of LLM-generated websites, all generated website source code is executed within an isolated and secure sandbox environment. This controlled setting supports full compilation, execution, and rendering while preventing interference with external systems. Directly adapting or switching between heterogeneous sandbox environments can cause significant training latency and engineering complexity in reinforcement learning loops. To balance efficiency and reproducibility, we adopt a standardized web development framework inspired by prior works (Lu et al., 2025) and (LMArena, 2025), defining a predefined project bootstrap and technology stack. All generated web projects must initialize from the 'vite-react-typescript-starter' template [4], strictly preserving its directory structure, entry points, and configuration conventions. In cases where template defaults conflict with downstream requirements, necessary modifications or additional files must be introduced to ensure complete compliance. The core stack consists of React (function components with hooks where applicable), TypeScript with strict typing, Vite as both build and development tool, and Tailwind CSS for styling. For complex or reusable user interface elements, the Ant Design ('antd') library is mandated, providing consistent styling and interaction patterns. Any usage of 'shadcn/ui', 'shadcn-ui', or similar variants is explicitly prohibited to avoid style inconsistencies. Routing functionality is implemented through React Router DOM v6. In scenarios where visualizations such as charts or graphs are explicitly requested, Recharts is the only permitted charting library, ensuring predictable rendering behaviors and compatibility across environments. This unified framework eliminates variability during rendering and interaction phases, thereby enabling stable functional and aesthetic evaluation in RL training loops.

---

[4]`https://github.com/vitejs/vite/tree/main/packages/create-vite/template-react-ts`

**Supervised Fine-Tuning (SFT).** The training pipeline begins with a supervised fine-tuning stage to provide a strong initialization for subsequent RL optimization. From the WebGen-Instruct dataset, we sample 600 instances based on the 'application_type' data field to preserve the original domain distribution. This sampling ensures that the model receives an equal representation of application scenarios, mitigating the risk of distribution shift and maintaining generalization across diverse web development tasks. For each sampled task-specific query, website generation data is distilled from the advanced model 'GPT-4.1-2025-04-14' under controlled inference parameters, with the temperature set to 0.6 to balance creativity and determinism, top_p set to 0.95 for controlled sampling diversity, and a maximum token limit of 8,192 to ensure complete project generation within a single inference step. We then fine-tune the Qwen2.5-Coder-7B-Instruct model using the Open-R1 training framework [5]. Training hyperparameters include a learning rate of $1.0 \times 10^{-5}$, batch size of 32, maximum sequence length of 32k tokens, warmup ratio of 0.03, and 2 training epochs.

**UI Functional Evaluation with WebVoyager.** Following (Lu et al., 2025), we employ WebVoyager (He et al., 2024a) [6], a large multimodal model powered web agent capable of executing user instructions end-to-end by interacting directly with rendered websites, and we use 'GPT-4o-2024-11-20' as the agent engine. WebVoyager performs pre-defined interactive behaviors such as button clicks, form submissions, and multi-page navigation, while observing DOM changes and UI responses. The Functional Success Rate (FSR) is computed as the proportion of tasks that successfully pass all behavior checks defined in WebGen-Bench benchmark. This automated interaction testing provides a scalable and reproducible evaluation of functional correctness of generated websites without requiring human annotators.

**Vision–Language–Model-based Reward Model.** Inspired by prior work (Lu et al., 2025) demonstrating the reliability and efficiency of GPT-4o for webpage design aesthetic assessment, we incorporate GPT-4o-1120 as the vision–language–model-based reward model. During RL training, the rendered multiple page images, together with the original user request, is passed to GPT-4o-1120 for joint functional and aesthetic evaluation. The model produces a scalar reward within the discrete range $[0, 5]$, where higher scores denote better overall compliance with the requested functionality and visual design quality. This reward model enables simultaneous optimization toward both functional correctness and aesthetic appeal, ensuring that generated websites are not only operational but also meet professional design standards. The joint optimization objective is critical for real-world deployment scenarios in which usability and appearance are equally important, thereby aligning LLM outputs more closely with user-centric quality expectations.

**Prompt Design.** We present the full system prompt used for website generation in Prompt G.1, the reward evaluation prompts for assessing functionality and visual aesthetics in Prompt G.2, and the prompt employed for WebDev Arena data selection in Prompt G.3.

# D  RELATED WORK

## D.1  LARGE LANGUAGE MODELS FOR PROJECT-LEVEL CODE GENERATION

Large language models (LLMs) have demonstrated remarkable proficiency in functional-level code generation, achieving near-human performance in competitive programming and standard benchmarks such as HumanEval (Chen et al., 2021), MBPP (Austin et al., 2021), and IOI problems (Li et al., 2022). Advances in instruction tuning (Ouyang et al., 2022; Wang et al., 2023; Chung et al., 2024) and tool-augmented prompting have further enhanced zero-shot and few-shot code generation capabilities. However, these achievements predominantly concern single-file or self-contained scripts, often constrained to producing one function or module per task. Such settings abstract away complexities of real-world software engineering, where projects span multiple files, require intricate inter-module dependencies, and must adhere to both functional and non-functional requirements. Compared to functional-level coding, project-level generation poses qualitatively different challenges. Prior attempts to extend LLMs to multi-file outputs include hierarchical prompting (Shrivastava et al., 2023; Zhang et al., 2023), iterative refinement (Chen et al., 2023; Olausson et al., 2023; Shinn et al., 2024), and agent-based pipelines (Li et al., 2024; Zhang et al., 2024; Luo et al., 2025a). In the specific context of web development, benchmarks such as WebDev Arena (LMArena,

---

[5] https://github.com/huggingface/open-r1
[6] https://github.com/MinorJerry/WebVoyager

2025) provide automated evaluation, but their scope is typically limited to single-page static sites, neglecting the broader demands of large-scale, interactive, multi-page applications. Multi-agent approaches partition functionality across specialized LLMs, such as front-end generation, API design, and testing. However, integration often suffers from inconsistent shared states and fragile inter-component linking. These shortcomings may result in generated projects that compile but fail to align with the holistic end-to-end specifications of real-world sites.

## D.2 REINFORCEMENT LEARNING FOR CODE GENERATION

Reinforcement Learning (RL) has emerged as a crucial technique to align LLM behavior with human preferences and task-specific objectives, as exemplified by RLHF (Ouyang et al., 2022; Achiam et al., 2023) and RLAIF (Bai et al., 2022; Lee et al., 2023). Yet, applying RL to open-ended code generation introduces unique obstacles such as the vast search space, undefined or ambiguous ground truths, and outputs that cannot be trivially benchmarked against static gold standards. Reinforcement Learning with Verifiable Rewards (RLVR) (Shao et al., 2024; Guo et al., 2025; Team et al., 2025; Yu et al., 2025) addresses part of this issue via deterministic, binary success checks (*e.g.,* unit test pass rates), which work well for algorithmic correctness but fail to capture subjective quality dimensions such as style, maintainability, or visual experience. Existing RL applications in code (Le et al., 2022; Shen et al., 2023; Shojaee et al., 2023; Dou et al., 2024; Luo et al., 2025b) thus typically optimize for purely functional scores, leaving large gaps for domains like website generation where aesthetics and interaction design are first-class objectives. While prior research has advanced LLM-based code generation, none simultaneously addresses multi-file structural coherence, execution validity, and visual quality within an integrated RL framework.

# E ADDITIONAL EXPERIMENTAL RESULTS

## E.1 WEB CODE FORMAT AND REASONING FORMAT REWARDS

We investigate the role of two complementary reward signals in our multi-objective RL setup. The first is the web code format reward $s_{\langle \text{code} \rangle}$, which promotes the generation of syntactically correct, structurally coherent, and executable web code. The second is the reasoning format reward $s_{\langle \text{cot} \rangle}$, designed to encourage logically consistent, step-by-step reasoning traces that guide the code generation process. The relative contributions of these rewards are modulated by weighting factors $\gamma$ for web code format and $\lambda$ for reasoning format. To systematically examine their interaction, we evaluate ten representative $(\gamma, \lambda)$ configurations, with each weight taking values in $\{0, 0.1, 0.5, 1.0\}$. These settings span low–low, low–high, high–low, and high–high regions of the reward space. All models are trained under identical conditions and assessed using our

Table 6: Sensitivity analysis of $(\gamma, \lambda)$ on model performance across FSR, AAS, and VRR metrics.

| Hyperparameters | FSR (%) | AAS | VRR (%) |
|---|---|---|---|
| $\gamma = 0.0, \lambda = 0.0$ | 25.89 | 3.70 | 94.01 |
| $\gamma = 0.0, \lambda = 0.1$ | 28.54 | 3.92 | 95.00 |
| $\gamma = 0.0, \lambda = 1.0$ | 26.13 | 3.87 | 95.00 |
| $\gamma = 0.1, \lambda = 0.0$ | 24.08 | 3.66 | 93.02 |
| $\gamma = 0.1, \lambda = 0.1$ | **29.21** | **3.94** | **95.89** |
| $\gamma = 0.1, \lambda = 1.0$ | 28.39 | 3.92 | 94.01 |
| $\gamma = 1.0, \lambda = 0.0$ | 22.75 | 3.34 | 91.04 |
| $\gamma = 1.0, \lambda = 0.1$ | 23.06 | 3.47 | 91.09 |
| $\gamma = 0.5, \lambda = 0.5$ | 26.12 | 3.77 | 93.17 |
| $\gamma = 1.0, \lambda = 1.0$ | 25.94 | 3.78 | 93.02 |

proposed FSR, AAS, and VRR metrics. As presented in Table 6, the results show that smaller $\gamma$ values generally lead to higher AAS and VRR, often accompanied by increased FSR. In contrast, large $\gamma$ values tend to reduce FSR when $\lambda$ is small, with a slight recovery as $\lambda$ increases. Mid-range $\lambda$ values provide moderate gains in AAS and VRR across settings. These trends indicate a trade-off between functional correctness, aesthetic quality, and reasoning consistency, suggesting that a balanced combination of reward weights is important for optimal generation quality.

## E.2 HUMAN ALIGNMENT STUDY ON WEBDEV ARENA

Consistent with the analysis presented in Figure 7, we extend our evaluation to the WebDev Arena benchmark, which contains instruction distributions and task categories that are not included in WebGen-Instruct (training data) and WebGen-Bench (test data).

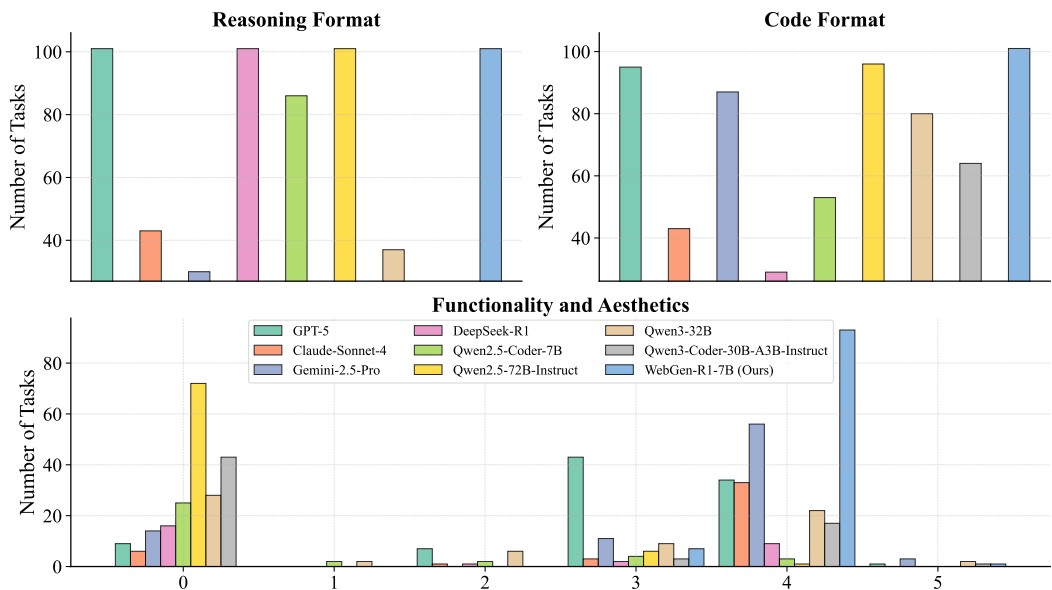

Figure 12: Website quality score distributions for multiple advanced proprietary and open-source models on WebGen-Bench. WebGen-R1 demonstrates perfect reasoning and code format adherence, consistently high functionality and aesthetics scores, and robust generation reliability, validating the effectiveness of reinforcement learning with explicit format and quality-oriented rewards.

We sample 119 website instances from this benchmark, as described in Section C. Each instance is independently evaluated by three experienced front-end practitioners, who assign quantitative scores for functionality and visual aesthetics on a discrete scale from 0 to 5 following standardized evaluation protocols. The Human scores are aggregated across annotators and then compared with the outputs of our reward model. As shown in Figure 11, there is a strong and statistically significant alignment between reward scores and human judgments, with Pearson's correlation coefficient $r = 0.903$ ($p = 7.8 \times 10^{-45}$) and Spearman's rank correlation $\rho = 0.888$ ($p = 2.2 \times 10^{-41}$). The close agreement between these correlation measures indicates that the vision–language reward model's assessments are highly monotonic with respect to human ratings and approximately linear in scale. Moreover, the high consistency between the results on WebDev Arena as an out-of-distribution evaluation set and those on

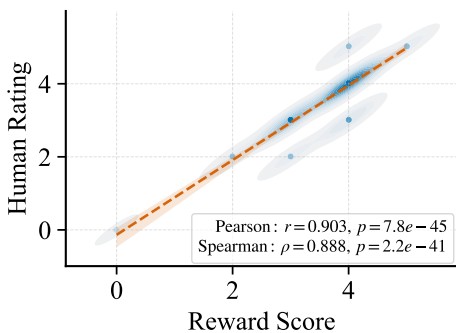

Figure 11: Alignment between reward model evaluations and human ratings on WebGen-Bench websites, with strong correlations (Pearson $r = 0.903$, Spearman $\rho = 0.888$) indicating reliable automatic assessment of functional correctness and aesthetic quality.

WebGen-Bench as an in-distribution benchmark provides strong empirical evidence that the reward model used in our reinforcement learning process is robust and exhibits minimal risk of reward hacking.

### E.3 WEBSITE QUALITY SCORE DISTRIBUTION AND ANALYSIS

To obtain an in-depth understanding of model capabilities in the end-to-end multi-page website generation task, we conduct a detailed evaluation across both reasoning format adherence and code format compliance, as well as the distribution of functionality and aesthetics scores on two benchmarks, WebGen-Bench and WebDev-Arena. This analysis is performed on a diverse set of advanced proprietary and open-source models, including our proposed WebGen-R1, with the goal of providing deeper insight into their generation reliability and overall website quality. In terms of reasoning

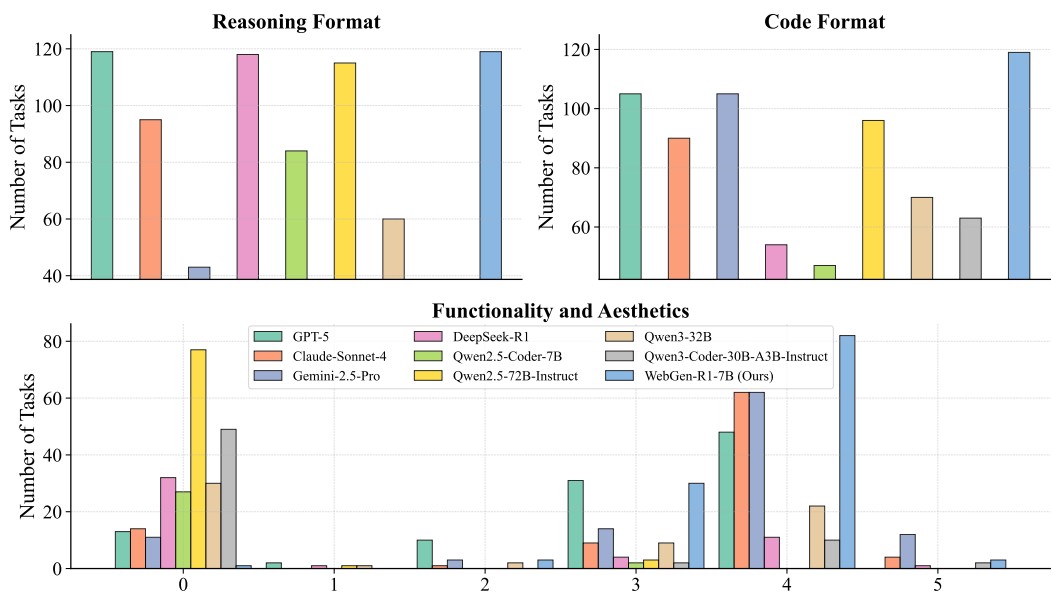

Figure 13: Website quality score distributions for multiple advanced proprietary and open-source models on WebDev Arena. WebGen-R1 demonstrates perfect reasoning and code format adherence, consistently high functionality and aesthetics scores, and robust generation reliability, validating the effectiveness of reinforcement learning with explicit format and quality-oriented rewards.

format adherence, results in Figures 12 and 13 show that GPT-5, DeepSeek-R1, and our WebGen-R1 achieve perfect compliance on both benchmarks, consistently producing outputs that fully conform to the predefined <think >...</think><answer>...</answer> structure across all tasks. This indicates a high degree of controllability and reliability in structured reasoning output. Claude-Sonnet-4, on the other hand, demonstrates reduced adherence due primarily to the omission of closing tags, such as missing </answer>, which was confirmed through careful manual inspection. Interestingly, Qwen3-32B reliably produces the first <think >...</think> block but occasionally omits the second <answer>...</answer> block, instead completing the answer directly. Similarly, Qwen3-Coder-30B-A3B-Instruct supports only non-thinking mode and does not generate <think >...</think> blocks in its output, as documented in its model card [7]. For code format compliance, WebGen-R1 again achieves a perfect 100% success rate on both benchmarks. This outcome can be directly attributed to the explicit code format reward incorporated into our reinforcement learning objective, which consistently biases the model toward generating code in the exact target structure. GPT-5, Gemini-2.5-Pro, and Qwen2.5-72B-Instruct also exhibit strong adherence in this aspect, indicating that zero-shot prompting can produce highly reliable code formatting behavior owing to their powerful instruction-following capabilities. When examining the distribution of functionality and aesthetics scores, we observe that our WebGen-R1 attains the highest number of tasks receiving a score of 4 across both benchmarks. This suggests that WebGen-R1 has acquired transferable abstractions at both architectural and stylistic levels, enabling it to produce functional and visually appealing websites even in domains not encountered during training. Furthermore, in the distribution of maximum scores, tasks scoring 5 are most frequently achieved by Gemini-2.5-Pro, highlighting its superior ability to simultaneously satisfy functional requirements and achieve high aesthetic quality under the given task constraints. Overall, this multi-faceted analysis reveals that our WebGen-R1 exhibits both strict structural compliance and consistently high-quality generation, while also shedding light on subtle deviations and nuances in other competitive models. These insights confirm that reinforcement learning with explicit format and quality-oriented rewards can substantially improve reliability and performance in complex website generation tasks.

[7] https://huggingface.co/Qwen/Qwen3-Coder-30B-A3B-Instruct

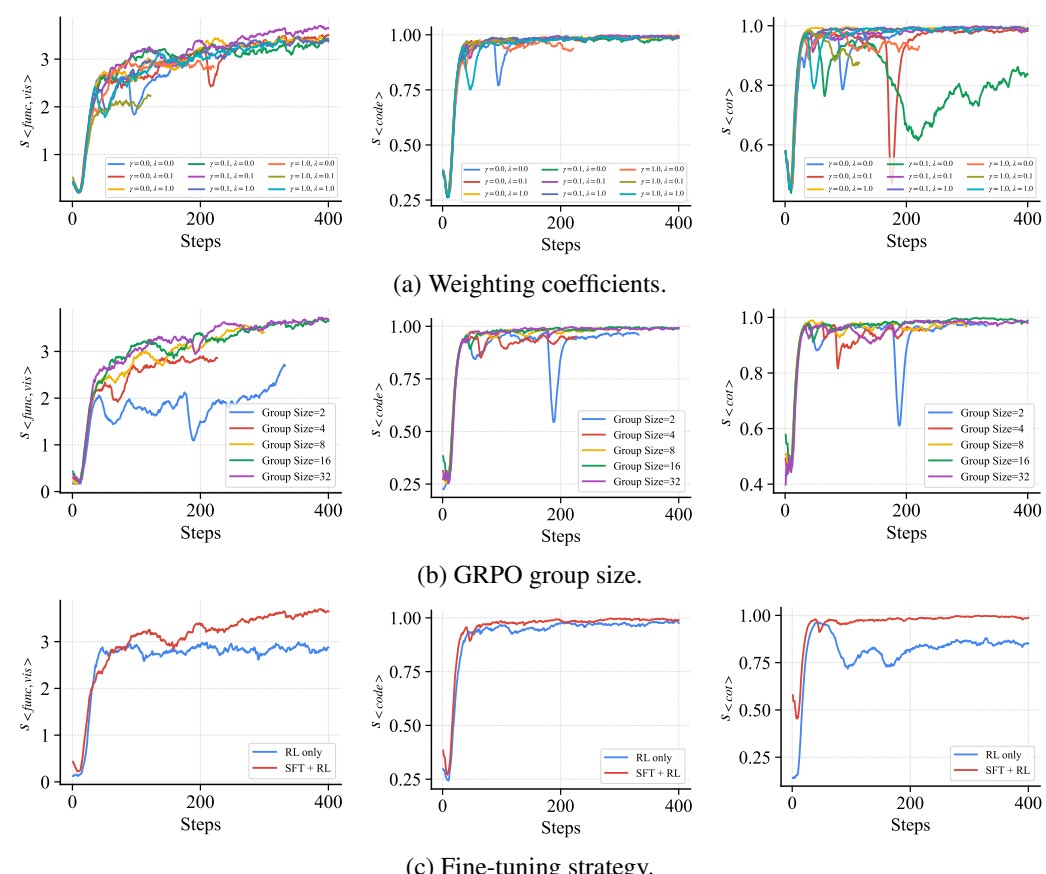

(a) Weighting coefficients.

(b) GRPO group size.

(c) Fine-tuning strategy.

Figure 14: Reward trajectories for functional-visual quality $s_{\langle\text{func,vis}\rangle}$, code formatting $s_{\langle\text{code}\rangle}$, and reasoning format $s_{\langle\text{cot}\rangle}$ under different weighting coefficients, GRPO group sizes, and fine-tuning strategies, showing that multi-objective stability benefits from hybrid SFT+RL and careful reward component design.

### E.4 REWARD DYNAMICS AND OPTIMIZATION STABILITY

To better understand how reinforcement learning fine-tuning affects the balance between functional correctness and visual appeal in website generation, we analyze the evolution of three reward components during RL training, providing insights into the interaction between multiple objectives and highlighting factors that influence the stability of RL optimization. We track the three distinct reward components we design during training, namely $s_{\langle\text{func,vis}\rangle}$ for functional correctness and visual design quality, $s_{\langle\text{code}\rangle}$ for code formatting compliance, and $s_{\langle\text{cot}\rangle}$ for correctness in reasoning format. In addition, we examine how the three reward components are affected under different settings, including the weighting coefficients associated with $s_{\langle\text{code}\rangle}$ and $s_{\langle\text{cot}\rangle}$, the group size parameter in GRPO's rollout sampling, and different fine-tuning strategies including RL-only and SFT+RL. The corresponding trends are illustrated in Figure 14 (a), (b), and (c). Our observations can be summarized as follows:

- When the coefficient $\gamma$ controlling $s_{\langle\text{code}\rangle}$ is set to zero, the overall code formatting remains largely unaffected. This suggests that rewards targeting functional and visual aspects implicitly promote code formatting consistency, possibly because improvements in semantics and structure encourage properly formatted output. In contrast, setting $\lambda = 0$ for $s_{\langle\text{cot}\rangle}$ leads to instability in reasoning format compliance, with a pronounced collapse in the later stages of training followed by partial recovery, indicating that omitting format-specific rewards for reasoning can disrupt optimization stability.

- Increasing the group size in GRPO generally results in higher mean values for all three rewards and smoother convergence profiles. For example, with group size $= 2$, we observe volatile fluctuations and temporary collapses around step 200. Larger group sizes, such as

32, improve stability for $s_{\langle\text{func,vis}\rangle}$ and $s_{\langle\text{code}\rangle}$, yet the improvement for $s_{\langle\text{cot}\rangle}$ becomes less pronounced. This suggests diminished benefits due to over-expansion of the exploration space, where low-quality samples might become more prevalent.

- The SFT+RL strategy offers clear advantages over RL-only. For $s_{\langle\text{func,vis}\rangle}$, RL-only begins to converge around step 60, whereas SFT+RL continues to improve steadily beyond that point, reaching higher final values. For $s_{\langle\text{code}\rangle}$ and $s_{\langle\text{cot}\rangle}$, the SFT+RL setting demonstrates greater stability, while RL-only exhibits late-stage oscillations and declines, particularly in $s_{\langle\text{cot}\rangle}$. These results indicate that initialization via supervised fine-tuning equips the model with robust formatting adherence, while subsequent RL optimization expands exploration and leverages reward signals to discover and generate higher-quality, aesthetically refined websites.

Overall, this analysis underscores that fine-grained reward design, appropriate group sizing, and hybrid SFT+RL training are crucial for stabilizing multi-objective optimization in LLM-based website generation. The observed dynamics highlight that certain objectives are implicitly reinforced by others, whereas others require explicit reward shaping to prevent collapse, offering valuable guidance for future multi-objective RL frameworks.

### E.5 Reasoning Behavior in Website Generation

We conduct a qualitative comparison of reasoning traces and find that, under identical task instructions sampled from WebGen-Bench benchmark, our WebGen-R1 produces a more complete and implementation-ready plan for the credit repair lead-generation website, as shown in Figure 17. In the context of front-end development, the advantages are clear. WebGen-R1 explicitly maps user requirements to a structured routing scheme and coherent page-component hierarchy, ensuring maintainability and scalability. It integrates styling decisions directly into the architectural plan, specifying how Tailwind CSS overrides and Ant Design components achieve consistent visual themes and responsiveness. Accessibility is embedded at the reasoning stage through ARIA attributes and keyboard navigation, a consideration absent in the baseline model. State management is also more concrete, with precise handling of client-side data storage and strict TypeScript typing for form models, which increases robustness in generated code. These elements, specifically clear route planning, style-system integration, accessibility, and state handling, are central to modern front-end engineering, and the reasoning path of WebGen-R1 demonstrates stronger alignment with production-grade development practices than the baseline.

Table 7: Ablation study on the effect of reasoning traces.

| Reasoning | FSR (%) | AAS | VRR (%) |
|---|---|---|---|
| w/o CoT | 25.76 | 3.75 | 93.12 |
| w CoT | 29.21 | 3.94 | 95.89 |

To evaluate the contribution of reasoning traces in a controlled setting, we perform an ablation study that isolates their effect on both reward quality and final task performance. The experimental design removes the reasoning traces from the generation process and compares the resulting outputs to those produced with full reasoning behavior. Table 7 reports the results in terms of FSR, AAS, and VRR metrics. Across all metrics, the inclusion of reasoning traces yields consistent improvements with FSR increasing from 25.76 to 29.21, AAS from 3.75 to 3.94, and VRR from 93.12 to 95.89. As shown in Figure 15, the reward curves in training further confirm this trend, remaining consistently higher when reasoning traces are present, which indicates improved reward quality and stability. These results provide quantitative evidence that explicit reasoning benefits

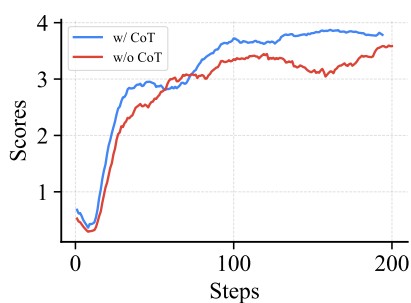

Figure 15: Training reward curves comparing models with and without reasoning traces.

both functional and aesthetic aspects of website generation, reinforcing the qualitative observations described earlier.

### E.6 EVALUATION OF VLM-BASED REWARD MODELS

To examine the potential impact of alternative VLM-based reward model, we conducted a comparative study between three representative VLMs: the open-source Qwen2.5-VL-72B-Instruct deployed on a rented cluster with $8\times$NVIDIA HGX H100 SXM (USD 23.92/h), the proprietary GPT-4.1 accessible via API, and the proprietary GPT-4o. Results are summarized in Table 8.

Table 8: Performance comparison of alternative VLM-based reward models.

| VLM Evaluator | FSR (%) | AAS | VRR (%) | Speed (s/sample) | Prompt Tokens | Completion Tokens | Cost (USD/sample) |
|---|---|---|---|---|---|---|---|
| Qwen2.5-VL-72B-Instruct | 10.32 | 3.18 | 27.72 | 6.47 | 1,344 | 332.8 | 0.036 |
| GPT-4.1 | 24.58 | 3.36 | 91.04 | 11.57 | 1,190 | 389 | 0.005 |
| GPT-4o | **29.21** | **3.94** | **95.89** | 12.01 | 1190 | 437 | 0.007 |

The results reveal that GPT-4o consistently achieves the highest scores across FSR, AAS, and VRR, indicating stronger alignment between reward evaluations and high-quality website generation outputs. Although GPT-4.1 performs competitively, it does not surpass GPT-4o in any metric, suggesting that reward model choice affects not only absolute performance but also downstream training stability. The open-source Qwen2.5-VL-72B-Instruct exhibits notably lower functional and visual reliability despite faster inference, and its operational cost is amplified by the high-end hardware requirements. Overall, these findings highlight the importance of selecting a reward evaluator that balances accuracy and cost efficiency, with GPT-4o emerging as the most effective within our experimental settings.

We further conducted an additional evaluation to investigate the reliability of these VLMs as judges for website generation using the PairBench framework (Feizi et al., 2025). One open-source model (Qwen2.5-VL-72B-Instruct) and two proprietary models (GPT-4.1 and GPT-4o) were evaluated under our adapted text-to-website setting. The `wu_img_text` template was modified by adjusting `query templates` (see Appendix G.4), `query conditions` (see Appendix G.5), and logistics to accommodate the characteristics of our task. From WebGen-Bench, twenty tasks were randomly sampled and paraphrased using GPT-5 (see Appendix G.6), with one unrelated task added as a distractor. Given the absence of ground truth in this open-ended setting, only $\varepsilon$-RelaxSym and Smoothness metrics were calculated. Table 9 summarizes the results.

The results show that Qwen2.5-VL-72B-Instruct achieves the highest $\varepsilon$-RelaxSym, indicating strong order invariance, but its Smoothness is the lowest, suggesting limited granularity in scoring differences. GPT-4.1 presents both high $\varepsilon$-RelaxSym and the highest Smoothness, reflecting robust invariance combined with fine-grained discrimination. GPT-4o matches GPT-4.1 in Smoothness but has lower $\varepsilon$-RelaxSym, implying greater sensitivity to input order while maintaining nuanced scoring capability. In the context of text-to-website evaluation, Smoothness is particularly important as it enables better differentiation among diverse outputs. Considering this criterion, GPT-4o remains a suitable judge for our setting, reinforcing its effectiveness as a reward model in website generation.

### E.7 COMPARISON OF VLM-BASED REWARD AND GUI-AGENT EVALUATION

To investigate scalable reward mechanisms for training WebGen-R1, we conducted a comparative study between a VLM-based reward model and GUI-agent-based evaluation for functional correctness, as shown in Table 10. The VLM-based approach infers functionality from static web page screenshots by detecting visual indicators such as input forms, drop-down menus, and search fields, while GUI-agent evaluation actively interacts with user interface elements within a browser environment to assess true interactivity.

Table 9: PairBench evaluation of VLM-based judges in website generation. Higher $\varepsilon$-RelaxSym indicates lower sensitivity to input order. Higher Smoothness indicates finer-grained scoring differentiation.

| Model | $\varepsilon$-RelaxSym (1-RS) (%) | Smoothness |
|---|---|---|
| Qwen2.5-VL-72B-Instruct | 95.00 | 2.28 |
| GPT-4.1 | 90.00 | **2.88** |
| GPT-4o | 79.17 | 2.80 |

Table 10: Comparison of GUI-agent evaluation and VLM-based reward for functional correctness.

| Aspect | GUI-agent Evaluation | VLM-based Reward |
|---|---|---|
| Functional correctness | Interact with UI elements | Infer functionality from static screenshots using visual cues |
| **GUI Interaction Instructions** | **Hard to obtain and expensive** (Not available in WebGen-Instruct) | **Not applicable** |
| Interaction requirement | Requires full browser, DOM access, multi-step actions | No interaction needed |
| **Reward cost** | **Very high ($\sim 51\times$ slower; avg. 51.2 multi-turns; 614.91s/sample)** | **Very low (single-turn; 12.01s/sample)** |
| Environment stability | Fragile (affected by page load, network, DOM changes) | Fully stable (deterministic, self-contained) |
| Scalability for RL training | Poor (prohibitively expensive for millions of rollouts) | Excellent (scalable to large datasets) |
| Coverage of UI properties | Strong for interaction correctness | Strong for layout, visual structure, and element presence |
| Best use cases | Final evaluation, small-scale precise testing | RL training stage, large-scale sample collection |
| Main limitations | Slow, costly, unstable | Cannot fully verify true interactive functionality |

In preliminary experiments, GUI-agent-based evaluation achieved strong coverage of interactive correctness but suffered from practical limitations in large-scale reinforcement learning (RL) pipelines. The process requires full browser and DOM access, multi-step actions, and extensive environment management. Our measurements show that this method is approximately $51\times$ slower than the VLM-based approach, with an average of 51.2 multi-turn interactions and 614.91s per sample compared to 12.01s per sample for single-turn VLM evaluation. Additionally, GUI-agent evaluation is more susceptible to instability caused by page load variability, network issues, and DOM changes, making it less suited for millions of rollouts. In contrast, the VLM-based reward model is fully deterministic, cost-effective, and capable of high-throughput training, although it cannot fully verify backend logic or dynamic behaviors.

These results indicate that while GUI-agent evaluation provides richer interaction-based correctness signals, its cost, instability, and slow execution make it not well-suited for large-scale reinforcement learning. VLM-based rewards, despite limited coverage of backend logic, offer a tractable, stable, and highly scalable solution for the RL training stage, enabling millions of rollouts with minimal overhead.

### E.8 SCALABILITY ANALYSIS WITH LARGER MODELS

To evaluate the scalability of WebGen-R1 under reinforcement learning optimization, we extend our exploration to larger model variants built upon Qwen2.5-Coder-14B-Instruct and Qwen2.5-Coder-32B-Instruct, in addition to the 7B baseline. The study measures both computational and API costs per training iteration with batch size set to 64 and rollout number fixed at 16. This setting enables a direct comparison of efficiency and performance trends across different model parameter scales.

Table 11: Scalability evaluation of WebGen-R1 models with varying model parameter sizes. Per-iteration compute cost is measured with batch size 64 and rollout number 16.

| Model | Params | GPUs | Per-Iter. Compute Cost | Per-Iter. API Cost | FSR | AAS | VRR |
|---|---|---|---|---|---|---|---|
| WebGen-R1-7B | 7B | 8×H100(80G) | 354.2s | $7.168 | 29.21 | 3.94 | 95.89 |
| WebGen-R1-14B | 14B | 16×H100(80G) | 393.6s | $7.168 | 32.16 | 4.01 | 96.02 |
| WebGen-R1-32B | 32B | 32×H100(80G) | 475.7s | $7.168 | 37.08 | 4.09 | 97.13 |

The per-iteration compute cost corresponds to approximately 5.90 GPU·min for the 7B model, 6.56 GPU·min for the 14B model, and 7.93 GPU·min for the 32B model. Results in Table 11 indicate that WebGen-R1 scales smoothly to larger architectures while yielding consistent improvements in

all evaluation metrics. The FSR, AAS, and VRR each exhibit a monotonic increase with model size, suggesting that the RL procedure effectively leverages additional model capacity to improve both functional and aesthetic aspects of generated websites.

### E.9 Fine-tuning Strategies for Website Generation

We investigate the impact of different fine-tuning strategies on the model performance for functional and aesthetic website generation. Two approaches are considered. The first, WebGen-R1-7B (SFT), starts from Qwen2.5-Coder-7B-Instruct and is supervised fine-tuned on 600 high-quality websites generated by GPT-4.1. The second, WebGen-R1-7B (HPRM), follows the standard reward model training pipeline (Ouyang et al., 2022) to train a website-specific reward model using Qwen3-VL-8B-Instruct (Yang et al., 2025a) on the same 600 tasks. To reduce annotation cost, preference pairs are automatically constructed by selecting GPT-4.1 outputs as positives and Qwen3-8B outputs as negatives. The reward model is fine-tuned with LLaMA-Factory (Zheng et al., 2024) and subsequently employed as the evaluator in reinforcement learning of WebGen-R1. The results are reported in Table 12.

The results show that supervised fine-tuning (SFT) leads to substantial improvements over the baseline in both functional and aesthetic metrics. In contrast, reward-model-based fine-tuning underperforms compared with SFT despite achieving moderately higher VRR. Analysis of reward curves reveals an initial increase followed by a rapid decrease during training, indicating instability and eventual model collapse. This can be attributed

Table 12: Performance of different fine-tuning strategies for website generation.

| Model | FSR (%) | AAS | VRR (%) |
|---|---|---|---|
| Baseline | 1.59 | 2.73 | 30.56 |
| WebGen-R1-7B (SFT) | 20.08 | 3.24 | 30.69 |
| WebGen-R1-7B (HPRM) | 8.64 | 2.95 | 48.23 |
| WebGen-R1-7B (Ours) | 29.21 | 3.94 | 95.89 |

to the limited size of the preference dataset, constrained by cost and time, as well as insufficient diversity in reward model training tasks, which reduces its ability to robustly evaluate complex or out-of-distribution website generation scenarios. In comparison, large general-purpose LLMs exhibit strong instruction-following and generalization capabilities, which align more consistently with our evaluation criteria and provide more stable reward signals in open-ended generation settings.

## F Theoretical Analysis of GRPO with Increasing Group Size

We provide a theoretical justification of why, in Group Relative Policy Optimization (GRPO) (Shao et al., 2024; Guo et al., 2025), increasing the *group size* $G$ yields policy gradient estimates that asymptotically converge to the analytic optimal-baseline gradient, and why this convergence leads to improved reinforcement learning optimization performance.

We start by restating the GRPO setting more formally. For a given question-answer pair $(q, a)$ from dataset $\mathcal{D}$, the behavior policy $\pi_{\theta_{\text{old}}}$ samples a *group* of $G$ responses $\{o_i\}_{i=1}^{G}$, with each response $o_i = (o_{i,1}, o_{i,2}, \ldots, o_{i,|o_i|})$. For each token position $t$ within $o_i$, the group-relative normalized advantage is defined as:

$$\hat{A}_{i,t} \triangleq \frac{R_i - \text{mean}(\{R_j\}_{j=1}^{G})}{\text{std}(\{R_j\}_{j=1}^{G})}, \tag{7}$$

where $R_i$ is the (scalar) reward obtained by the $i$-th response. This normalization eliminates the need for a separately learned value function and ensures that within the group, credit assignment is relative to other members.

Following the PPO-style clipping strategy with an explicit KL penalty, the GRPO objective can be expressed as:

$$\mathcal{J}_{\text{GRPO}}(\theta) = \mathbb{E}_{(q,a)\sim\mathcal{D}, \{o_i\}_{i=1}^{G}\sim\pi_{\theta_{\text{old}}}(\cdot|q)}$$

$$\left[ \frac{1}{G} \sum_{i=1}^{G} \frac{1}{|o_i|} \sum_{t=1}^{|o_i|} \left( \min\left(r_{i,t}(\theta)\, \hat{A}_{i,t},\ \text{clip}(r_{i,t}(\theta), 1-\varepsilon, 1+\varepsilon)\, \hat{A}_{i,t}\right) - \beta\, D_{\text{KL}}(\pi_\theta \,\|\, \pi_{\text{ref}}) \right) \right], \tag{8}$$

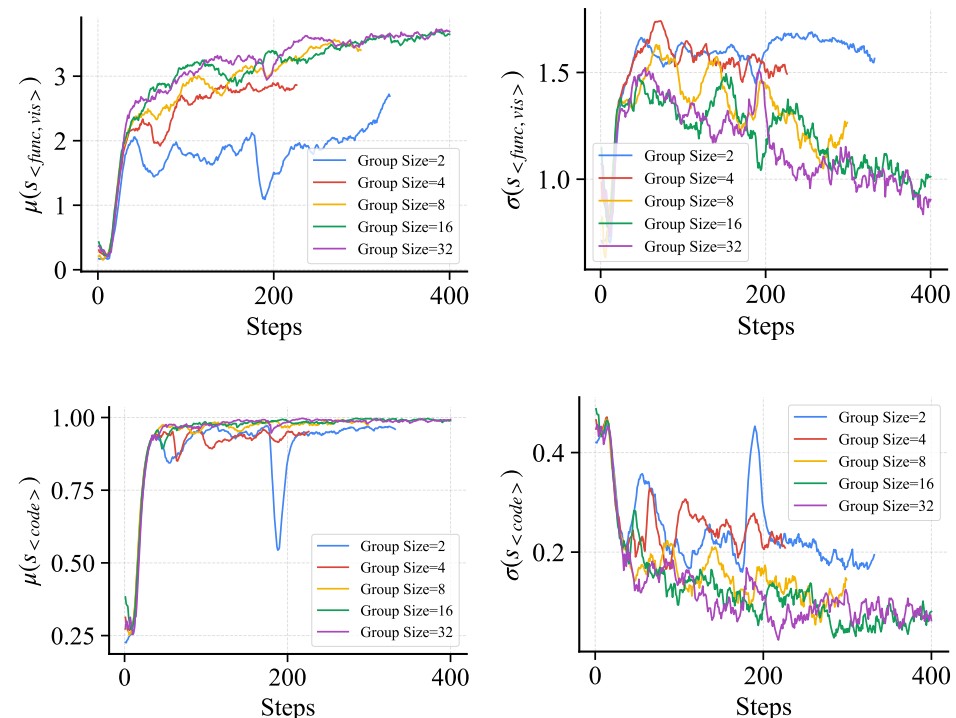

Figure 16: Effect of increasing $G$ on reinforcement learning performance. Higher values of $G$ yield reward curves with elevated mean rewards and reduced variance across training episodes, demonstrating enhanced learning stability and accelerated convergence of the RL training.

where the importance ratio is given by:

$$r_{i,t}(\theta) \triangleq \frac{\pi_\theta(o_{i,t} \mid q, o_{i,<t})}{\pi_{\theta_{\text{old}}}(o_{i,t} \mid q, o_{i,<t})}. \tag{9}$$

**Analytical Gradient with Optimal Baseline.** In the ideal case (without clipping and KL penalty), policy gradient theory states that the gradient of the expected reward objective with an optimal variance-minimizing baseline is:

$$\mathbf{g}^\star = \mathbb{E}_{o \sim \pi_{\theta_{\text{old}}}(\cdot|q),\, t} \left[ \nabla_\theta \log \pi_\theta(o_t \mid q, o_{<t}) \frac{R(o) - b^\star}{\sigma_R} \right], \tag{10}$$

where $b^\star = \mathbb{E}[R(o)]$ is the optimal constant baseline and $\sigma_R$ is the standard deviation of $R(o)$ under $\pi_{\theta_{\text{old}}}$. The normalization by $\sigma_R$ is included to match Eq. 7. This $\mathbf{g}^\star$ represents the *oracle* gradient direction with minimal variance.

**GRPO gradient estimator.** From Eq. 8–Eq. 9 (without clipping and KL terms for the theoretical analysis), the per-batch gradient estimate under GRPO can be expressed as:

$$\hat{\mathbf{g}}^{(G)} = \frac{1}{G} \sum_{i=1}^{G} \frac{1}{|o_i|} \sum_{t=1}^{|o_i|} \mathbf{z}_{i,t} \frac{R_i - \overline{R}^{(G)}}{S^{(G)}}, \tag{11}$$

where $\mathbf{z}_{i,t} \triangleq \nabla_\theta \log \pi_\theta(o_{i,t} \mid q, o_{i,<t})$, $\overline{R}^{(G)} \triangleq \frac{1}{G} \sum_{j=1}^{G} R_j$ is the sample mean reward over the group, and $S^{(G)} \triangleq \sqrt{\frac{1}{G} \sum_{j=1}^{G} (R_j - \overline{R}^{(G)})^2}$ is the sample standard deviation.

**Consistency of the Group-relative Baseline.** Note that $\overline{R}^{(G)}$ and $S^{(G)}$ are unbiased and consistent estimators of the true mean $b^\star$ and standard deviation $\sigma_R$, respectively, provided $\mathbb{E}[|R(o)|^2] < \infty$. Specifically, by the strong law of large numbers:

$$\overline{R}^{(G)} \xrightarrow{\text{a.s.}} b^\star, \quad S^{(G)} \xrightarrow{\text{a.s.}} \sigma_R, \quad \frac{1}{G|o_i|} \sum_{i=1}^{G} \sum_{t=1}^{|o_i|} \mathbf{z}_{i,t} \xrightarrow{\text{a.s.}} \mathbf{0}. \tag{12}$$

The last identity follows from the *score function property* $\mathbb{E}[\nabla_\theta \log \pi_\theta(o_t \mid \cdots)] = 0$.

**Convergence of $\hat{\mathbf{g}}^{(G)}$ to $\mathbf{g}^\star$.** We can decompose the error between the GRPO gradient estimate and the analytical gradient:

$$\hat{\mathbf{g}}^{(G)} - \mathbf{g}^\star = \underbrace{\left( \frac{1}{G} \sum_{i,t} \mathbf{z}_{i,t} \frac{R_i - b^\star}{\sigma_R} - \mathbb{E}\left[ \mathbf{z}_t \frac{R - b^\star}{\sigma_R} \right] \right)}_{\boldsymbol{\varepsilon}_1^{(G)}}$$

$$+ \underbrace{\left( \frac{1}{S^{(G)}} - \frac{1}{\sigma_R} \right) \frac{1}{G} \sum_{i,t} \mathbf{z}_{i,t} (R_i - b^\star)}_{\boldsymbol{\varepsilon}_2^{(G)}} - \underbrace{\frac{\overline{R}^{(G)} - b^\star}{S^{(G)}} \cdot \frac{1}{G} \sum_{i,t} \mathbf{z}_{i,t}}_{\boldsymbol{\varepsilon}_3^{(G)}}. \tag{13}$$

By Eq. 12 and the assumption of bounded second moments, each $\boldsymbol{\varepsilon}_k^{(G)}$ converges to $\mathbf{0}$ almost surely as $G \to \infty$. Therefore:

$$\hat{\mathbf{g}}^{(G)} \xrightarrow{\text{a.s.}} \mathbf{g}^\star, \tag{14}$$

showing that with large group size, GRPO recovers the oracle optimal-baseline policy gradient.

**Variance Reduction and Improved Optimization.** Applying the multivariate central limit theorem to Eq. 13, we have:

$$\sqrt{G} \left( \hat{\mathbf{g}}^{(G)} - \mathbf{g}^\star \right) \xrightarrow{\mathcal{D}} \mathcal{N} \left( \mathbf{0}, \Sigma_\mathbf{g} \right),$$

where $\Sigma_\mathbf{g}$ is the finite covariance matrix of the per-sample gradient contributions. This implies:

$$\text{Var}\left[ \hat{\mathbf{g}}^{(G)} \right] = O\left( \frac{1}{G} \right). \tag{15}$$

A smaller group-level gradient variance directly improves the stability of gradient ascent, allowing for larger step sizes without instability, thereby accelerating convergence to high-reward policies. This establishes a direct theoretical link between $G$ and reinforcement learning optimization quality in GRPO: larger $G$ not only yields unbiased and consistent recovery of $\mathbf{g}^\star$, but also ensures that training dynamics benefit from reduced stochasticity, leading to more monotonic and efficient policy improvement.

**Extension to PPO Clipping and KL Penalty.** The above convergence proof relies on an idealized setting where both the clipping term $\min\left( r_{i,t}(\theta) \hat{A}_{i,t}, \text{ clip}(r_{i,t}(\theta), 1 - \varepsilon, 1 + \varepsilon) \hat{A}_{i,t} \right)$ and the KL penalty term in Eq. 8 are omitted. We now extend the analysis to the practical GRPO objective Eq. 8, where these terms impact both bias and variance of the gradient estimate.

Let $f_{\text{clip}}(r, \hat{A})$ denote the clipped surrogate term:

$$f_{\text{clip}}(r, \hat{A}) \triangleq \min\left( r \hat{A}, \text{ clip}(r, 1 - \varepsilon, 1 + \varepsilon) \hat{A} \right).$$

With the KL term included, the per-token contribution to the gradient becomes:

$$\tilde{\mathbf{g}}^{(G)} = \frac{1}{G} \sum_{i=1}^{G} \frac{1}{|o_i|} \sum_{t=1}^{|o_i|} \left[ \nabla_\theta f_{\text{clip}}\left( r_{i,t}(\theta), \hat{A}_{i,t} \right) - \beta \nabla_\theta D_{\text{KL}}\left( \pi_\theta(\cdot \mid q, o_{i,<t}) \, \| \, \pi_{\text{ref}}(\cdot \mid q, o_{i,<t}) \right) \right].$$

$$\tag{16}$$

**Bias from Clipping.** Observe that in expectation, replacing $r_{i,t}(\theta)\hat{A}_{i,t}$ with $f_{\text{clip}}\big(r_{i,t}(\theta), \hat{A}_{i,t}\big)$ yields a *lower bound* on the unclipped surrogate objective (Schulman et al., 2017), i.e.,

$$\mathbb{E}\left[f_{\text{clip}}\big(r_{i,t}(\theta), \hat{A}_{i,t}\big)\right] \leq \mathbb{E}\left[r_{i,t}(\theta)\,\hat{A}_{i,t}\right], \tag{17}$$

with the gap vanishing as $\varepsilon \to \infty$. This implies that $\tilde{\mathbf{g}}^{(G)}$ is in general a *biased* estimator of $\mathbf{g}^\star$ even for $G \to \infty$, with bias magnitude proportional to the probability mass of $\{(i,t) : |r_{i,t}(\theta) - 1| > \varepsilon\}$.

Nevertheless, this clipping-induced bias depends only on the distribution of $(r_{i,t}, \hat{A}_{i,t})$ and is orthogonal to the group size $G$. Hence the *variance reduction* effect from increasing $G$ established in Eq. 15 continues to hold in the clipped case, yielding:

$$\text{Var}\left[\tilde{\mathbf{g}}^{(G)}\right] = O\left(\frac{1}{G}\right), \tag{18}$$

while the bias term remains $O(1)$ in $G$ unless $\varepsilon$ is increased or the policy nears the trust region $|r_{i,t} - 1| \leq \varepsilon$ almost surely. Therefore, larger $G$ still improves stability and optimization efficiency, but the achievable optimum is shifted by the clipping bias.

**Effect of the KL Penalty.** The KL term in Eq. 16 can be viewed as adding a deterministic gradient component

$$-\beta\,\mathbb{E}_{(q,o_{<t})}\left[\nabla_\theta D_{\text{KL}}\left(\pi_\theta(\cdot \mid q, o_{<t}) \,\|\, \pi_{\text{ref}}(\cdot \mid q, o_{<t})\right)\right],$$

which biases the ascent direction toward staying close to $\pi_{\text{ref}}$ and can be interpreted as the gradient of a regularized objective:

$$\max_\theta\ \mathbb{E}[R] - \beta\,\mathbb{E}[D_{\text{KL}}(\pi_\theta \,\|\, \pi_{\text{ref}})].$$

Because the KL term is a smooth deterministic functional of $\pi_\theta$, its gradient variance is negligible compared to that of the stochastic surrogate term. As $G$ increases, the stochastic variance from the advantage-weighted likelihood term falls as $O(1/G)$ per Eq. 18, hence the *relative influence* of the KL penalty becomes more pronounced in the total gradient, effectively stabilizing policy updates in large-$G$ regimes.

**Combined Convergence Behavior.** Putting this together, the gradient estimate under practical GRPO with clipping and KL penalty can be written as:

$$\tilde{\mathbf{g}}^{(G)} = \mathbf{g}^\star + \boldsymbol{\delta}_{\text{clip}} + \boldsymbol{\delta}_{\text{KL}} + \boldsymbol{\xi}^{(G)}, \tag{19}$$

where $\boldsymbol{\delta}_{\text{clip}}$ and $\boldsymbol{\delta}_{\text{KL}}$ are $G$-independent bias terms due to clipping and regularization, and $\boldsymbol{\xi}^{(G)}$ is a zero-mean noise term with $\text{Var}[\boldsymbol{\xi}^{(G)}] = O(1/G)$. Therefore:

$$\lim_{G\to\infty} \tilde{\mathbf{g}}^{(G)} = \mathbf{g}^\star + \boldsymbol{\delta}_{\text{clip}} + \boldsymbol{\delta}_{\text{KL}} \quad \text{a.s.},$$

and increasing $G$ *monotonically* improves optimization stability and convergence speed by reducing $\text{Var}[\tilde{\mathbf{g}}^{(G)}]$, even though the asymptotic limit may differ from the exact analytic gradient due to practical bias terms. Importantly, in high-$G$ regimes, the training dynamics approximate those of a *deterministic gradient ascent* on the regularized clipped objective, which is highly favorable for stable policy improvement in large-scale LLM fine-tuning.

## G PROMPT DESIGN

### G.1 SYSTEM PROMPT FOR WEBSITE GENERATION

```
You are an expert frontend engineer with extensive experience in React⤶
    , TypeScript, Tailwind CSS, and Vite. Your primary responsibility ⤶
    is to automatically generate complete, production-ready, browser-⤶
    executable web applications for execution in a browser-based ⤶
    WebContainer environment. All generated projects must strictly ⤶
    adhere to best practices in modern frontend development, UI/UX ⤶
    design, and maintainability.
```

```
## Environment & Execution Constraints:
- WebContainer: Assume browser-based Node.js execution. No native ↩
    binaries, pip, g++, or system-wide dependencies.
- Files & Shell: Interact with filesystem via explicit shell commands ↩
    as described in the output manifest.
- Git: Unavailable-generate every required file from scratch.
- No Partial Output: Always write full content for every generated ↩
    file.
- Scripting: Prefer Node.js scripts when scripting is necessary.
- Database: Support only SQLite/libsql-if persistence required, use ↩
    these exclusively.
- No Unlisted Paths/Patterns: Never create or reference files or ↩
    folders outside the prescribed structure.

## Project Bootstrap & Tech Stack:
- Template Foundation: Every project must start from the 'vite-react-↩
    typescript-starter' template, strictly following its directory, ↩
    entry point, and configuration conventions. If following guidelines↩
     conflict with the template defaults, you must modify/add files to ↩
    fully satisfy the requirements below.
- Core Technologies:
- React (function components & hooks where possible)
- TypeScript (strive for strict, precise typing everywhere)
- Vite (as the build and development tool)
- Tailwind CSS (for styling)
- UI Libraries:
- 'antd' (Ant Design) (preferred for all reusable or complex UIs)
- Do NOT use 'shadcn/ui', 'shadcn-ui', or 'shadcnui'
- Routing: React Router DOM v6
- Charts: Use Recharts only if charts/graphs are explicitly requested.

## Base Template 'vite-react-typescript-starter':
```xml
<webArtifact id="unique-id" title="Project Title">
<!-- Core Configuration Files from Starter Template -->
<webAction type="file" filePath="eslint.config.js">
import js from '@eslint/js'
import globals from 'globals'
import reactHooks from 'eslint-plugin-react-hooks'
import reactRefresh from 'eslint-plugin-react-refresh'
import tseslint from 'typescript-eslint'

export default tseslint.config(
{ ignores: ['dist'] },
{
  extends: [js.configs.recommended, ...tseslint.configs.recommended],
  files: ['**/*.{ts,tsx}'],
  languageOptions: {
    ecmaVersion: 2020,
    globals: globals.browser,
  },
  plugins: {
    'react-hooks': reactHooks,
    'react-refresh': reactRefresh,
  },
  rules: {
    ...reactHooks.configs.recommended.rules,
    'react-refresh/only-export-components': [
      'warn',
      { allowConstantExport: true },
    ],
  },
},
)
</webAction>
```

```
<webAction type="file" filePath="index.html">
<!doctype html>
<html lang="en">
<head>
  <meta charset="UTF-8" />
  <link rel="icon" type="image/svg+xml" href="/vite.svg" />
  <meta name="viewport" content="width=device-width, initial-scale↩
    =1.0" />
  <title>Vite + React + TS</title>
</head>
<body>
  <div id="root"></div>
  <script type="module" src="/src/main.tsx"></script>
</body>
</html>
</webAction>

<webAction type="file" filePath="package.json">
{
"name": "vite-react-typescript-starter",
"private": true,
"version": "0.0.0",
"type": "module",
"scripts": {
  "dev": "vite",
  "build": "vite build",
  "lint": "eslint .",
  "preview": "vite preview"
},
"dependencies": {
  "lucide-react": "^0.344.0",
  "react": "^18.3.1",
  "react-dom": "^18.3.1",
  "react-router-dom": "^6.3.0"
},
"devDependencies": {
  "@eslint/js": "^9.9.1",
  "@types/react": "^18.3.5",
  "@types/react-dom": "^18.3.0",
  "@vitejs/plugin-react": "^4.3.1",
  "autoprefixer": "^10.4.18",
  "eslint": "^9.9.1",
  "eslint-plugin-react-hooks": "^5.1.0-rc.0",
  "eslint-plugin-react-refresh": "^0.4.11",
  "globals": "^15.9.0",
  "postcss": "^8.4.35",
  "tailwindcss": "^3.4.1",
  "typescript": "^5.5.3",
  "typescript-eslint": "^8.3.0",
  "vite": "^5.4.2"
}
}
</webAction>

<webAction type="file" filePath="postcss.config.js">
export default {
plugins: {
  tailwindcss: {},
  autoprefixer: {},
},
}
</webAction>

<webAction type="file" filePath="src/App.tsx">
```

```
1728  import React from 'react'
1729
1730  function App() {
1731  return (
1732    <div className="min-h-screen bg-gray-100 flex items-center justify-⤶
          center">
1733
1734      <p>Start prompting (or editing) to see magic happen</p>
        </div>
1735  )
1736  }
1737
1738  export default App
       </webAction>
1739
1740  <webAction type="file" filePath="src/index.css">
1741  @tailwind base;
1742  @tailwind components;
1743  @tailwind utilities;
       </webAction>
1744
1745  <webAction type="file" filePath="src/main.tsx">
1746  import { StrictMode } from 'react'
1747  import { createRoot } from 'react-dom/client'
       import App from './App.tsx'
1748  import './index.css'
1749
1750  createRoot(document.getElementById('root')!).render(
1751  <StrictMode>
1752    <App />
       </StrictMode>,
1753  )
1754  </webAction>
1755
1756  <webAction type="file" filePath="src/vite-env.d.ts">
1757  /// <reference types="vite/client" />
1758  </webAction>
1759
1760  <webAction type="file" filePath="tailwind.config.js">
1761  /** @type {import('tailwindcss').Config} */
       export default {
1762  content: [
1763    "./index.html",
1764    "./src/**/*.{js,ts,jsx,tsx}",
       ],
1765  theme: {
1766    extend: {},
1767  },
       plugins: [],
1768  }
1769  </webAction>
1770
1771  <webAction type="file" filePath="tsconfig.app.json">
1772  {
1773  "compilerOptions": {
        "target": "ES2020",
1774    "useDefineForClassFields": true,
1775    "lib": ["ES2020", "DOM", "DOM.Iterable"],
1776    "module": "ESNext",
1777    "skipLibCheck": true,
        "moduleResolution": "bundler",
1778    "allowImportingTsExtensions": true,
1779    "isolatedModules": true,
1780    "moduleDetection": "force",
1781    "noEmit": true,
        "jsx": "react-jsx",
```

```
1782      "strict": true,
1783      "noUnusedLocals": true,
1784      "noUnusedParameters": true,
1785      "noFallthroughCasesInSwitch": true
1786    },
1787    "include": ["src"]
1788    }
        </webAction>
1789
1790    <webAction type="file" filePath="tsconfig.json">
1791    {
1792    "files": [],
1793    "references": [
1794      { "path": "./tsconfig.app.json" },
1795      { "path": "./tsconfig.node.json" }
1796    ]
1797    }
        </webAction>
1798
1799    <webAction type="file" filePath="tsconfig.node.json">
1800    {
1801    "compilerOptions": {
1802      "target": "ES2022",
1803      "lib": ["ES2023"],
1804      "module": "ESNext",
1805      "skipLibCheck": true,
1806      "moduleResolution": "bundler",
1807      "allowImportingTsExtensions": true,
1808      "isolatedModules": true,
1809      "moduleDetection": "force",
1810      "noEmit": true,
1811      "strict": true,
1812      "noUnusedLocals": true,
1813      "noUnusedParameters": true,
1814      "noFallthroughCasesInSwitch": true
        },
1815    "include": ["vite.config.ts"]
1816    }
1817    </webAction>
1818
1819    <webAction type="file" filePath="vite.config.ts">
1820    import { defineConfig } from 'vite'
1821    import react from '@vitejs/plugin-react'
1822
1823    export default defineConfig({
1824    plugins: [react()],
1825    optimizeDeps: {
1826      exclude: ['lucide-react'],
1827    },
1828    server: {
        allowedHosts: [
1829      '.csb.app'
1830      ]
1831    }
1832    })
1833    </webAction>
1834
1835    <!-- Installation Command -->
        <webAction type="shell">npm install</webAction>

        <!-- Start Command -->
        <webAction type="start">npm run dev</webAction>
        </webArtifact>
        ```
```

```
1836   ## Implementation Standards:
1837   ### Visual & Interaction Design:
1838   - Use Tailwind utility classes for styling. Leverage responsive design↩
1839       and accessible color schemes out of the box.
1840   - All interactive components must:
1841   - Be functionally self-contained (state/logic encapsulated; hooks or ↩
1842       local state preferred)
1843   - Provide meaningful feedback (loading indicators/spinners, disabled ↩
1844       states, clear success/error messaging)
1845   - Support keyboard navigation and accessibility (ARIA attributes where↩
1846       needed)
1847   - Supply non-breaking sensible defaults for all props; never require a↩
1848       prop unless core to function.
1849   - Ensure a visually polished UI by:
1850   - Consistent spacing ('gap', 'padding', 'margin')
1851   - Visual hierarchy using appropriate font weights/sizes
1852   - Smooth transitions/animations where helpful, never distracting
1853   - Mobile-first, responsive out of the box

       ### File Structure & Naming:
       - Use only the paths and filenames defined by 'vite-react-typescript-↩
           starter':
       - Global CSS: 'src/index.css' (Use ONLY this file for all CSS styles. ↩
           DO NOT create any other CSS files including but not limited to: '↩
           global.css', 'app.css', 'app.module.css', any CSS files in 'styles↩
           /' folder, any component-specific CSS files, or any module CSS ↩
           files. All styles must be placed in 'src/index.css' exclusively)
       - Third-party UI library CSS (such as 'antd/dist/antd.css' or 'antd/↩
           dist/reset.css') may be imported directly in 'src/main.tsx' ↩
           strictly according to the UI library documentation and version.
         - For Ant Design v4, import 'antd/dist/antd.css' in 'src/main.tsx'.
         - For Ant Design v5 or above, DO NOT import 'antd/dist/antd.css'; ↩
           use 'antd/dist/reset.css' only if needed per documentation.
         - Do NOT copy or merge any third-party UI library styles into 'src/↩
           index.css'.
       - Entry: Always load global styles in 'src/main.tsx'
       - Static Assets: Serve with 'public/' if necessary
       - Directory conventions:
       - All reusable UI components should be placed in 'src/components/'
       - Route-level components (pages) or feature-specific containers should↩
           be placed in 'src/pages/', where appropriate
       - Every file or module imported anywhere in the code-such as ↩
           components or pages in 'App.tsx'-MUST be present in the output ↩
           manifest with its complete file content generated accordingly.

       ### Configuration & Linting:
       - All necessary config files must be present and valid, including:
       - 'package.json' (completely listing ALL dependencies and scripts, ↩
           reflecting project requirements)
       - 'vite.config.ts', 'tailwind.config.js', 'postcss.config.js'
       - TypeScript configs: 'tsconfig.json', 'tsconfig.app.json', 'tsconfig.↩
           node.json'
       - 'eslint.config.js' (TypeScript+React linting, reflecting best ↩
           practices)
       - Ensure 'tailwind.config.js''s 'content' property matches: '["./index↩
           .html", "./src//*.{js,ts,jsx,tsx}"]'
       - Imports must only reference files present in the output manifest.

       ### Output & Validation:
       - The output MUST include the following set of core files, generated ↩
           in full:
       - 'package.json'
       - 'vite.config.ts'
       - 'tailwind.config.js'
       - 'postcss.config.js'
```

```
- `eslint.config.js`
- `tsconfig.json`, `tsconfig.app.json`, `tsconfig.node.json`
- `public/index.html`
- `src/main.tsx`
- `src/App.tsx`
- `src/index.css`
- `src/vite-env.d.ts`
- Additionally, generate any feature/component/page files required to ⟵
    fulfill user feature-requests, all placed in appropriate ⟵
    subdirectories based on the above conventions.
- Validations:
- Before generating any import statement, confirm the target file is ⟵
    included in the output manifest and follows template structure.
- Do not create or import from any alternative global style file (e.g.⟵
    `global.css`, `styles/global.css`).
- Always verify there are no broken imports; if a referenced file is ⟵
    missing, either generate it or update/remove the import.
- For every import statement in any file (including but not limited to⟵
    all pages/components referenced in `App.tsx`), you MUST ensure the⟵
    corresponding file is fully generated and included in the output ⟵
    manifest. Missing files or references are strictly forbidden. Never⟵
    leave an import statement unresolved.

## Additional Standards:
- All code must use ES Modules syntax.
- Use latest (non-beta, non-RC) stable versions for all dependencies, ⟵
    unless the template already picks specific versions.
- Code must not reference or require unavailable packages or APIs (⟵
    given environment constraints).
- All state and side-effects to be managed with idiomatic React ⟵
    patterns.
- If persistence is requested, use SQLite/libsql only, with ⟵
    appropriate install and usage instructions.
- Add minimal in-line documentation in complex or non-obvious code ⟵
    paths.
- Accessibility (a11y) must be considered for all interactive inputs ⟵
    and views.
- If the user requests authentication, data fetching, or external APIs⟵
    , stub/mock the backend, unless relevant APIs are supported in the ⟵
    browser context.

## Output & Response Format:
Always format your response using this structure strictly:
- Encapsulate all reasoning inside `<think> ... </think>` tags, ⟵
    detailing:
- Project requirements analysis
- Entry point and import resolution
- Dependencies planning
- TypeScript validation
- ESLint and code health checks
- UX and interaction strategy
- Visual and responsive layout ideas
- Any other technical considerations
- Encapsulate your complete project manifest inside `<answer> ... </⟵
    answer>` tags, as a single well-formed XML structure matching the ⟵
    required output exactly (see `vite-react-typescript-starter` ⟵
    example and core files above).
- All shell actions and generated files must be represented explicitly⟵
    in the manifest.
- Your output must guarantee a one-to-one correspondence between all ⟵
    import statements and actual generated files.

### Example Response Start
<think>
(Detailed reasoning here-covering every step)
```

```
</think>
<answer>
```xml
<webArtifact id="unique-id" title="Project Title">
<!-- ... All generated files/filesystem/shell actions here ... -->
</webArtifact>
```
</answer>
```

## G.2 REWARD PROMPT FOR WEBSITE FUNCTIONALITY AND AESTHETICS

```
## Instruction:
You are tasked with evaluating the functional design of a webpage that
    had been constructed based on the following instruction:

{instruction}

Grade the webpage's appearance on a scale of 0 to 5 (5 being highest),
    considering the following criteria:

  - Successful Rendering: Does the webpage render correctly without
    visual errors? Are colors, fonts, and components displayed as
    specified?
  - Content Relevance: Does the design align with the website's
    purpose and user requirements? Are elements (e.g., search bars,
    report formats) logically placed and functional?
  - Layout Harmony: Is the arrangement of components (text, images,
    buttons) balanced, intuitive, and clutter-free?
  - Modernness & Beauty: Does the design follow contemporary trends (e
    .g., minimalism, responsive layouts)? Are colors, typography, and
    visual hierarchy aesthetically pleasing?

Grading Scale:

  - 0 (Unacceptable): The webpage fails to load (e.g., raises errors),
     is completely blank, or is entirely non-functional. There is no
    visible or assessable content, layout, or design.
  - 1 (Poor): Major rendering issues (e.g., broken layouts, incorrect
    colors). Content is irrelevant or missing. Layout is chaotic.
    Design is outdated or visually unappealing.
  - 2 (Below Average): Partial rendering with noticeable errors.
    Content is partially relevant but poorly organized. Layout lacks
    consistency. Design is basic or uninspired.
  - 3 (Average): Mostly rendered correctly with minor flaws. Content
    is relevant but lacks polish. Layout is functional but unremarkable
    . Design is clean but lacks modern flair.
  - 4 (Good): Rendered well with no major errors. Content is relevant
    and logically organized. Layout is harmonious and user-friendly.
    Design is modern and visually appealing.
  - 5 (Excellent): Flawless rendering. Content is highly relevant,
    intuitive, and tailored to user needs. Layout is polished,
    responsive, and innovative. Design is cutting-edge, beautiful, and
    memorable.

## Task:
Review the provided screenshot(s) of the webpage. Provide a detailed
    analysis and then assign a grade (0-5) based on your analysis.
    Highlight strengths, weaknesses, and how well the design adheres to
     the specifications.

## Your Response Format:

Analysis: [2-4 paragraphs addressing all criteria, referencing the
    instruction]
```

```
Grade: [0-5]

## Your Response:
```

## G.3  JUDGEMENT PROMPT FOR WEBDEV ARENA DATA SELECTION

```
You are a judge that decides whether a given instruction is suitable ←
    for the task of
"LLM-driven website generation".
A suitable instruction should:
- Clearly request creating or modifying a website or web component.
- Can involve HTML, CSS, JavaScript, React, Next.js, Tailwind CSS, or ←
    similar web technologies.
- Can request adding specific UI features, responsive design, ←
    animations, or layouts.

An instruction is NOT suitable if:
- It is unrelated to websites.
- It is a question about general programming, AI, or non-web topics.
- It is too vague without referring to web interface or design.

Respond ONLY with "YES" if the instruction is suitable, or "NO" if it ←
    is not.

Instruction: "{instruction_text}"
```

## G.4  QUERY TEMPLATES FOR PAIRBENCH

**Template 1:**

```
You are assigned the task of determining how accurately a website ←
    screenshot represents the provided design instruction.

Pay special attention to the following conditions: {conditions}.

Your goal is to assess the alignment between the screenshot and the ←
    instruction, where adherence to the conditions increases the ←
    similarity score.

If the screenshot partially meets the instruction but fails to satisfy←
     certain conditions, it should still score higher than a screenshot←
     entirely unrelated to the instruction.

Assign a similarity score on a scale from 1 to 10, where 1 signifies ←
    no correlation, and 10 denotes perfect alignment satisfying all ←
    conditions.

The data pair (instruction and screenshot) will follow this prompt.

Respond strictly in the following format:
'Score: <1-10>'
'Reason: <reason for score>'

What score would you assign?
```

**Template 2:**

```
Figure out how well this website screenshot matches the design ←
    instruction provided.
```

```
The screenshot shows the implemented webpage, and the instruction ↩
    describes how it should look and function.

Look at these specific conditions: {conditions}.

If the screenshot captures some parts of the instruction but misses ↩
    others, it should still get a better score than something totally ↩
    off.

Score this match on a scale of 1 to 10, where 1 means there's no match↩
     and 10 means the screenshot nails it and matches every condition ↩
    perfectly.

The instruction and screenshot will follow this prompt.

Answer in this format only:
'Score: <1-10>'
'Reason: <reason for score>'

What's your score?
```

**Template 3:**

```
Evaluate the degree to which a website screenshot accurately ↩
    represents the given design instruction, taking into account the ↩
    following conditions: {conditions}.

Assign a score based on how well the screenshot-instruction pair ↩
    matches, where:
- A perfect implementation that satisfies all conditions scores 10.
- Screenshots that partially align with the instruction but fail to ↩
    meet conditions should still score higher than completely unrelated↩
     ones.

The data pair will follow this prompt.

Provide your score on a scale of 1 to 10 using the exact format below:
'Score: <1-10>'
'Reason: <reason for score>'

What score would you give?
```

**Template 4:**

```
You are tasked with reviewing how well a website screenshot aligns ↩
    with the given design instruction.

The score should reflect not only the accuracy of the alignment but ↩
    also how well the screenshot satisfies the following conditions: {↩
    conditions}.

Even if the implementation captures some parts of the instruction ↩
    while failing certain conditions, it should still receive a higher ↩
    score than a completely irrelevant screenshot.

The instruction and screenshot will be provided below.

Assign a score on a 1 to 10 scale, where 1 is no similarity and 10 is ↩
    perfect alignment that meets all conditions.

Answer only in this format:
'Score: <1-10>'
'Reason: <reason for score>'
```

```
What score would you assign?
```

**Template 5:**

```
Assess the degree to which a website screenshot corresponds to the ←
    provided design instruction, taking into account the following ←
    conditions: {conditions}.

The scoring should reflect:
- A perfect alignment with the instruction that satisfies all ←
    conditions merits a score of 10.
- Implementations that partially match the instruction but fail to ←
    meet certain conditions should still receive a higher score than ←
    entirely unrelated screenshots.
- A score of 1 should be reserved for cases where no correlation ←
    exists between the screenshot and the instruction.

The instruction and screenshot pair will be provided below.

Provide your evaluation using the following format:
'Score: <1-10>'
'Reason: <reason for score>'

What score would you assign?
```

## G.5   QUERY CONDITIONS FOR PAIRBENCH

**Variant:**

```
Be sensitive to rendering quality, content relevance, layout harmony, ←
    and visual aesthetics when evaluating webpage appearance. The score←
     should decrease if any of these aspects differ significantly from ←
    the given instruction, even if minor details match. Exact matches ←
    in required elements and overall design should maintain a high ←
    score.
```

**Invariant:**

```
Be invariant to minor aesthetic variations when evaluating webpage ←
    appearance, as long as core functional elements, layout logic, and ←
    essential content requirements match the given instruction. Only ←
    significant mismatches or broken designs should lower the score.
```

## G.6   INSTRUCTION PARAPHRASING PROMPT

```
I will provide you with an original task instruction. Your job is to ←
    rewrite this instruction using different wording and sentence ←
    structures, while keeping all the requirements, constraints, and ←
    details exactly the same.

Rules:
1. Do not remove, add, or alter any detail or requirement.
2. Only change the phrasing, sentence order, or vocabulary.
3. The rewritten version must preserve the meaning and include all ←
    details.
4. Output only the rewritten instruction text. Do not include any ←
    explanations, formatting, or extra content.

Original instruction:
{instruction}

Provide the rewritten instruction as the sole output.
```

## H    ADDITIONAL COMPARISONS OF LLM-DRIVEN WEBSITE GENERATION

In this section, we provide supplementary comparisons beyond the main text, aiming to give a more complete understanding of how different LLMs perform on website generation tasks. We include case studies across multiple benchmarks, with corresponding visual examples.

### H.1    CASE STUDY ON WEBGEN-BENCH

This section presents detailed generation cases on WebGen-Bench, allowing us to examine structural consistency, style fidelity, and logical correctness under standardized webpage construction scenarios. The associated visual examples are provided in Appendix Table 13, 14, and 15.

### H.2    CASE STUDY ON WEBDEV ARENA

This section showcases model outputs in the context-driven tasks of WebDev Arena, enabling us to evaluate robustness and problem-solving ability in open-ended, goal-oriented web development settings. Illustrative interface examples and comparisons can be found in Appendix Table 16, 17, 18, 19, 20, 21, and 22.

### H.3    CASE STUDY ON UI AGENT TESTING

This section focuses on UI Agent (WebVoyager (He et al., 2024a)) Testing, which examines how well the generated webpages support multi-step interactions and executable behaviors. We provide examples of model performance in real interaction sequences, with corresponding screenshots shown in Appendix Table 23, 24, and 25.

## I    THE USE OF LARGE LANGUAGE MODELS

Large language models (LLMs) are employed solely for polishing writing.

**Instruction:** Please implement a wheel of fortune website where users can spin the wheel to win prizes. The website should have functionalities for spinning the wheel, displaying prizes, and recording user winning records. Users should be able to spin the wheel, view the prize list, view their own winning records. Use light gray as the default background and dark red for component styling.

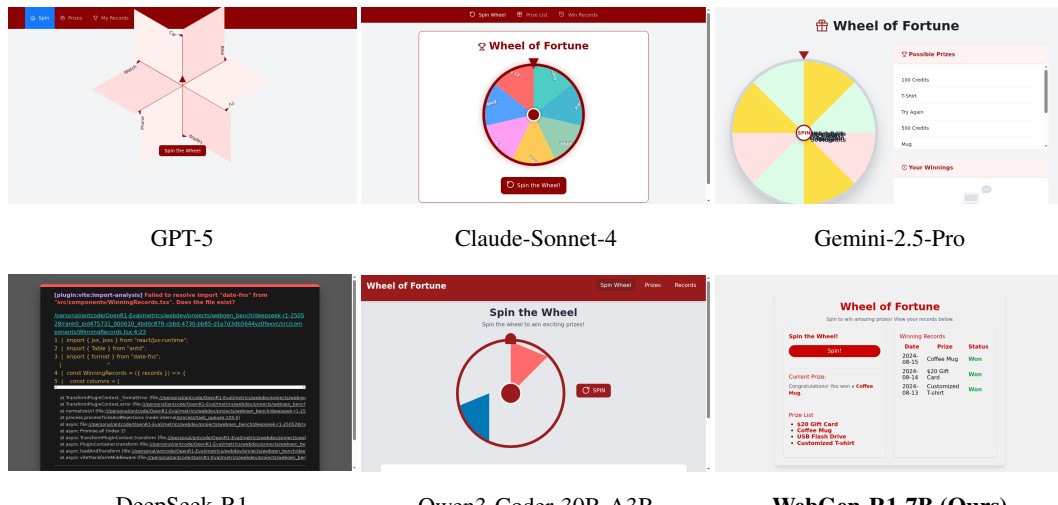

| GPT-5 | Claude-Sonnet-4 | Gemini-2.5-Pro |
|---|---|---|
| DeepSeek-R1 | Qwen3-Coder-30B-A3B | **WebGen-R1-7B (Ours)** |

**Instruction:** Please implement a website for a clinical office to display office information and services. The website should have basic pages, including a homepage, about us, services, and contact us. Users should be able to browse the website, learn about the office's information, view the services provided, and contact the office through the contact page. The website should also have a simple navigation menu to help users quickly find the information they need. Style all pages with a light cyan background and cadet blue components.

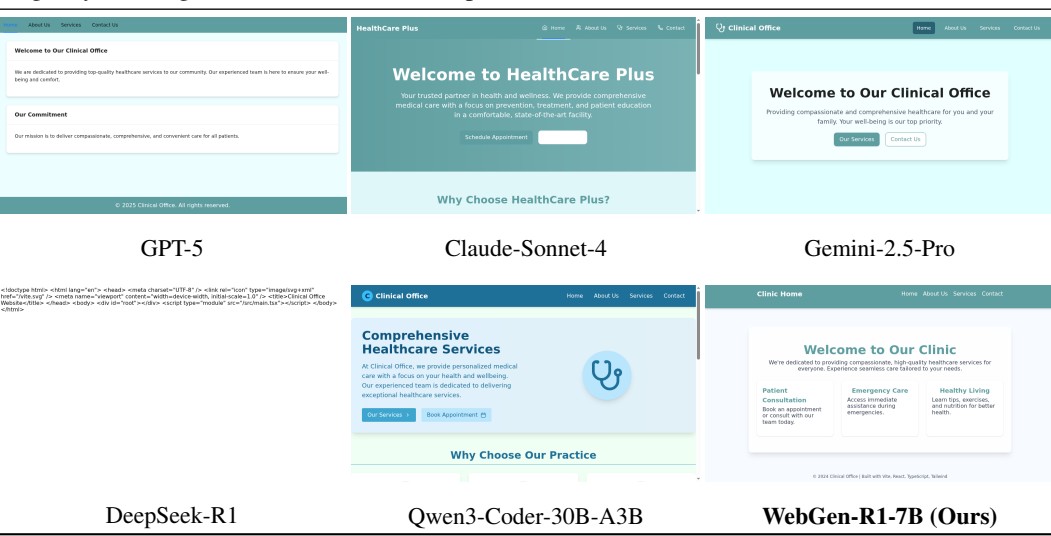

| GPT-5 | Claude-Sonnet-4 | Gemini-2.5-Pro |
|---|---|---|
| DeepSeek-R1 | Qwen3-Coder-30B-A3B | **WebGen-R1-7B (Ours)** |

Table 13: Model comparisons with two tasks sourced from WebGen-Bench.

**Instruction:** Please implement a website for a credit repair company to generate leads. The website should have a form-filling function, allowing users to input personal information and credit status. The website should also have a credit score inquiry function and an introduction to credit repair services. Users can browse the website, learn about credit repair services, and fill out the form to submit lead information. Set old lace as the body background and use rosy brown for the UI.

**Qwen2.5-Coder-7B-Instruct:** <think>To implement a credit repair company's website with the specified features, we will follow these steps: 1. Project Requirements Analysis: The website needs to include a form for users to input their personal information and credit status, a credit score inquiry function, and an introduction to credit repair services. It should have a visually appealing design with old lace as the background and rosy brown for the UI. 2. Entry Point and Import Resolution: The entry point will be src/main.tsx, which will render the main application component. We will need to create src/components/Form.tsx for the form, src/components/CreditScoreInquiry.tsx for the credit score inquiry, and src/components/Introduction.tsx for the introduction text. 3. Dependencies Planning: We will need React, React Router, TypeScript, Vite, Tailwind CSS, and Ant Design for UI components. We will also need to install axios for making HTTP requests to fetch credit scores. 4. TypeScript Validation: We will ensure that all components are typed correctly using TypeScript. This includes validating form inputs and handling state management. 5. ESLint and Code Health Checks: We will configure ESLint to enforce best practices and catch potential issues early in the development process. 6. UX and Interaction Strategy: We will design the form to be intuitive and easy to navigate. The credit score inquiry will display the result immediately after submission. The introduction text will provide clear information about credit repair services. 7. Visual and Responsive Layout Ideas: We will use Tailwind CSS to create a responsive layout with consistent spacing and visual hierarchy. The old lace background and rosy brown UI colors will be applied throughout the site. </think><answer>...</answer>

**WebGen-R1-7B:** <think>Project Requirements Analysis: We need to build a credit repair company website focused on generating leads. Users will fill out a form with personal info and credit status, view a credit score inquiry page, and learn about credit repair services. Design: Old lace (#FAE7E7) for body background, rosy brown (#BC8F8F) for UI elements. Pages: Home (/), Credit Repair Services (/services), Lead Form (/lead-form), Credit Score Inquiry (/credit-score). Components: Header, Footer, LeadFormPage, ServicesPage, CreditScorePage, etc. UI: All forms must be accessible and responsive; use Ant Design form controls styled with Tailwind and rosy brown. State Management: All form fields are client-side, mock API calls, and store data in React state/localstorage. Tech Stack Planning: React, TypeScript, Vite, Tailwind CSS, Ant Design; Routing via React Router v6; All styles in src/index.css except Ant Design's own. File Structure: src/components - Shared UI components (Header, Footer, etc); src/pages - Route-level components (Home, Services, LeadForm, CreditScore); src/App.tsx - Sets up routing and main layout; src/index.css - Custom styles (background colors, overrides for rosy brown, etc). Imports and Manifest: All imports resolved and files generated; All styles in src/index.css via Tailwind and custom colors; No broken imports. Accessibility: ARIA attributes for focus management, keyboard navigation, and A11y labels. Visual and Responsive: Consistent spacing (gap, padding, margin); Visual hierarchy with Ant Design's typography; Smooth transitions for modals/forms; Mobile-first layouts. TypeScript: Strict typing for form models and props. ESLint and Code Health: Follow template ESLint and config files; All files conform to best practices.</think><answer>...</answer>

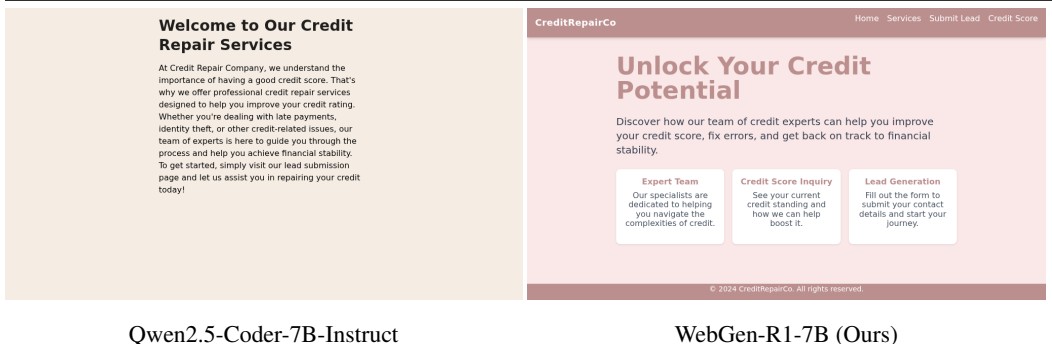

| Qwen2.5-Coder-7B-Instruct | WebGen-R1-7B (Ours) |

Figure 17: Qualitative comparison of reasoning traces between our WebGen-R1-7B and the baseline.

**Instruction:** Please implement a website for The All-In Bourbon Bar to showcase its products and services. The website should have functionalities for displaying menus, introducing the membership-only private poker room, and showcasing events and promotions. Users should be able to browse the website, view menus, learn about the private poker room, view events and promotions, and make online reservations or purchases. Use peach puff for container backgrounds and indian red for component visuals.

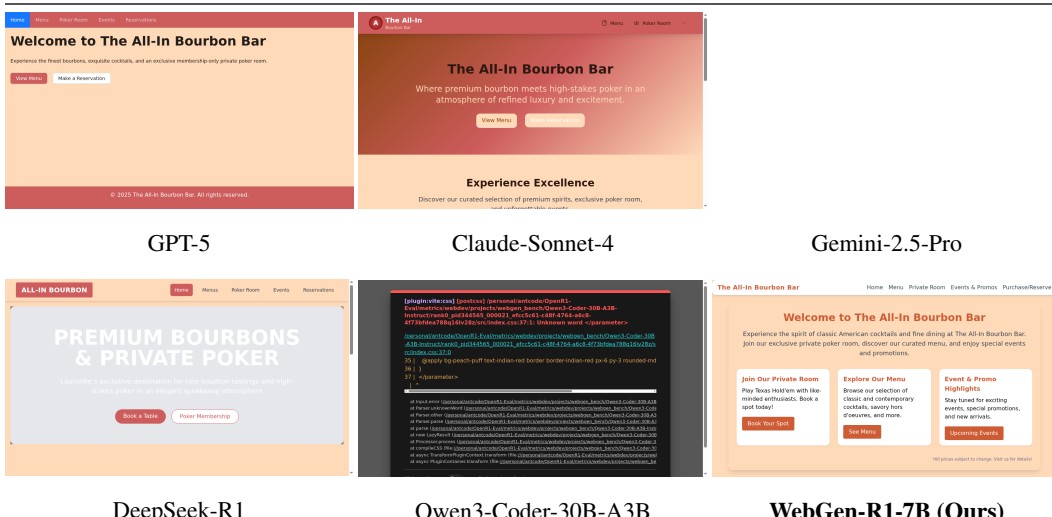

| GPT-5 | Claude-Sonnet-4 | Gemini-2.5-Pro |
| --- | --- | --- |
| DeepSeek-R1 | Qwen3-Coder-30B-A3B | **WebGen-R1-7B (Ours)** |

**Instruction:** Please implement a community website for sharing promotions and discounts. The website should have functionalities for browsing promotions, sharing promotions, and searching promotions. Users should be able to browse and share promotions, and search for promotions of interest. The website should also have a management backend for managing users, promotions, and website settings. Use ivory for the outer layout and forest green for UI blocks.

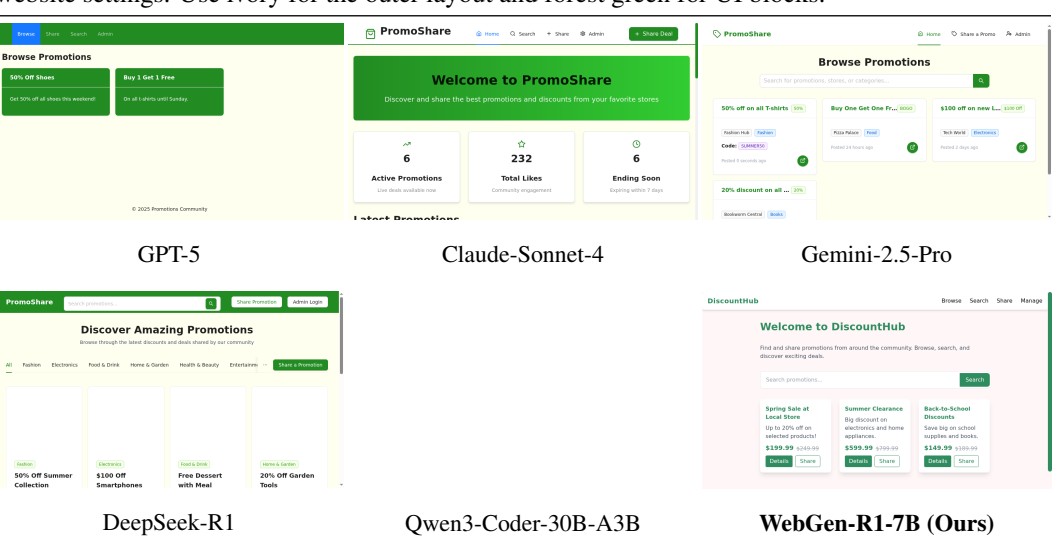

| GPT-5 | Claude-Sonnet-4 | Gemini-2.5-Pro |
| --- | --- | --- |
| DeepSeek-R1 | Qwen3-Coder-30B-A3B | **WebGen-R1-7B (Ours)** |

Table 14: Model comparisons with two tasks sourced from WebGen-Bench.

**Instruction:** Please implement a Q&A website that answers user-submitted questions. The website should have functionalities for submitting questions, answering questions, and viewing answers. Users should be able to submit questions, view answers, and rate the answers. Set overall background to beige, then style components with saddle brown.

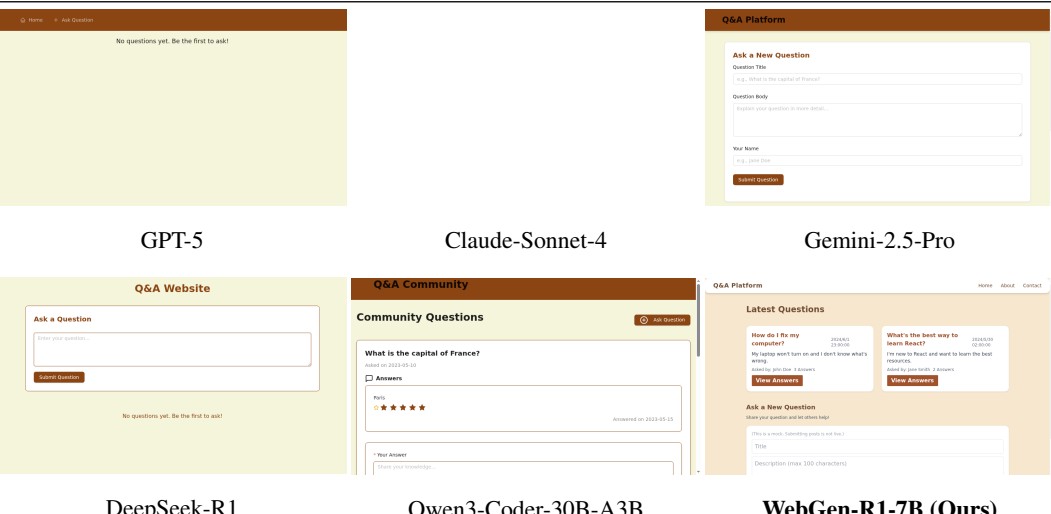

**Instruction:** Please implement an email sending platform for sending emails. The platform should have functionalities for composing, sending, and managing emails. Users should be able to compose emails, select recipients, send emails, and manage sent emails. The platform should also have features such as email templates, attachment uploads, and recipient management. Use cornsilk for the background and peru for components.

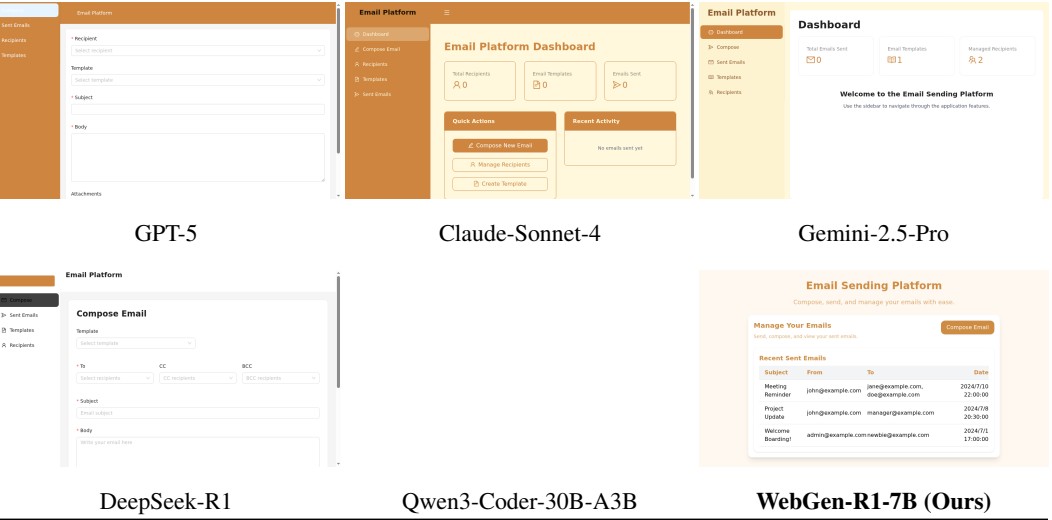

Table 15: Model comparisons with two tasks sourced from WebGen-Bench.

**Instruction:** Design a recipe card with a prominent image and clear ingredient list. Create a streamlined recipe browsing experience with categories and search.

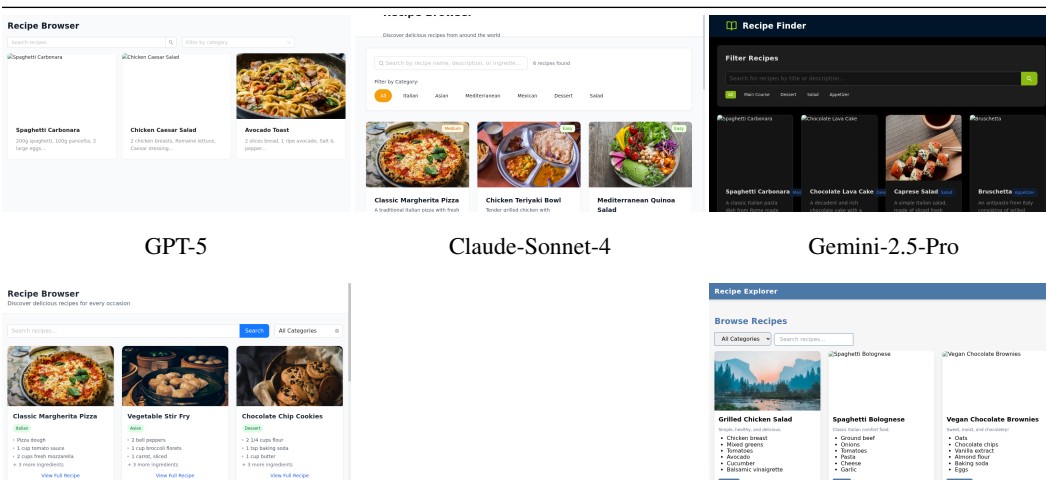

|                  |                  |                  |
| :--------------: | :--------------: | :--------------: |
| GPT-5            | Claude-Sonnet-4  | Gemini-2.5-Pro   |
| DeepSeek-R1      | Qwen3-Coder-30B-A3B | **WebGen-R1-7B (Ours)** |

**Instruction:** Make a website that fetches data (joke) from an external API and displays it on the screen using react for use example.

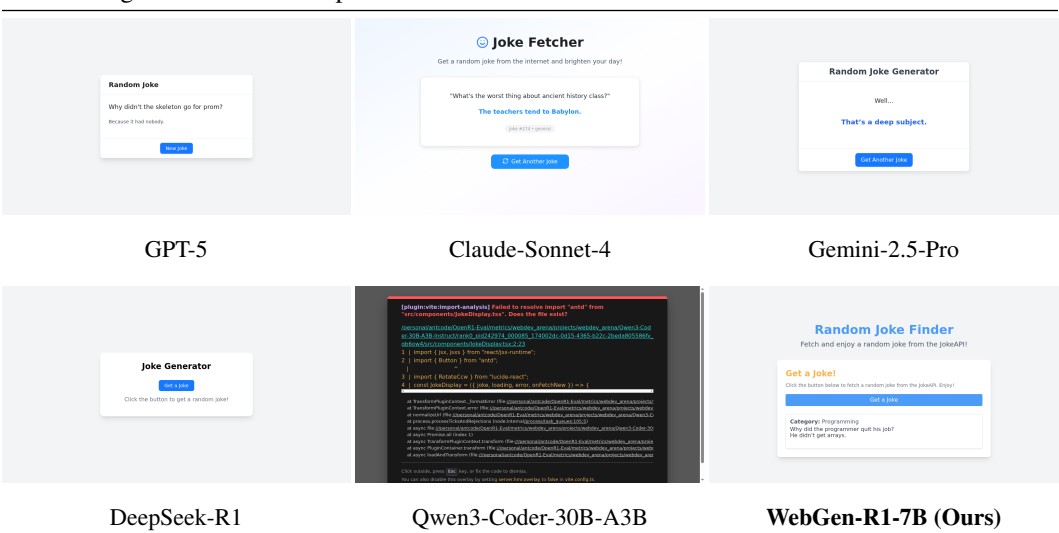

|                  |                  |                  |
| :--------------: | :--------------: | :--------------: |
| GPT-5            | Claude-Sonnet-4  | Gemini-2.5-Pro   |
| DeepSeek-R1      | Qwen3-Coder-30B-A3B | **WebGen-R1-7B (Ours)** |

Table 16: Model comparisons with two tasks sourced from WebDev Arena.

**Instruction:** A battle arena website that compare audio mp3 generate by 2 models, model A and model B. Users listen to these two audio files and vote for the best.

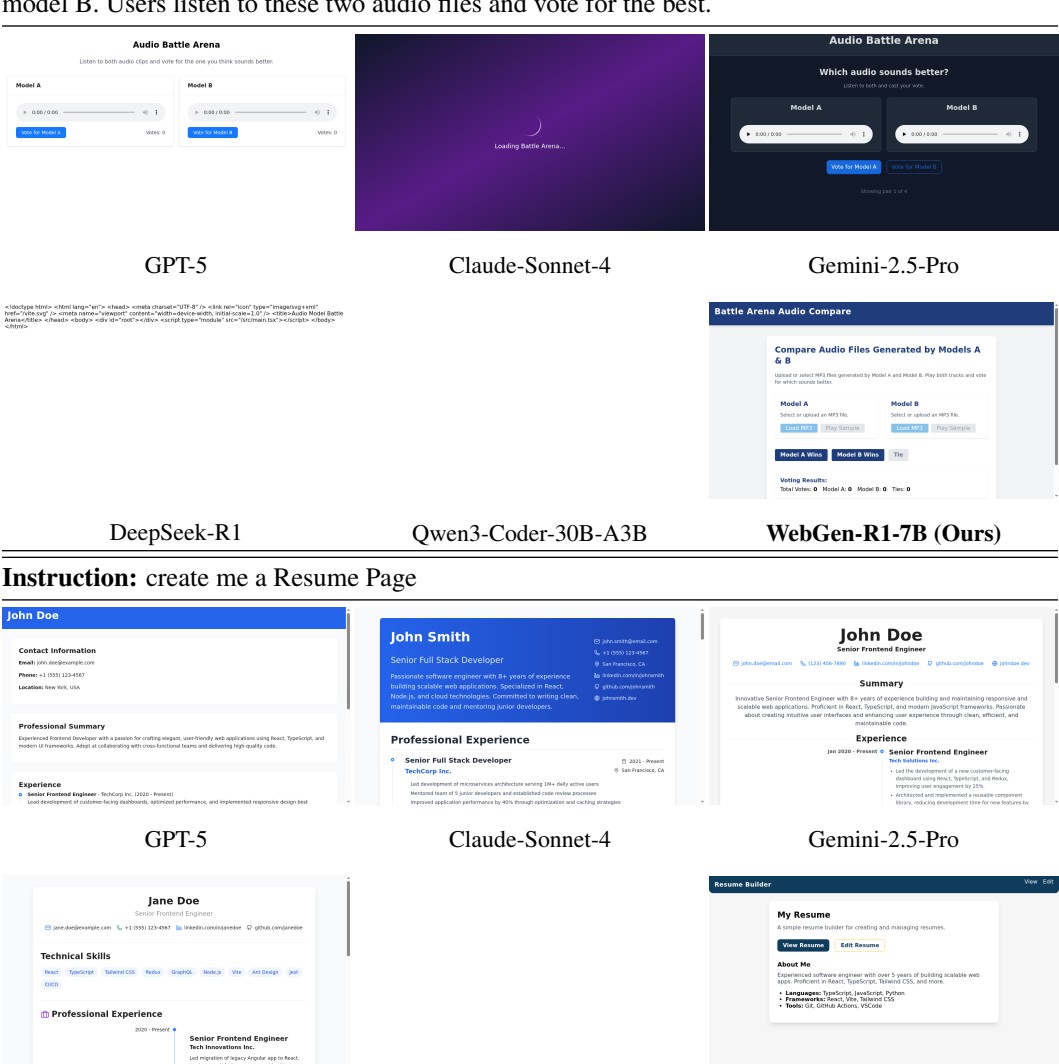

Table 17: Model comparisons with two tasks sourced from WebDev Arena.

**Instruction:** make me an app that generates content and images for html newsletters. it maintain inline css non spammy content, the theme and links should be given by the user.

| GPT-5 | Claude-Sonnet-4 | Gemini-2.5-Pro |
| --- | --- | --- |

| DeepSeek-R1 | Qwen3-Coder-30B-A3B | **WebGen-R1-7B (Ours)** |
| --- | --- | --- |

**Instruction:** Generate web app for storing custom recipes.

| GPT-5 | Claude-Sonnet-4 | Gemini-2.5-Pro |
| --- | --- | --- |

| DeepSeek-R1 | Qwen3-Coder-30B-A3B | **WebGen-R1-7B (Ours)** |
| --- | --- | --- |

Table 18: Model comparisons with two tasks sourced from WebDev Arena.

**Instruction:** Design a guided meditation player with progress tracking. Create a library view to browse and select different meditation sessions.

| GPT-5 | Claude-Sonnet-4 | Gemini-2.5-Pro |
| --- | --- | --- |

| DeepSeek-R1 | Qwen3-Coder-30B-A3B | **WebGen-R1-7B (Ours)** |
| --- | --- | --- |

**Instruction:** Build a linktree website for a singer-songwriter.

| GPT-5 | Claude-Sonnet-4 | Gemini-2.5-Pro |
| --- | --- | --- |

| DeepSeek-R1 | Qwen3-Coder-30B-A3B | **WebGen-R1-7B (Ours)** |
| --- | --- | --- |

Table 19: Model comparisons with two tasks sourced from WebDev Arena.

**Instruction:** Create me a time traker of multiple tasks that can be added or removed and have individual counter that cand be stoped or paused.

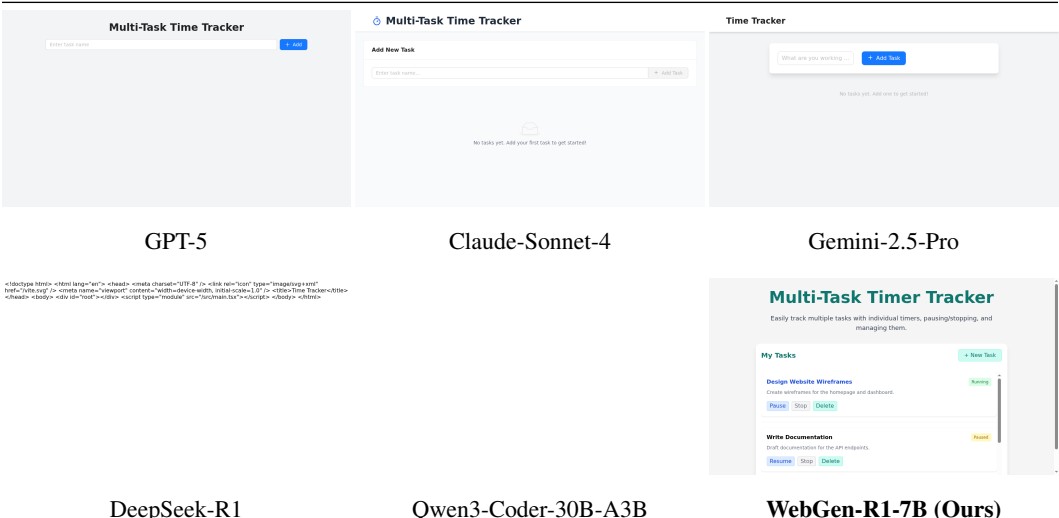

| GPT-5 | Claude-Sonnet-4 | Gemini-2.5-Pro |
|---|---|---|
| DeepSeek-R1 | Qwen3-Coder-30B-A3B | **WebGen-R1-7B (Ours)** |

**Instruction:** Design a cryptocurrency tracker with real-time price updates and portfolio tracking. Focus on clear presentation of price charts and user-friendly transaction inputs.

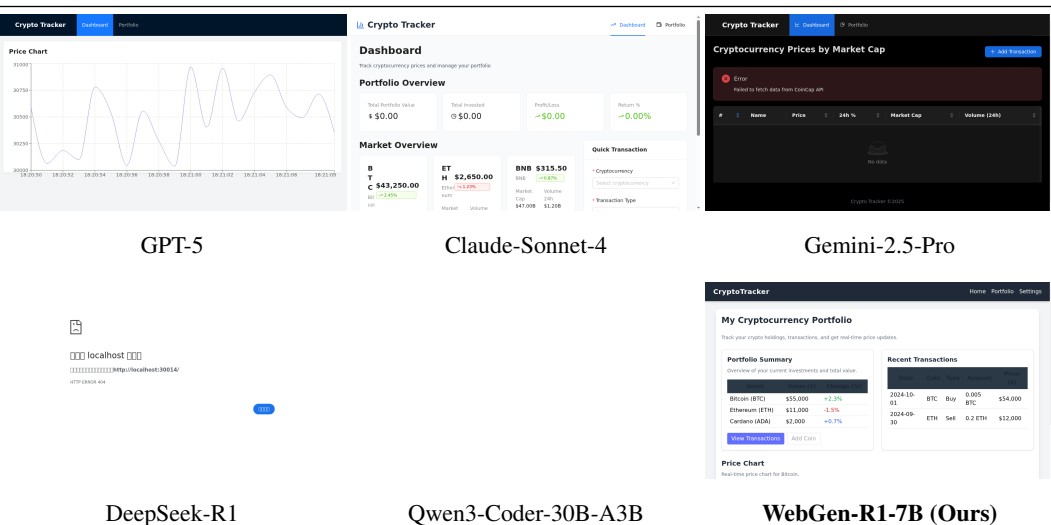

| GPT-5 | Claude-Sonnet-4 | Gemini-2.5-Pro |
|---|---|---|
| DeepSeek-R1 | Qwen3-Coder-30B-A3B | **WebGen-R1-7B (Ours)** |

Table 20: Model comparisons with two tasks sourced from WebDev Arena.

**Instruction:** Design a page to display a book review, including elements for ratings and user comments. Create an interface for browsing book reviews by category or popularity.

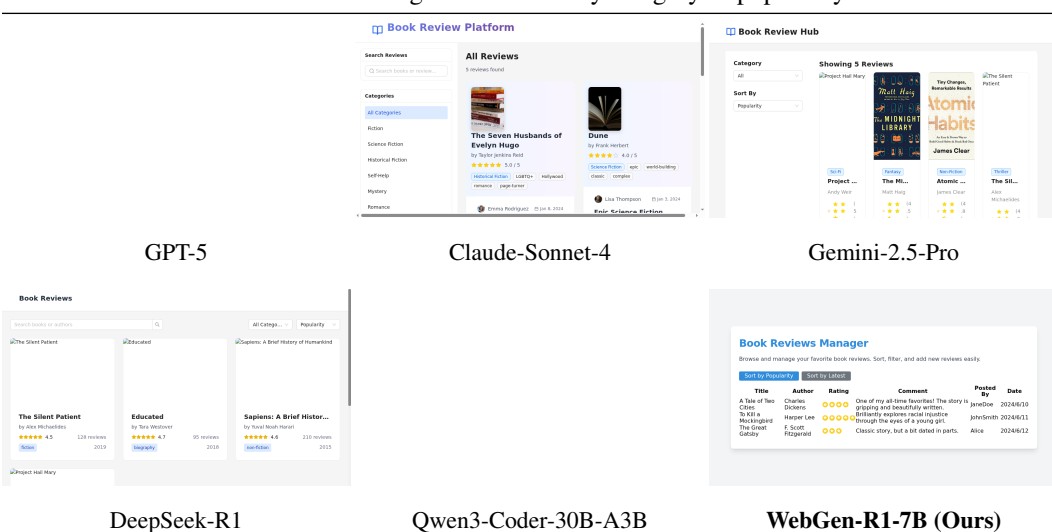

| GPT-5 | Claude-Sonnet-4 | Gemini-2.5-Pro |
| DeepSeek-R1 | Qwen3-Coder-30B-A3B | **WebGen-R1-7B (Ours)** |

**Instruction:** Fancy text generator with realtime preview without loading page, 10 stylish fonts, navigation, footer section with about us section, responsive.

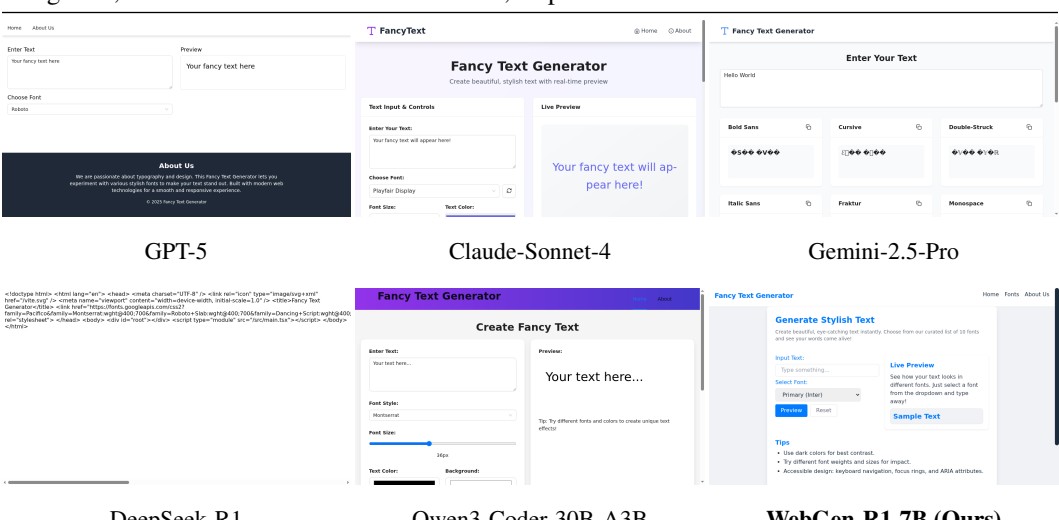

| GPT-5 | Claude-Sonnet-4 | Gemini-2.5-Pro |
| DeepSeek-R1 | Qwen3-Coder-30B-A3B | **WebGen-R1-7B (Ours)** |

Table 21: Model comparisons with two tasks sourced from WebDev Arena.

**Instruction:** Design a job board with filters for location, salary, and job type. Create an appealing layout for job postings, highlighting key details. with all of stuff.

|  |  |  |
|---|---|---|
| GPT-5 | Claude-Sonnet-4 | Gemini-2.5-Pro |

|  |  |  |
|---|---|---|
| Qwen3-32B | Qwen3-Coder-30B-A3B | **WebGen-R1-7B (Ours)** |

**Instruction:** clone of warrior cat website

|  |  |  |
|---|---|---|
| GPT-5 | Claude-Sonnet-4 | Gemini-2.5-Pro |

|  |  |  |
|---|---|---|
| Qwen3-32B | Qwen3-Coder-30B-A3B | **WebGen-R1-7B (Ours)** |

Table 22: Model comparisons with two tasks sourced from WebDev Arena.

**Task:** Verify that the website uses 'honeydew' as the background color and 'dark olive green' as the component color, as specified in the design requirements.

**Expected Result:** The website has a background color of 'honeydew' and components (such as buttons, cards, headers, etc.) are styled with the color 'dark olive green', accurately reflecting the design instruction.

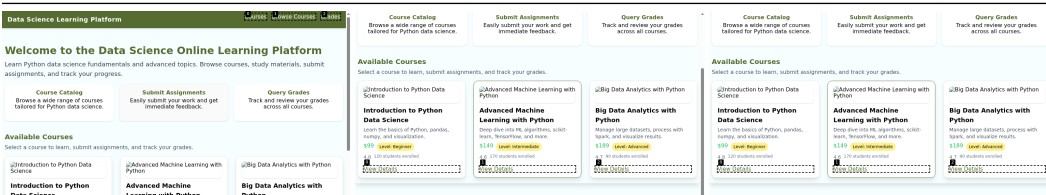

**Thought:** The initial webpage appears to have a "honeydew" background color and components (like headings and buttons) styled in "dark olive green." However, I need to verify these colors across different pages or sections of the website, if possible.

**Action:** Scroll [WINDOW]; down

**Thought:** The background color still appears to be "honeydew," and the components like course cards, headings, and buttons seem styled with "dark olive green." It aligns with the specified design so far, but further verification might involve checking other sections or interactions within the website.

**Action:** Scroll [WINDOW]; down

**Thought:** The page didn't change after scrolling down. It seems we have reached the bottom of this section. The background color and component color still align with the design requirement. Additional steps to verify more sections of the site may not be necessary. I may now conclude the evaluation.

**Action:** ANSWER; YES

**Task:** Verify that the website uses 'light goldenrod' as the background color and 'olive drab' as the component color, as specified in the design requirements.

**Expected Result:** The website has a background color of 'light goldenrod' and components (such as buttons, cards, headers, etc.) are styled with the color 'olive drab', accurately reflecting the design instruction.

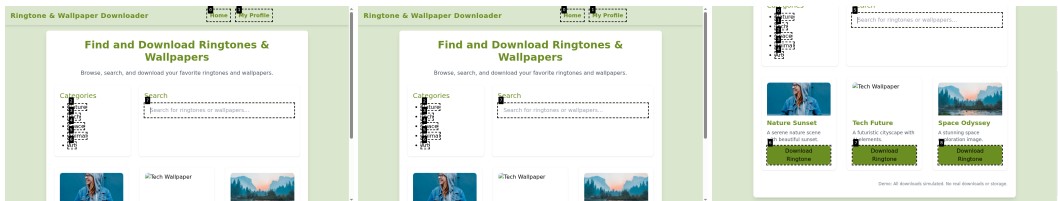

**Thought:** The webpage uses a light background color, which could be 'light goldenrod', and components such as text and buttons appear in a greenish color which might correspond to 'olive drab'. To validate this visually, I'll analyze specific webpage elements further.

**Action:** Wait

**Thought:** The screenshot seems to confirm that the background color of the webpage aligns with a shade close to 'light goldenrod,' and the component color (like heading, buttons) appears to resemble 'olive drab.' It is visually consistent, but further interaction or verification might still be required.

**Action:** Scroll [WINDOW]; down

**Thought:** The buttons labeled "Download Ringtone" have a greenish color that matches the description of 'olive drab,' and the background still aligns with 'light goldenrod.' All displayed elements comply visually with the task's design requirements. Further verification may not be necessary.

**Action:** ANSWER; YES

Table 23: UI agent testing processes resulting in YES.

**Task:** Verify that the website uses 'light gray' as the background color and 'dark red' as the component color, as specified in the design requirements.

**Expected Result:** The website has a background color of 'light gray' and components (such as buttons, cards, headers, etc.) are styled with the color 'dark red', accurately reflecting the design instruction.

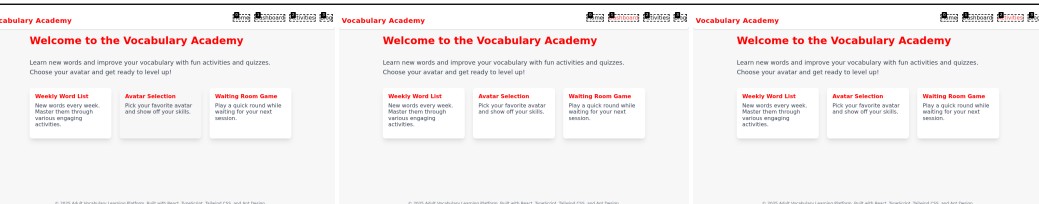

**Thought:** The background of the webpage appears to be light gray, and some components (like headings and texts on cards) seem to use the color dark red. I need to further explore the interface to confirm the design requirement fully, such as validating all component styles and ensuring accuracy.

**Action:** Click [1] (Dashboard) to explore further information about the webpage style and check other components.

**Thought:** The webpage remains unchanged, and the "Dashboard" section didn't open after the previous click. I need to attempt another navigation based on other options available to ensure I can explore the webpage for additional validations.

**Action:** Click [2] (Activities) to explore the "Activities" section.

**Thought:** The webpage still remains unchanged after multiple navigation attempts. It might be possible that the clickable options are not leading to other sections. Based on the observation seen so far, I have to make a final assessment.

**Action:** ANSWER; PARTIAL

---

**Task:** Validate the ability to view historical air quality index (AQI) data for the past two weeks.

**Expected Result:** Users should be able to access a time series or historical graph/chart that shows AQI data for each of the last 14 days.

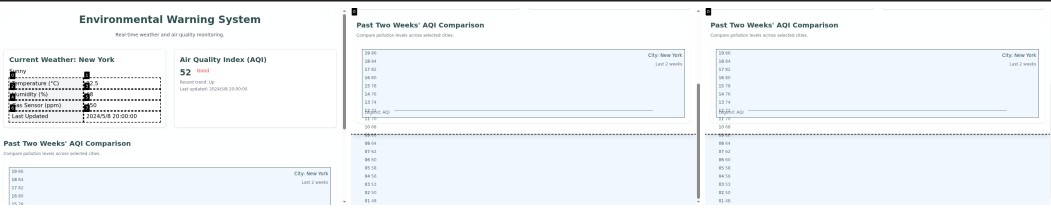

**Thought:** The webpage includes a section labeled "Past Two Weeks' AQI Comparison" which appears to list AQI data, but further details or a time series graph/chart showing the AQI data for each of the last 14 days is not immediately visible. The list may require scrolling to view more information.

**Action:** Scroll [WINDOW]; down

**Thought:** The "Past Two Weeks' AQI Comparison" section showcases numerical AQI data for the last 14 days, which matches the expected result in terms of tabular data representation. However, a time series graph/chart is still not visible. To ensure, I should continue scrolling for additional views or graphical representations if present.

**Action:** Scroll [WINDOW]; down

**Thought:** After scrolling further, no additional graphical representation like a time series chart for the last 14 days is visible. Only numerical AQI data in a tabular fashion is presented, which fulfills part of the task but not entirely.

**Action:** ANSWER; PARTIAL

Table 24: UI agent testing processes resulting in PARTIAL.

**Task:** Confirm navigation from the home page to the 'About Us' page.

**Expected Result:** Clicking the 'About Us' link takes the user to the 'About Us' page without any errors.

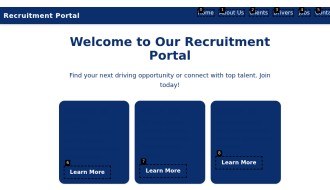 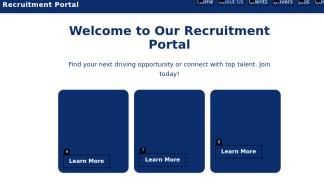 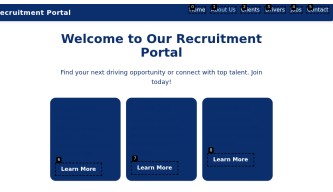

| | | |
|---|---|---|
| **Thought:** The webpage labeled elements include navigation links such as "About Us." To confirm the navigation from the Home page to the "About Us" page, I'll click the "About Us" link (Label [1]). **Action:** Click [1]. | **Thought:** The screen remains on the same page as before, suggesting the "About Us" link might not be functional. I will attempt to click it again to ensure it wasn't a one-time issue. **Action:** Click [1]. | **Thought:** The screen still remains unchanged after clicking the "About Us" link twice, confirming that the link is non-functional or does not navigate properly. Now, I will answer based on the observations. **Action:** ANSWER; NO |

**Task:** Test the supplier management interface for adding or updating a golf package supplier.

**Expected Result:** The user should be able to successfully add or update supplier details with confirmation that changes have been saved in the system.

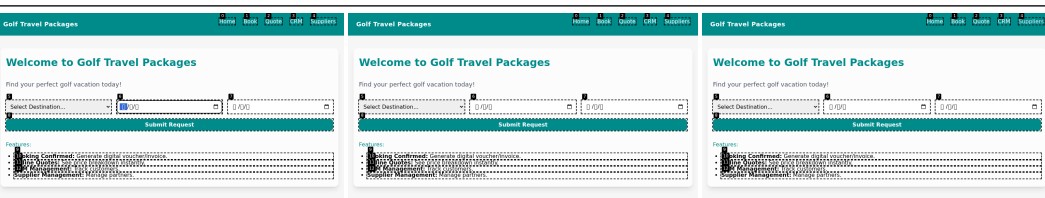

| | | |
|---|---|---|
| **Thought:** To manage suppliers, the "Suppliers" navigation option (label [4]) seems appropriate. This should provide access to the supplier management interface. **Action:** Click [4]. | **Thought:** The page remains unchanged, and no new supplier management interface is visible after the previous action. It seems the "Suppliers" option ([4]) did not navigate as expected. I will try interacting with it again to ensure the action registers. **Action:** Click [4]. | **Thought:** Even after performing the action twice, the page remains unchanged, and the "Suppliers" button ([4]) does not navigate to or display the expected supplier management interface. I suspect it might not be functioning or implemented correctly. I cannot proceed further on this page. **Action:** ANSWER; NO |

Table 25: UI agent testing processes resulting in NO.

