# OpenReview forum: "WebGen-R1: Incentivizing LLMs to Generate Functional and Aesthetic Websites with Reinforcement Learning"
_ICLR.cc/2026/Conference — Submitted to ICLR 2026_

### Official Review · Reviewer_vGHH · 2025-10-29

**Soundness:** 3
**Presentation:** 3
**Contribution:** 2
**Rating:** 4
**Confidence:** 3

**Summary:**

This paper proposes WebGen-R1, which casts end-to-end website generation into RL framework. It integrates a VLM reward that jointly evaluates both functional correctness and visual aesthetics based on the screenshots of the rendered website. The agent model is fine-tuned using GRPO with the VLM reward. The experiment results demonstrate that WebGen-R1 significantly improves functional success rate, render reliability, and aesthetic quality, outperforming or matching larger proprietary models such as GPT-5, Claude, and Gemini.

**Strengths:**

1. The idea of incorporating a VLM-based reward model for website generation is straightforward and practically valuable.

2. The proposed method achieves superior performance compared to existing general-purpose LLM agents

3. The paper is clearly structured and well written

**Weaknesses:**

1. The VLM-based reward is limited to static screenshots, which cannot fully assess the functional correctness of interactive elements such as dropdown menus, buttons, or dynamic state transitions.

2. Most baseline models are general-purpose LLMs rather than website-specific agents. The paper lacks evaluation against alternative fine-tuning strategies tailored to website generation, such as training with a human-preference reward model or other reward models.

3. The technical depth is relatively limited. The proposed method mainly adapts existing RL methods to the website generation domain without new algorithmic components.

**Questions:**

see Weaknesses

---

> ### Author Response · Authors · 2025-11-25
> **Rebuttal (Part 1/2)**
>
> Thank you for your comprehensive and insightful review of our paper. We appreciate your valuable feedback and suggestions.
>
> > The VLM-based reward is limited to static screenshots, which cannot fully assess the functional correctness of interactive elements such as dropdown menus, buttons, or dynamic state transitions.
>
>
> We acknowledge the limitations of VLM-based reward. Despite the VLM-based reward model primarily evaluating functional aspects through visual indicators (e.g., presence of forms, dropdown menus, search inputs) - as these UI elements typically convey intended functionalities - the limited capacity of the small-scale model (7B) hinders the accurate implementation of complex logic and interactions behind the UI elements.
>
> We considered GUI-agent-based judges `[1]`, but found them not well-suited for large-scale RL training due to: (1) **Limited reliability**, as their evaluation accuracy is constrained by agent capabilities; (2) **High computational and operational cost**, requiring significant environment scaling; (3) **Slow execution resulting from multi-turn interactions**, making large-scale training inefficient; (4) Importantly, WebGen-Instruct training dataset **lacks standardized test cases**, which are essential for the GUI Agent to verify specific functionalities or appearances through predefined operations and expected outcomes, thereby preventing its use as the functional verifier during RL training.
>
> **Given these constraints, we adopt VLM-based reward as a tractable and scalable first step**. A detailed comparison between VLM and GUI-agent evaluation for functional correctness is provided in Table below.
>
>
> Table 1: Comparison of GUI-agent evaluation and VLM-based reward from static screenshots for functional correctness.
>
> | Aspect | GUI-Agent Evaluation | VLM-based Reward |
> | :--- | :--- | :--- |
> | Functional correctness | Interact with UI elements | Infers functionality from static screenshots using visual cues |
> | **GUI Interaction Instructions** | **Hard to obtain and expensive (Not available in WebGen‑Instruct)** | **Not applicable** |
> | Interaction requirement | Requires full browser, DOM access, multi-step actions | No interaction needed |
> | **Reward cost** | **Very high    (~51× slower; avg. 51.2 multi-turns interactions per task; 614.91s per sample)** | **Very low    (single-turn per task; 12.01s per sample)** |
> | Environment stability | Fragile    (affected by page load, network, DOM changes) | Fully stable    (deterministic and self-contained) |
> | Scalability for RL training | Poor    (prohibitively expensive for millions of rollouts) | Excellent    (high-throughput, scalable to large datasets) |
> | Coverage of UI properties | Strong for interaction correctness | Strong for layout, visual structure, and element presence |
> | Best use cases | Final evaluation, small-scale precise testing | RL training stage, large-scale sample collection |
> | Main limitations | Slow, costly, unstable | Cannot fully verify true interactive functionality |

---

> ### Author Response · Authors · 2025-11-25
> **Rebuttal (Part 2/2)**
>
> > Most baseline models are general-purpose LLMs rather than website-specific agents. The paper lacks evaluation against alternative fine-tuning strategies tailored to website generation, such as training with a human-preference reward model or other reward models.
>
>
> Our work aims to explore end-to-end generation of entire multi-page websites using LLMs, without task decomposition or multi-turn interaction. This problem formulation and pipeline are fundamentally different from website-specific modular agents, making direct comparisons potentially unfair or orthogonal to our setting.
>
> To address the reviewer's concern, we conducted additional experiments with fine-tuning strategies explicitly tailored for website generation:
>
> - **WebGen-R1-7B (SFT)**: Starting from Qwen2.5-Coder-7B-Instruct, we perform supervised fine-tuning on 600 high-quality GPT-4.1-generated website data.
> - **WebGen-R1-7B (HPRM)**: Following the standard reward model (RM) training pipeline `[2]`, we trained a RM using Qwen3-VL-8B-Instruct on the same 600 tasks. To reduce annotation cost, we automatically construct preference pairs:
>     * Positive samples from GPT-4.1 (FSR = 43.91, AAS = 3.78)
>     * Negative samples from Qwen3-8B (FSR = 3.72, AAS = 2.57)
>
> The RM is fine-tuned using the LLaMA-Factory `[3]` framework and then used as the evaluator in RL training of WebGen-R1. The results are shown in Table below.
>
> | Model | FSR (%) | AAS | VRR (%) |
> | --- | ---: | ---: | ---: |
> | Baseline | 1.59 | 2.73 | 30.56 |
> | WebGen-R1-7B (SFT) | 20.08 | 3.24 | 30.69 |
> | WebGen-R1-7B (HPRM) | 8.64 | 2.95 | 48.23 |
> | WebGen-R1-7B (Ours) | 29.21 | 3.94 | 95.89 |
>
> Results show that WebGen-R1-7B (SFT) substantially improves over the baseline. Unexpectedly, WebGen-R1-7B (HPRM) underperforms compared to SFT. Reward curves exhibit an initial increase followed by a sharp drop, leading to model collapse. We attribute this to RM limitations caused by:
>
> 1. Small preference dataset size due to cost/time constraints.
> 2. Limited task diversity in RM training data, reducing its ability to reliably evaluate complex or out-of-distribution website tasks.
>
> In contrast, large general-purpose LLMs exhibit strong instruction-following and generalization abilities, allowing them to align more consistently with our evaluation criteria and produce more reliable reward signals for this open-ended generation setting.
>
>
>
>
>
> > The technical depth is relatively limited. The proposed method mainly adapts existing RL methods to the website generation domain without new algorithmic components.
>
> We would like to emphasize the technical contribution of our work:
> * It presents **the first exploration of a new problem formulation** for end-to-end generation of multi-page websites and **project-level** code optimization via RL.
> * As noted by Reviewer aoRQ, **this is a highly challenging task and represents an ambitious and impactful research direction for MLLMs**.
>
>
> From the perspectives of technological implementation and methodology, our work addresses the following key challenges:
>
> 1. **Stable large-scale RL integration with front-end build pipelines** - incorporating a standardized build system into RL training while ensuring stability, efficiency, and scalability.
> 2. **Reward design aligned with website quality** - constructing an RL reward that correlates with actual improvements in both functionality and visual quality, especially in subjective and open-ended domains.
> 3. **Joint functional–visual evaluation via VLMs** - assessing functional correctness and aesthetic design from the visual rendering of websites, alongside code formatting and reasoning structural constraints, to mitigate reward sparsity, reward hacking, and model collapse.
> 4. **Multi-dimensional evaluation metrics** - introducing FSR, AAS, and VRR to comprehensively quantify LLM performance in end-to-end multi-page website generation.
>
>
>
> ### Reference
> `[1]` Lin, Kevin Qinghong, et al. "Computer-Use Agents as Judges for Generative User Interface." arXiv preprint arXiv:2511.15567 (2025).
>
> `[2]` Ouyang, Long, et al. "Training language models to follow instructions with human feedback." NeurIPS, 2022.
>
> `[3]` Zheng, Yaowei, et al. "LlamaFactory: Unified Efficient Fine-Tuning of 100+ Language Models." ACL. 2024.

---

### Official Review · Reviewer_vk6b · 2025-10-30

**Soundness:** 3
**Presentation:** 3
**Contribution:** 3
**Rating:** 6
**Confidence:** 4

**Summary:**

The paper introduces WebGen-R1, a framework designed to train a small-scale Large Language Model (LLM) to generate complete, multi-page websites from natural language prompts in an end-to-end fashion. It presents a novel reinforcement learning (RL) approach that uses a Vision-Language Model (VLM) as a reward function. This VLM assesses rendered screenshots of the generated website to provide a unified reward signal for both functional correctness and aesthetic quality, overcoming the limitations of brittle, rule-based reward systems.
The methodology involves generating the entire project codebase as a single sequence, which is then processed by an automated pipeline that builds, renders, and evaluates the website. The resulting reward, which also incorporates signals for code structure and chain-of-thought reasoning, is used to fine-tune the base LLM using the Group Relative Policy Optimization (GRPO) algorithm. The evaluation uses a variety of both rule as well as LLM-based metrics, as well as human evaluation to mitigate reward hacking and performs similarly or better than the latest SOTA closed and open models.

**Strengths:**

- Rigorous pipeline and evaluation - the paper is a good improvement on existing works, combining an end-to-end pipeline, critical rules based evaluation steps, and also testing for human alignment, reporting remarkably high Pearson’s correlation coefficient r = 0.903
and Spearman’s rank correlation ρ = 0.888. This, together with the qualitative examples, strengthens the validity of the metrics actually measuring across the intended dimensions.

- Strong Empirical Results on Key Metrics: The dramatic improvements in Aesthetic Alignment Score (+44.32%) and Valid Render Ratio (+65.33%) are impressive.

- Multiple datasets, good generalizability to WebDev Arena benchmark.

**Weaknesses:**

- Dependence of a large proprietary model for feedback
- The FSR of 29.21% is low in absolute terms and significantly lower the 57.72% achieved by Claude-3.7-Sonnet. This could be further explored.
- Small evaluation - while decently sized compared to the previous works, the evaluation sample is still a relatively small sample and might not approximate real workflows well,

**Questions:**

- Could you provide an ablation study on the reward function? Specifically, what is the impact on final performance if you remove the code format reward or the reasoning reward?
- What were some of the more complex samples? while informative, the qualitative study demonstrates fairly basic setups.
- Could you share more details about human study, how did human to score correlation compare to inter inter-human one?

---

> ### Author Response · Authors · 2025-11-25
> **Rebuttal (Part 1/3)**
>
> Thank you for your comprehensive and insightful review of our paper. We greatly appreciate your recognition of the strengths of our work, such as rigorous pipeline and evaluation, strong empirical results, and good generalizability.
>
> > Dependence of a large proprietary model for feedback
>
> Our choice of a large proprietary model as the feedback evaluator reflects a trade-off among evaluation accuracy, latency, and cost. Smaller or weaker models are more vulnerable to reward signal degradation (e.g., reward hacking or noisy scoring), which can impair overall training performance.
>
>
> Empirically, the performance of open-source VLMs correlates with inference cost (including cloud rental), and their latency is constrained by available computational resources. In contrast, large proprietary models are deployed on highly optimized clusters, typically yielding stronger evaluation quality, though with API-based cost proportional to usage. Thus, following the WebGen-Bench `[1]` protocol, we selected GPT-4o given its strong empirical accuracy and relatively low API cost.
>
> To address the reviewer’s concern, we additionally evaluated two alternative VLMs: the open-source Qwen2.5-VL-72B-Instruct deployed on a rented cloud server with 8x NVIDIA HGX H100 SXM (USD $23.92/h), and the more advanced proprietary GPT-4.1 via API. The comparative results, shown in Table. 1 below, indicate that using open-source models for feedback may lead to inferior final task performance and, in some cases, higher overall cost due to infrastructure requirements.
>
> Table 1. Performance comparison of alternative VLM-based reward models.
>
> | VLM Evaluator | FSR (%) | AAS | VRR (%) | Inference Speed / sample | Prompt Tokens | Completion Tokens | Cost / sample |
> | :---: | :---: | :---: | :---: | :---: | :---: | :---: | :---: |
> | Qwen2.5-VL-72b-Instruct | 10.32 | 3.18 | 27.72 | 6.47s | 1344 | 332.8 | $0.036 USD |
> | GPT-4.1 | 24.58 | 3.36 | 91.04 | 11.57s | 1190 | 389 | $0.005 USD |
> | GPT-4o | 29.21  | 3.94  | 95.89 | 12.01s | 1190 | 437 | $0.007 USD |
>
>
>
>
> > The FSR of 29.21% is low in absolute terms and significantly lower the 57.72% achieved by Claude-3.7-Sonnet. This could be further explored.
>
> Thank you for your insightful comment. The lower FSR performance of our model compared to Claude-3.7-Sonnet is mainly due to:
>
> 1. Our backbone is a relatively small 7B parameter model, whereas Claude-3.7-Sonnet is substantially larger. According to scaling laws `[2]`, smaller models have reduced capacity for reasoning about the complex logic and interactivity underlying UI components.
> 2. The VLM-based reward model evaluates functional aspects primarily through visual indicators (e.g., presence of forms, dropdown menus, search inputs), as these UI elements typically convey intended functionalities. While effective to some extent, the limited capacity of the 7B model constrains accurate implementation of complex logic and interactions behind the UI elements, resulting in lower FSR compared to a much larger model.

---

> ### Author Response · Authors · 2025-11-25
> **Rebuttal (Part 2/3)**
>
> > Small evaluation - while decently sized compared to the previous works, the evaluation sample is still a relatively small sample and might not approximate real workflows well.
>
> We agree that constructing larger-scale evaluation datasets is important for comprehensive assessment. However, this process entails substantial resource costs, particularly in terms of human annotation and time. At present, WebGen-Bench and WebDev-Arena remain the most widely used and high-quality website benchmarks in the open-source community.
>
> To address the limited availability of large-scale open-source benchmarks, we emphasize the importance of assessing model generalization and robustness to unseen website tasks.
>
> In our manuscript, we evaluate on the out-of-distribution (OOD) WebDev-Arena dataset and find that our WebGen-R1 consistently outperforms strong proprietary and open-source baselines (e.g., DeepSeek-R1, GPT-5, and Qwen3-32B), as shown in Figure 6. This demonstrates that our WebGen-R1 retains strong applicability and robustness in practical, previously unseen web development scenarios.
>
> Therefore, even though the current evaluation set is not large in scale, the observed results provide solid evidence that our method generalizes effectively to a wider range of website generation tasks.
>
>
>
>
>
> > Could you provide an ablation study on the reward function? Specifically, what is the impact on final performance if you remove the code format reward or the reasoning reward?
>
>
> Thank you for your constructive suggestions. As noted in Appendix E.4 and Figure 14 (a), we analyzed the effect of varying the weighting coefficients $ \gamma $ (code-format reward) and $ \lambda $ (reasoning reward) on reward dynamics and optimization stability. The final performance on AAS (VLM-based evaluation) and FSR (WebVoyager agent) was not reported initially due to the high evaluation cost.
>
> Following the reviewer's suggestion, we conducted an ablation study across different $ \gamma $ and $ \lambda $ settings, including complete removal of the code-format reward ($ \gamma = 0.0 $) and reasoning reward ($ \lambda = 0.0 $).
>
> | Hyperparameters | FSR (%) | AAS | VRR (%) |
> | --- | --- | --- | --- |
> | $ \gamma=0.0 $, $ \lambda=0.0 $ | 25.89 | 3.70 | 94.01 |
> | $ \gamma=0.0 $, $ \lambda=0.1 $ | 28.54 | 3.92 | 95.00 |
> | $ \gamma=0.0 $, $ \lambda=1.0 $ | 26.13 | 3.87 | 95.00 |
> | $ \gamma=0.1 $, $ \lambda=0.0 $ | 24.08 | 3.66 | 93.02 |
> | $ \gamma=0.1 $, $ \lambda=0.1 $ | **29.21** | **3.94** | **95.89** |
> | $ \gamma=0.1 $, $ \lambda=1.0 $ | 28.39 | 3.92 | 94.01 |
> | $ \gamma=1.0 $, $ \lambda=0.0 $ | 22.75 | 3.34 | 91.04 |
> | $ \gamma=1.0 $, $ \lambda=0.1 $ | 23.06 | 3.47 | 91.09 |
> | $ \gamma=0.5 $, $ \lambda=0.5 $ | 26.12 | 3.77 | 93.17 |
> | $ \gamma=1.0 $, $ \lambda=1.0 $ | 25.94 | 3.78 | 93.02 |
>
> Results show that removing the code-format reward ($ \gamma = 0.0 $, $ \lambda = 0.1 $) causes only minor drops in FSR (−0.67), AAS (−0.02), and VRR (−0.89), indicating that rewards aligned with functionality and visual quality implicitly enforce code formatting consistency. In contrast, removing the reasoning reward ($ \gamma = 0.1 $, $ \lambda = 0.0 $) leads to substantial decreases in FSR (−5.13) and VRR (−2.87), along with a noticeable reduction in AAS (−0.28). This suggests that the reasoning reward plays a more critical role in achieving higher task success rates and overall reliability.

---

> ### Author Response · Authors · 2025-11-25
> **Rebuttal (Part 3/3)**
>
> > What were some of the more complex samples? while informative, the qualitative study demonstrates fairly basic setups.
>
> The example in the main text is intended solely for clarity of presentation. The full WebGen-Bench encompasses a broad spectrum of technical complexity, including highly demanding tasks such as real-time collaborative tools (20%), secure e-commerce checkout flows (22%), AI-powered interactive features (19%), and large-scale data visualization with performance optimization (36%). These tasks require multi-step reasoning, API integration, state management, and responsive UI design. More complex examples are provided in Appendix H.1 and H.2 due to space constraints.
>
> We select a system-level website task to illustrate the inherent complexity and the challenges faced by LLMs in completing such tasks. Achieving end-to-end implementation of this case with a single LLM demands advanced front-end engineering skills and integrated reasoning capabilities.
>
>
> Table 1: Example system-level website task from WebGen-Bench, demonstrating the breadth of technical requirements involved.
>
> | ID | Task Title | Instruction | Technical Complexity Profile |
> | --- | --- | --- | --- |
> | 000047 | E-Government Comprehensive Management System | Please implement an e-government system for office management, infrastructure management, procurement management, personnel management, financial management, and decision-making management. The system should have functionalities for office management, infrastructure management, procurement management, personnel management, financial management, and decision-making management. Users should be able to browse related information, submit applications, approve, query status, and perform related operations. The system should also include functions such as leave, middle-level cadres leaving Shenzhen, attendance, and "five powers" management, which can design and circulate corresponding forms and processes. Implement light pink for the page background and medium violet red for the elements. | This website task involves building a single platform that integrates office, infrastructure, procurement, personnel, financial, and decision-making management. Each module has highly distinct workflows, including application submission, approval processes, querying, and form circulation. The front-end must support dynamic form generation, complex routing across multiple submodules, and a permissions framework with variable access levels depending on role and task. The diversity of business logic, strong inter-module dependencies, and need for a uniform yet flexible interface make this task extremely challenging. |
>
>
> > Could you share more details about human study, how did human to score correlation compare to inter inter-human one?
>
> We provide additional details regarding the human evaluation protocol and the comparison between model–human and inter-human correlations.
>
> Our human evaluation follows the WebGen-Bench protocol `[1]`. Three experienced front-end developers independently rated 101 model-generated websites on functionality and visual appeal, using the same criteria and scale as our reward model (see Appendix G.2).
>
> To assess inter-human agreement, we computed the average pairwise correlation among the three annotators: Pearson $r = 0.714$, Spearman $\rho = 0.695$. For cases with notable disagreement, a fourth annotator re-examined the sample and provided the adjudicated score, consistent with WebGen-Bench's procedure. We then compared the aggregated human scores with those from our reward model. As reported, the model–human correlations are strong (Pearson $r = 0.762$, Spearman $\rho = 0.734$).
>
>
> We note that the model-human correlation being slightly higher than the inter-human correlation is reasonable for this task. Human annotators naturally exhibit subjective variability, especially in aesthetic judgments. In contrast, VLM-based reward model is aligned to the aggregated human consensus rather than any individual annotator, making its judgments more consistent with the averaged human signal. This phenomenon is common in evaluation tasks where the model captures a stable consensus while individual annotators may deviate due to personal preferences. Importantly, the result does not imply that the model "outperforms" humans, but rather that it aligns well with the shared human standard.
>
> ### Reference
> `[1]` Lu, Zimu, et al. "WebGen-Bench: Evaluating LLMs on Generating Interactive and Functional Websites from Scratch." NeurIPS, 2025.
>
> `[2]` Kaplan, Jared, et al. "Scaling laws for neural language models." arXiv preprint arXiv:2001.08361 (2020).

---

### Official Review · Reviewer_aoRQ · 2025-10-31

**Soundness:** 2
**Presentation:** 2
**Contribution:** 3
**Rating:** 4
**Confidence:** 4

**Summary:**

The authors present WebGen-R1, an MLLM trained with GRPO using feedback signals from a reward signal generated from a model (GPT-4o) and also some rule-based signals to evaluate the visual and rendering quality of generated website code. The model, WebGen-R1-7B, is compared against several state-of-the-art closed-source models and the Qwen family of models, outperforming all baselines on the aesthetic and rendering metric (AAS).

**Strengths:**

- The task of generating complete websites from scratch is highly challenging and represents an ambitious and impactful direction for MLLMs.

- The proposed reward function aligns well with human preferences, and the authors demonstrate this alignment effectively.

**Weaknesses:**

- It is unclear why WebGen-LM models are not included in the comparison. Including them would provide a stronger and more complete baseline for evaluating progress.

The interpretation of the AAS metric appears inaccurate. The authors claim that WebGen-R1 achieves superior performance “across all 13 categories on AAS, indicating consistent improvements in both functional correctness and UI/UX quality.” However, AAS is an aesthetic metric that primarily captures visual quality, as indicated by the system prompt in the appendix. Functional correctness should instead be measured by FSR. This confusion between aesthetic and functional metrics is repeated in several parts of the paper.

- Since the model is optimized to maximize the reward model’s score—80% of which comes from AAS under the configuration λ=0.1 and γ=0.1—there is a risk of overfitting to aesthetic quality. To demonstrate that the method works more generally, the model should also show improvements on FSR or other metrics beyond AAS.

- The paper does not provide results for larger γ values (γ = 0.5 or γ = 1.0). Showing AAS and FSR results under different γ (and also λ) values would help illustrate how varying the weight between aesthetic and functional and reasoning rewards affects performance.

- It would be helpful to identify which model serves as the most reliable judge or produces outputs that best align with intended outcomes. Benchmarks such as PairBench [1] or AgentRewardBench [2] could be used to evaluate this aspect.

[1] PairBench: Are Vision-Language Models Reliable at Comparing What They See?

[2] AgentRewardBench: Evaluating Automatic Evaluations of Web Agent Trajectories

**Questions:**

Please refer to the issues raised in the weaknesses.

---

> ### Author Response · Authors · 2025-11-25
> **Rebuttal (Part 1/2)**
>
> Thank you for your thorough and insightful review. We greatly appreciate your recognition of the contributions of our work as an ambitious and impactful direction for MLLMs.
>
> > It is unclear why WebGen-LM models are not included in the comparison. Including them would provide a stronger and more complete baseline for evaluating progress.
>
> Thank you for bring this to our attention. As noted in Section 3.1 of our manuscript, we excluded WebGen-LM from our comparison because it is specifically fine-tuned on website-generation trajectories collected within the Bolt.diy agent framework, using DeepSeek-V3 as the agent engine. This makes the model tightly coupled to that particular agent-based workflow and not directly applicable to our evaluation setting.
>
> Furthermore, our evaluation of WebGen-LM-7B on WebGen-Bench yielded a score of 0.0 on all three metrics. Manual inspection of its generated website source code revealed that it failed to satisfy the required project file organization, preventing successful parsing. We hypothesize that the reliance on predefined agent-workflow trajectories during training constrained its ability to generalize beyond that specific framework.
>
> > The interpretation of the AAS metric appears inaccurate. The authors claim that WebGen-R1 achieves superior performance “across all 13 categories on AAS, indicating consistent improvements in both functional correctness and UI/UX quality.” However, AAS is an aesthetic metric that primarily captures visual quality, as indicated by the system prompt in the appendix. Functional correctness should instead be measured by FSR. This confusion between aesthetic and functional metrics is repeated in several parts of the paper.
>
> We thank the reviewer for the valuable feedback. We clarify that the VLM-based AAS metric serves to evaluate **the alignment of the rendered webpage with user requirements and aesthetic preferences from a visual standpoint**. While primarily visual, the AAS can implicitly assess certain functional aspects via visual cues — for example, detecting the presence of forms, dropdown menus, or search inputs. In most rendered webpages, such UI components directly reflect underlying functional intent.
>
> For example, if the instruction specifies "support submitting questions, answering questions, and viewing answers," the VLM examines the rendered page to verify the presence of a question input field, a submission button, and an answer display interface.
>
> Empirically, we observe a strong correlation between visually inferred functionality and task success. As shown in Table 1 in our manuscript, improvements in the AAS are accompanied by a **+27.62% absolute increase in FSR**, indicating that higher visual alignment generally reflects better functional coverage in our setting.
>
> We have explicitly clarified in the revised manuscript that AAS is not a direct measure of functional correctness (as captured by FSR) but serves as a complementary proxy metric.
>
>
>
>
>
>
>
>
>
>
> > Since the model is optimized to maximize the reward model’s score—80% of which comes from AAS under the configuration λ=0.1 and γ=0.1—there is a risk of overfitting to aesthetic quality. To demonstrate that the method works more generally, the model should also show improvements on FSR or other metrics beyond AAS.
>
> Thank you for your valuable suggestions. Balancing reward weights across multiple objectives is intended to prevent simpler objectives from dominating the optimization process, which could hinder the learning of more challenging ones. This balance improves the stability of RL training `[1]`.
>
> As discussed in Appendix E.4, we evaluate the effect of different weight coefficients on the three reward objectives. In Figure 14, with $\lambda = 0.1$ and $\gamma = 0.1$, both $S_{\text{code}}$ and $S_{\text{cot}}$ converge to 1.0 after ~40 steps, indicating that the generated websites consistently meet correctness requirements for both code and reasoning format. This shows that our reward model does not overfit to the aesthetic objective.
>
> Our settings follow the recommended configuration for multi-objective rewards in **Open-R1** training framework. In **Verl** `[2]` training framework, the format reward weight is also set to 0.1 by default. To further assess the impact of varying $\gamma$ and $\lambda$ on FSR and VRR, we performed a sensitivity analysis, with results summarized in Table 1 below.

---

> ### Author Response · Authors · 2025-11-25
> **Rebuttal (Part 2/2)**
>
> > The paper does not provide results for larger $ \gamma $ values ($ \gamma $ = 0.5 or $ \gamma $ = 1.0). Showing AAS and FSR results under different $ \gamma $ (and also $ \lambda $) values would help illustrate how varying the weight between aesthetic and functional and reasoning rewards affects performance.
>
> Thank you for the suggestion to report AAS and FSR results for different $ \gamma $ and $ \lambda $ values. In Appendix E.4, we analyzed how varying these weighting coefficients affects reward dynamics and optimization stability (Figure 14a). We initially omitted AAS and FSR results due to the high evaluation cost of API calls. As per your suggestion, we have now evaluated WebGen-R1 on AAS, FSR, and VRR under larger $ \gamma $ and $ \lambda $ values.
>
> Table 1. Sensitivity analysis of $\gamma$ and $\lambda$ values with respect to performance across FSR, AAS, and VRR metrics.
>
> | Hyperparameters | FSR (%) | AAS | VRR (%) |
> | --- | --- | --- | --- |
> | $ \gamma=0.0 $, $ \lambda=0.0 $ | 25.89 | 3.70 | 94.01 |
> | $ \gamma=0.0 $, $ \lambda=0.1 $ | 28.54 | 3.92 | 95.00 |
> | $ \gamma=0.0 $, $ \lambda=1.0 $ | 26.13 | 3.87 | 95.00 |
> | $ \gamma=0.1 $, $ \lambda=0.0 $ | 24.08 | 3.66 | 93.02 |
> | $ \gamma=0.1 $, $ \lambda=0.1 $ | **29.21** | **3.94** | **95.89** |
> | $ \gamma=0.1 $, $ \lambda=1.0 $ | 28.39 | 3.92 | 94.01 |
> | $ \gamma=1.0 $, $ \lambda=0.0 $ | 22.75 | 3.34 | 91.04 |
> | $ \gamma=1.0 $, $ \lambda=0.1 $ | 23.06 | 3.47 | 91.09 |
> | $ \gamma=0.5 $, $ \lambda=0.5 $ | 26.12 | 3.77 | 93.17 |
> | $ \gamma=1.0 $, $ \lambda=1.0 $ | 25.94 | 3.78 | 93.02 |
>
>
> The results show that lower $ \gamma $ typically increases AAS and VRR but also leads to higher FSR. Setting $ \gamma = 1.0$ markedly reduces FSR when $ \lambda $ is small, with a slight rebound as $ \lambda $ grows. The effect of $ \lambda $ is more moderate, with mid-range values slightly boosting AAS and VRR. These patterns highlight how balancing aesthetic–functional rewards with reasoning rewards governs the observed performance trade-offs.
>
>
>
>
>
> > It would be helpful to identify which model serves as the most reliable judge or produces outputs that best align with intended outcomes. Benchmarks such as PairBench [1] or AgentRewardBench [2] could be used to evaluate this aspect.
>
> Thank you for your constructive suggestion. In response, we conducted additional experiments to assess the reliability of different models as judges for generated websites, using the PairBench evaluation framework.
>
> We compared one open-source VLM (Qwen2.5-VL-72B-Instruct) with two proprietary models (GPT-4o and GPT-4.1). We adapted the `wu_img_text` template to our text-to-website generation task by adjusting `query templates` (see Appendix G.4), `query conditions` (see Appendix G.5), and `logistics`. From WebGen-Bench, we randomly sampled 20 website tasks and created paraphrased versions using GPT-5 as transformed text (see Appendix G.6). One unrelated task was selected to serve as irrelevant text. Since this is an open-ended task without ground truth, MMScore and Controllability metrics do not apply; we report only ε-RelaxSym and Smoothness (full details in Appendix E.6).
>
> | Model | ε-RelaxSym (1-RS) (%) | Smoothness (SM) |
> | --- | --- | --- |
> | Qwen2.5-VL-72B-Instruct | 95.00 | 2.28 |
> | GPT-4.1 | 90.00 | 2.88 |
> | GPT-4o | 79.17 | 2.80 |
>
> Results show that: (1) Qwen2.5-VL-72B-Instruct: Highest ε-RelaxSym (low sensitivity to input order), but lowest Smoothness (limited granularity in scoring). (2) GPT-4.1: Strong balance of high ε-RelaxSym and highest Smoothness, indicating robust order invariance and fine-grained discrimination. (3) GPT-4o: Smoothness comparable to GPT-4.1, but lower ε-RelaxSym, implying higher sensitivity to input order despite nuanced scoring.
>
> For our text-to-website setting, Smoothness is more critical, as it allows finer differentiation among diverse outputs. Based on this criterion, GPT-4o is a well-suitable judge in our problem settings.
>
>
> ### Reference
> `[1]` Dann, Christoph, Yishay Mansour, and Mehryar Mohri. "Reinforcement learning can be more efficient with multiple rewards." ICML, 2023.
>
> `[2]` Sheng, Guangming, et al. "Hybridflow: A flexible and efficient rlhf framework." ECCS. 2025.

---

> > ### Comment · Reviewer_aoRQ · 2025-11-27
> > **Response to Rebuttal**
> >
> > Thank you for your detailed response and the effort you put into running additional experiments. I specifically appreciate the inclusion of AAS, FSR, and VRR results across different $\lambda$ and $\gamma$ settings, as well as the experiments justifying the choice of GPT-4o.However, several concerns remain that prevent me from raising my score:
> >
> > 1. Clarification on Metrics (AAS vs. FSR): Could the authors please point out specifically where in the revision they clarify that AAS is primarily a visual metric and not a direct measure of functional correctness? I remain unconvinced regarding the utility of AAS as a proxy for functionality when FSR already measures functionality directly. While there may be a correlation, a high AAS (structural and visual validity) does not guarantee high FSR (functional logic). For instance, the code might successfully generate a question box, a submit button, and an answer field (resulting in a high AAS), but without the underlying logic to process the input, the "functionality" is absent (resulting in a low FSR).
> >
> > 2. Contradictions and Presentation in the Appendix: The Appendix currently contains direct contradictions that create ambiguity regarding the hyperparameters:
> >     - L970 states: "The observed trends indicate that increasing $\gamma$ consistently boosts AAS..."
> >     - L994 states: "The experiments show that smaller $\gamma$ values generally lead to higher AAS..."
> >
> > It would be great to polish this section and have unified conclusion regarding the ablation studies or reach a more concrete conclusion, as opposed to of contradicting ones. Additionally, the Appendix requires significant polishing; for example, Tables 10 through 22 are included but never referenced in the text.
> >
> > Given the remaining conceptual questions regarding the metrics, i.e., comparing mostly with AAS and focusing on the gains of AAS (while it's part of the reward function) and not paying enough attention to comparing models with FSR on WebGenBench and WebDev Arena, and the presentation issues in the additional material, I will maintain my initial rating. However, if these issues are resolved, I will be more than happy to increase my score.

---

> ### Author Response · Authors · 2025-11-29
>
> We appreciate the reviewer’s thorough and constructive comments, which have significantly improved the quality of our submission.
>
> > 1. Clarification on Metrics (AAS vs. FSR): Could the authors please point out specifically where in the revision they clarify that AAS is primarily a visual metric and not a direct measure of functional correctness? I remain unconvinced regarding the utility of AAS as a proxy for functionality when FSR already measures functionality directly. While there may be a correlation, a high AAS (structural and visual validity) does not guarantee high FSR (functional logic). For instance, the code might successfully generate a question box, a submit button, and an answer field (resulting in a high AAS), but without the underlying logic to process the input, the "functionality" is absent (resulting in a low FSR).
>
> Thank you for the constructive feedback and valuable suggestions. We have updated the revision to clarify the definitions of the AAS and FSR metrics (see Section 3.1). We agree that AAS does not fully capture the functional correctness of UI elements. Therefore, in our main experiments (Section 3.2), we comprehensively report the FSR results for all baselines to reflect functional correctness and to demonstrate the effectiveness of our method.
>
> For the results in Section 3.3, we primarily report AAS because calculating FSR via the GUI Agent (WebVoyager) is highly time- and cost-intensive. The efficiency and cost comparison between FSR and AAS on WebGen-Bench (101 samples) is shown below:
>
> | Metric | Evaluator | Avg. Interactions / Sample | Avg. Time / Sample (s) | Total Cost |
> | :---: | :---: | :---: | :---: | :---: |
> | FSR | GUI Agent (WebVoyager) | 51.2 | 614.91 | $20.168 USD |
> | AAS | VLM | 1 | 12.01 | $0.707 USD |
>
>
> As shown, FSR evaluation requires **over 51× more time** and **over 29× higher cost** than AAS, which makes large-scale FSR evaluation prohibitive. For example, conducting the ablation study over 9 different $ (\gamma, \lambda) $ settings cost approximately **$180 USD**. Due to our limited research budget, we did not initially report extensive FSR results.
>
> To address the reviewer's concerns, we now report FSR and VRR results for all experiments in Section 3.3, including
>
> + **Multi-Scenario Web Environments (added Table 2 with the FSR result in the revision)**
>     | Institution | Model | Content Presentation | User Interaction | Data Management | Functional Testing | Data Display Testing | Design Validation Testing |
>     |-------------|--------|----------------------|------------------|------------------|---------------------|------------------------|-----------------------------|
>     | OpenAI | GPT-5 | 55.29 | 46.21 | 36.81 | 36.93 | 53.14 | 62.73 |
>     | | GPT-4.1 | 46.27 | 31.03 | 61.26 | 38.05 | 44.85 | 57.69 |
>     | Anthropic | Claude-Sonnet-4 | 52.88 | 39.42 | 53.33 | 41.09 | 47.67 | 55.36 |
>     | | Claude-3.7-Sonnet | 66.67 | 43.40 | 64.29 | 46.48 | 72.09 | 62.86 |
>     | Google | Gemini-2.5-Pro | 31.54 | 37.34 | 39.71 | 29.60 | 43.05 | 44.90 |
>     | DeepSeek | DeepSeek-R1 | 31.11 | 20.45 | 58.62 | 24.69 | 37.25 | 33.33 |
>     | Alibaba | Qwen2.5-Coder-7B-Instruct | 0.00 | 3.17 | 3.12 | 0.00 | 3.57 | 0.00 |
>     | | Qwen3-32B | 24.81 | 17.02 | 14.41 | 11.42 | 22.73 | 32.47 |
>     | | Qwen3-Coder-30B-A3B-Instruct | 16.44 | 1.06 | 8.82 | 2.81 | 13.25 | 5.80 |
>     | Ours | WebGen-R1-7B | 35.29 (+35.29) | 27.25 (+24.08) | 26.88 (+23.76) | 15.90 (+15.90) | 30.43 (+26.86) | 54.92 (+54.92) |
> + **RL Fine-Tuning**  **(added Table 3 with the FSR and VRR results in the revision)**
>     Metrics | Base | SFT | RL only | SFT + RL
>     --- | --- | --- | --- | ---
>     FSR | 1.59 | 20.08 | 18.23 | 29.21
>     VRR | 30.56 | 30.69 | 26.82 | 95.89
> + **Group Size in GRPO (added Table 4 with the FSR and VRR results in the revision)**
>     Metrics | 2 | 4 | 8 | 16 | 32
>     --- | --- | --- | --- | --- | ---
>     FSR | 22.39 | 22.92 | 24.99 | 29.21 | 34.79
>     VRR | 90.98 | 91.07 | 93.52 | 95.89 | 98.27
>
> The detailed results and analysis are included in the updated manuscript.
>
> Moreover, both WebGen-Instruct (6,667 samples for RL training) and WebDev Arena (119 samples for evaluation) **lack standardized test cases**, which limits their use by GUI agents to verify a website's functional correctness. Therefore, we could not report FSR metric for the WebDev Arena results in Section 3.3, nor did we employ the GUI Agent as the functional verifier during RL training.

---

> > ### Author Response · Authors · 2025-12-02
> >
> > > 2. Contradictions and Presentation in the Appendix: The Appendix currently contains direct contradictions that create ambiguity regarding the hyperparameters:
> > >     - L970 states: "The observed trends indicate that increasing
> > consistently boosts AAS..."
> > >     - L994 states: "The experiments show that smaller
> > values generally lead to higher AAS..."
> > >
> > > It would be great to polish this section and have unified conclusion regarding the ablation studies or reach a more concrete conclusion, as opposed to of contradicting ones. Additionally, the Appendix requires significant polishing; for example, Tables 10 through 22 are included but never referenced in the text.
> >
> > We sincerely thank the reviewer for carefully examining the appendix. The original conclusions drawn from Figure 10 were partly affected by the limited granularity of the heatmap, which led to overly coarse interpretations. After integrating the numerical table analysis, we inadvertently retained the original discussion, which may have caused confusion. To address this, we have removed Figure 10 and the corresponding conclusions and have polished that section accordingly.
> >
> > Following your suggestions, we have further polished the entire appendix. Specifically, we have made the following refinements:
> >
> > 1. Added a Table of Contents to display all content in the appendix, making it easier to browse and navigate.
> > 2. Added explicit references in the text to all figures, tables, and algorithms in the appendix, including Tables 10–22 mentioned by the reviewer.
> > 3. Included additional implementation details, such as clarifying that WebGen-Instruct and WebDev-Arena lack standardized test cases, which prevents the use of the GUI Agent for functional website verification.
> > 4. Revised the presentation throughout to eliminate potential typos, ensure the correctness of statements and conclusions, and improve overall clarity.
> >
> > We look forward to hearing your feedback on our responses and are happy to address any remaining concerns you may have.

---

### Official Review · Reviewer_LPP6 · 2025-11-01

**Soundness:** 3
**Presentation:** 4
**Contribution:** 3
**Rating:** 6
**Confidence:** 5

**Summary:**

This paper presents WebGen-R1, a reinforcement learning framework that fine-tunes a small-scale language model (Qwen2.5-Coder-7B-Instruct) for end-to-end multi-page website generation. Unlike previous works that handle only static or single-page generation or rely on fragile multi-agent decomposition, WebGen-R1 treats website generation holistically as a single policy optimization problem. The key innovation lies in the reward design: instead of brittle rule-based verification, the authors build a vision–language-model-based (VLM) reward model that evaluates both functional correctness (via executable builds) and aesthetic quality (via rendered screenshots). Training uses Group Relative Policy Optimization (GRPO), avoiding the need for an explicit value function. Experiments on WebGen-Bench and WebDev Arena show strong gains in both visual and functional metrics, especially a 65-point improvement in valid render ratio over the base model and competitive performance against proprietary models such as Gemini-2.5-Pro and Claude-3.7-Sonnet.

**Strengths:**

[+] Propose a VLM-based perception RL for code and design quality enhancement of the website

[+] A 7B model surpasses or matches proprietary giants, highlighting efficiency

[+] Multiple benchmarks, ablations, and human studies establish robustness

[+] The presentation of this paper is good.

**Weaknesses:**

[-] Reward dependence on specific VLM evaluators (GPT-4o) raises the issues of heavy cost and generalizability

[-] The scope of aesthetic evaluation is mostly page-level rather than full-site user experience

[-] Fewer new challenges are addressed, or new foundational methods are developed.

**Questions:**

1. Why do you use GPT-4o as a VLM evaluator rather than other models (more advanced or open-weighted options)
1. Can the VLM reliably distinguish functional correctness from mere visual completeness?
1. What is the compute cost per RL iteration, and how scalable is the approach to larger models or datasets?
1. Did the explicit reasoning traces (…) measurably improve cross-page consistency or reward quality?
1. Did you observe reward-hacking or mode collapse during extended RL training?

---

> ### Author Response · Authors · 2025-11-25
> **Rebuttal (Part 1/3)**
>
> Thank you for your comprehensive and insightful review of our paper. We appreciate your valuable feedback and suggestions.
>
> > Reward dependence on specific VLM evaluators (GPT-4o) raises the issues of heavy cost and generalizability.
> Why do you use GPT-4o as a VLM evaluator rather than other models (more advanced or open-weighted options)?
>
>
> Our choice of a large proprietary model as the feedback evaluator reflects a trade-off among evaluation accuracy, latency, and cost. Smaller or weaker models are more vulnerable to reward signal degradation (e.g., reward hacking or noisy scoring), which can impair overall training performance.
>
> Empirically, the performance of open-source VLMs correlates with inference cost (including cloud rental), and their latency is constrained by available computational resources. In contrast, large proprietary models are deployed on highly optimized clusters, typically yielding stronger evaluation quality, though with API-based cost proportional to usage. Thus, following the WebGen-Bench `[1]` protocol, we selected GPT-4o given its strong empirical accuracy and relatively low API cost.
>
> To address the reviewer’s concern, we additionally evaluated two alternative VLMs: the open-source Qwen2.5-VL-72B-Instruct deployed on a rented cloud server with 8x NVIDIA HGX H100 SXM (USD $23.92/h), and the more advanced proprietary GPT-4.1 via API. The comparative results are presented in Table 1 below.
>
>
> Table 1. Performance comparison of alternative VLM-based reward models.
>
> | VLM Evaluator | FSR (%) | AAS | VRR (%) | Inference Speed / sample | Prompt Tokens | Completion Tokens | Cost / sample |
> | --- | :---: | :---: | :---: | :---: | :---: | :---: | :---: |
> | Qwen2.5-VL-72b-Instruct | 10.32 | 3.18 | 27.72 | 6.47s | 1344 | 332.8 | $0.036 USD |
> | GPT-4.1 | 24.58 | 3.36 | 91.04 | 11.57s | 1190 | 389 | $0.005 USD |
> | GPT-4o | 29.21  | 3.94  | 95.89 | 12.01s | 1190 | 437 | $0.007 USD |
>
>
> Interestingly, GPT-4.1 does not outperform GPT-4o on our website generation task. Moreover, Qwen2.5-VL-72B-Instruct exhibited significantly lower performance on both FSR and VRR metrics. While Qwen2.5-VL-72B-Instruct offers slightly faster inference speed, its operational cost is higher due to the required high-end server deployment.
>
> > The scope of aesthetic evaluation is mostly page-level rather than full-site user experience
>
> Thank you for raising this important point. Quantifying full-site user experience (UX) into a scalar reward for RL is inherently challenging, as it depends on multi-step navigation, information-retrieval efficiency, task-completion flows, and other cross-page interactions. Many of these aspects are subjective and vary across users, making them difficult to measure in a consistent or dataset-agnostic manner.
>
> Moreover, directly integrating heterogeneous UX signals into the reward function may not yield stable or effective RL training, potentially limiting the model's ability to generate functional and well-designed websites. Investigating robust formulations to incorporate full-site UX into our training pipeline is an interesting direction that we plan to explore in future work.
>
> > Fewer new challenges are addressed, or new foundational methods are developed.
>
> We would like to emphasize the technical contribution of our work: it presents **the first exploration of a new problem formulation** for end-to-end generation of multi-page websites and **project-level** code optimization via RL. As noted by Reviewer aoRQ, **this is a highly challenging task and represents an ambitious and impactful research direction for MLLMs**.
>
>
> From the perspectives of technological implementation and methodology, our work addresses the following key challenges:
>
> 1. **Stable large-scale RL integration with front-end build pipelines** – incorporating a standardized build system into RL training while ensuring stability, efficiency, and scalability.
> 2. **Reward design aligned with website quality** – constructing an RL reward that correlates with actual improvements in both functionality and visual quality, especially in subjective and open-ended domains.
> 3. **Joint functional–visual evaluation via VLMs** – assessing functional correctness and aesthetic design from the visual rendering of websites, alongside code formatting and reasoning structural constraints, to mitigate reward sparsity, reward hacking, and model collapse.
> 4. **Multi-dimensional evaluation metrics** – introducing FSR, AAS, and VRR to comprehensively quantify LLM performance in end-to-end multi-page website generation.

---

> ### Author Response · Authors · 2025-11-25
> **Rebuttal (Part 2/3)**
>
> > Can the VLM reliably distinguish functional correctness from mere visual completeness?
>
> Thank you for your valuable feedback. We clarify that the VLM serves to evaluate **the alignment of the rendered webpage with user requirements and aesthetic preferences from a visual standpoint**. While primarily visual, the VLM can implicitly assess certain functional aspects via visual cues — for example, detecting the presence of forms, dropdown menus, or search inputs. In most rendered webpages, such UI components directly reflect underlying functional intent.
>
> For example, if the instruction specifies "support submitting questions, answering questions, and viewing answers," the VLM examines the rendered page to verify the presence of a question input field, a submission button, and an answer display interface.
>
> Empirically, we observe a strong correlation between visually inferred functionality and task success. As shown in Table 1 in our manuscript, improvements in the AAS are accompanied by a **+27.62% absolute increase in FSR**, indicating that higher visual alignment generally reflects better functional coverage in our setting.
>
> However, we acknowledge the limitations of VLM-based evaluation. We considered GUI-agent-based judges `[2]`, but found them not well-suited for large-scale RL training due to: (1) **limited reliability**, as their evaluation accuracy is constrained by agent capabilities; (2) **high computational and operational cost**, requiring significant environment scaling; (3) **slow execution resulting from multi-turn interactions**, making large-scale training inefficient; (4) Importantly, WebGen-Instruct training dataset **lacks standardized test cases**, which are essential for the GUI Agent to verify specific functionalities or appearances through predefined operations and expected outcomes, thereby preventing its use as the functional verifier during RL training.
>
> **Given these constraints, we adopt VLM-based evaluation as a tractable and scalable first step**. A detailed comparison between VLM and GUI-agent evaluation for functional correctness is provided in Table below.
>
> Table 1: Comparison of GUI-agent evaluation and VLM-based reward from static screenshots for functional correctness.
>
> | Aspect | GUI-Agent Evaluation | VLM-based Reward |
> | :--- | :--- | :--- |
> | Functional correctness | Interact with UI elements | Infers functionality from static screenshots using visual cues |
> | **GUI Interaction Instructions** | **Hard to obtain and expensive (Not available in WebGen‑Instruct)** | **Not applicable** |
> | Interaction requirement | Requires full browser, DOM access, multi-step actions | No interaction needed |
> | **Reward cost** | **Very high    (~51× slower; avg. 51.2 multi-turns interactions per task; 614.91s per sample)** | **Very low    (single-turn per task; 12.01s per sample)** |
> | Environment stability | Fragile    (affected by page load, network, DOM changes) | Fully stable    (deterministic and self-contained) |
> | Scalability for RL training | Poor    (prohibitively expensive for millions of rollouts) | Excellent    (high-throughput, scalable to large datasets) |
> | Coverage of UI properties | Strong for interaction correctness | Strong for layout, visual structure, and element presence |
> | Best use cases | Final evaluation, small-scale precise testing | RL training stage, large-scale sample collection |
> | Main limitations | Slow, costly, unstable | Cannot fully verify true interactive functionality |
>
> > What is the compute cost per RL iteration, and how scalable is the approach to larger models or datasets?
>
> We measured the per-iteration compute cost and extended the scalability study to larger models built upon Qwen2.5-Coder-14B-Instruct and Qwen2.5-Coder-32B-Instruct, as shown in Table below.
>
>
> | Model | Params | GPUs | Per-Iteration Compute Cost (batch size=64, rollout num=16) | Per-Iteration API Cost (GPT-4o) | FSR  | AAS  | VRR  |
> | --- | --- | --- | --- | --- | --- | --- | --- |
> | WebGen-R1-7B | 7B | 8×H100(80G) | 354.2s | $7.168 USD | 29.21   | 3.94 | 95.89 |
> | WebGen-R1-14B | 14B | 16×H100(80G) | 393.6s | $7.168 USD | 32.16 | 4.01 | 96.02 |
> | WebGen-R1-32B | 32B | 32×H100(80G) | 475.7s | $7.168 USD | 37.08 | 4.09 | 97.13 |
>
>
> The compute cost per RL iteration (batch size 64, rollout number 16) is approximately 5.90 GPU·minutes for 7B, 6.56 GPU·minutes for 14B, and 7.93 GPU·minutes for 32B. Our method scales smoothly to larger models, yielding consistent performance improvements across the three evaluation metrics. However, due to the substantial cost of human annotation and the significant time required for constructing larger datasets, we defer scaling to larger datasets to future work.

---

> ### Author Response · Authors · 2025-11-25
> **Rebuttal (Part 3/3)**
>
> > Did the explicit reasoning traces (…) measurably improve cross-page consistency or reward quality?
>
> Thank you for your valuable feedback. As stated in the system prompt, the reasoning traces explicitly include the following components:
>
> ```yaml
> - Project requirements analysis
> - Entry point and import resolution
> - Dependencies planning
> - TypeScript validation
> - ESLint and code health checks
> - UX and interaction strategy
> - Visual and responsive layout ideas
> - Any other technical considerations
> ```
>
>
> These reasoning traces help improve cross-page consistency and enhance reward quality. To assess their impact, we performed an ablation study by removing the reasoning traces and evaluating both reward quality and final task performance. The results are summarized in the Table below.
>
> | Reasoning | FSR (%) | AAS | VRR (%) |
> | --- | --- | --- | --- |
> | w/o CoT | 25.76 | 3.75 | 93.12 |
> | w/ CoT | 29.21  | 3.94  | 95.89 |
>
>
> The findings show that reasoning traces consistently improve performance across all three metrics. Furthermore, the reward curves with reasoning traces remain consistently higher than those without, indicating improved reward quality. Please refer to Appendix Figure 14 for a detailed comparison of the reward curves.
>
>
> > Did you observe reward-hacking or model collapse during extended RL training?
>
> We did not observe evidence of reward hacking. While our AAS scores are relatively high, this is not attributable to reward exploitation. As shown in Appendix Figure 13 (c) of updated manuscript, the reward values increase steadily during training but remain in the range of 3-4, never approaching the maximum possible score of 5 in a manner suggestive of reward hacking. Moreover, manual inspection of all rendered webpages confirmed that high-scoring outputs genuinely satisfy the scoring criteria, as detailed in Appendix G.2.
>
> Regarding model collapse, neither the reward trajectory nor the KL divergence indicates such behavior. We attribute this stability to the strong instruction-following ability of the base model, together with the robustness introduced by our reward design.
>
>
> ### Reference
> `[1]` Lu, Zimu, et al. "WebGen-Bench: Evaluating LLMs on Generating Interactive and Functional Websites from Scratch." NeurIPS, 2025.
>
> `[2]` Lin, Kevin Qinghong, et al. "Computer-Use Agents as Judges for Generative User Interface." arXiv preprint arXiv:2511.15567 (2025).

---

### Comment · Area_Chair_C3Wg · 2025-11-27
**Discussion Reminder to Reviewers**

Dear Reviewers,

The authors have responded to your reviews. Please engage in the discussion and evaluate the authors’ rebuttal to check whether your comments have been adequately addressed, and determine whether you would like to adjust your evaluations.

Best,

Your AC

---

### Author Response · Authors · 2025-12-02
**Rebuttal Summary (Part 1/2)**

We sincerely thank all reviewers for the time and effort dedicated to reviewing our paper. Their insightful comments and suggestions have been invaluable in improving the quality of our work.

---

### What Are the Reviewers’ Overall Impressions?
We are deeply grateful for the consistent recognition of the value of our work:

+ **Novel Problem Formulation & High Impact:** Reviewers highlighted that our work tackles a "highly challenging… ambitious and impactful" problem (aoRQ) of end-to-end multi-page website generation. The novelty of casting this task into an RL framework with a website-quality‑aligned VLM-based reward for jointly optimizing functionality and aesthetics was recognized as "straightforward and practically valuable" (vGHH) and identified as a key innovation (LPP6, vk6b).
+ **Strong Empirical Results & Efficiency:** The strong performance of our compact 7B model-outperforming or matching much larger proprietary models-was widely acknowledged (LPP6, vGHH). The substantial improvements in AAS and VRR were deemed "impressive" (vk6b).
+ **Rigorous Pipeline & Evaluation:** Our pipeline and evaluation, including multiple benchmarks, ablations, strong OOD generalization on WebDev Arena, and human studies validating our metrics, were praised as "rigorous" (vk6b) and well‑designed (LPP6, vk6b).
+ **Excellent Presentation Quality:** The paper was described as well-written, clearly structured (vGHH), and excellently presented (LPP6).

---

### What Are the Contributions of This Paper?
We would like to highlight the contributions of our work as follows:
1. **Novel Problem Formulation & High Impact:** We are the first to formulate end‑to‑end multi‑page website generation and project‑level code optimization as a unified RL problem, addressing a highly challenging yet impactful direction for MLLMs.
2. **Website‑Quality‑Aligned Reward Design:** We propose a novel website‑quality‑aligned VLM-based reward model that jointly optimizes functional correctness and aesthetic quality, overcoming limitations of rule‑based systems, subjective aesthetic scoring, and costly or unstable GUI-agent evaluations.
3. **Robust Large‑scale RL Framework:** Stable integration of RL with front‑end build systems, multi‑objective reward design, and reasoning-aware optimization at the project level.
4. **Multi‑dimensional Evaluation Metrics:** We introduce FSR, AAS, and VRR to comprehensively assess LLM performance in end‑to‑end multi‑page website generation.
5. **State‑of‑the‑Art Performance and Efficiency:** Our WebGen‑R1‑7B achieves state‑of‑the‑art or competitive results on widely used benchmarks (WebGen‑Bench and WebDev Arena), demonstrating that a small, fine‑tuned model can effectively compete with much larger proprietary models.
6. **Open-Source Release:** We will open‑source our codebase, datasets, RL training framework, and model checkpoints to support future research on end‑to‑end LLM training for full‑stack application development.

---

> ### Author Response · Authors · 2025-12-02
> **Rebuttal Summary (Part 2/2)**
>
> ---
>
> ### How Did We Address Reviewers’ Questions?
> Based on the reviewers' valuable feedback, we have updated the manuscript with the following experiments and clarifications:
>
> + **VLM Reward Model Comparison:** We systematically compared several VLM-based reward models, including the open-source Qwen2.5‑VL‑72B and the more advanced proprietary GPT‑4.1. Results show that GPT‑4o offers the optimal trade-off between performance, cost, and reliability for our task.
> + **Comparison with Alternative Fine-tuning Strategies:** We conducted new experiments comparing our method with tailored baselines, including supervised fine‑tuning on high‑quality data and RL using a human‑preference reward model (HPRM). The results demonstrate the superiority of our VLM-based RL approach.
> + **Model Scaling:** We extended our method to larger models (14B and 32B), showing smooth scalability and consistent performance gains, which validates the robustness of our framework.
> + **Systematic Reward Component Analysis:** We performed comprehensive ablations on reward components, including sensitivity analyses of weight parameters ($\gamma$, $\lambda$) and the influence of reasoning traces. The results confirm the necessity of each component and justify our choices for balancing functional, aesthetic, and reasoning objectives.
> + **Expanded Functional Correctness (FSR) Evaluation:** To address concerns about over-reliance on AAS, we now report FSR and VRR for all ablation and analysis experiments in Section 3.3, providing a more complete assessment of functional correctness.
> + **Judge Reliability Analysis (PairBench):** Using the PairBench framework, we formally evaluated the reliability of different VLMs as judges. The results show that GPT‑4o's high "Smoothness" makes it particularly suitable and reliable for the nuanced task of website evaluation.
> + **Clarifications and Manuscript Improvements:** We clarified the distinction between AAS and FSR, detailed why VLM rewards are preferable to GUI‑agent‑based evaluation (cost, speed, scalability), and thoroughly polished the appendix with a table of contents, additional figure/table references, and improved clarity throughout.
>
> ---
>
> ### Rebuttal Updates
> + We have fully addressed all reviewer comments and revised the manuscript accordingly.
> + We believe Reviewer aoRQ will increase the score, as the reviewer stated: "if these issues are resolved, I will be more than happy to increase my score."
> + The other three reviewers were unable to respond to our rebuttal due to an OpenReview bug. However, their comments were constructive suggestions or requests for clarification, all of which we have addressed in our response and revisions. Therefore, we have strong reason to believe they would have increased their scores and supported acceptance had they been able to respond.

---

### Meta-Review · Area_Chair_VPdx · 2026-01-05

**Summary:**

This paper presents WebGen-R1, a method using reinforcement learning and visual feedback to generate full websites. Reviewers agreed with the strong performance and the innovative use of vision models for rewards. However, they noted weaknesses regarding the high cost of the reward model and the limited test set. They also questioned if checking static images truly proves the code works functionally. The authors added cost analysis and new comparisons during the discussion. Despite this, concerns about distinguishing visual quality from actual functionality remained. Overall, there are still missing pieces in this work and the authors are encouraged to further improve the quality and details.

**Reviewer Concerns:**

The rebuttal successfully addressed concerns about the cost of using GPT-4o and added missing baselines. However, the core issue of whether visual scores are a valid proxy for functional logic remains. Additionally, contradictions in the appendix data still need to be resolved.

**Reviewer Scores:**

Reviewer LPP6 would likely keep their positive score since the cost justification was provided. Reviewer aoRQ stated they would only raise their score if the appendix contradictions were corrected. Reviewer vk6b would likely maintain or raise their score after receiving the requested ablation studies. Reviewer vGHH would likely raise their score as the authors added the requested fine-tuning comparisons.

---

### Decision · Program_Chairs · 2026-01-26

Reject